# Local human impacts disrupt depth-dependent zonation of tropical reef fish communities

Laura E. Richardson [1] ✉, Adel Heenan[1], Adam J. Delargy [1,2], Philipp Neubauer[3], Joey Lecky [4,5], Jamison M. Gove [4], J. A. Mattias Green [1], Tye L. Kindinger [4], Kurt E. Ingeman[4,6] & Gareth J. Williams [1]

The influence of depth and associated gradients in light, nutrients and plankton on the ecological organization of tropical reef communities was first described over six decades ago but remains untested across broad geographies. During this time humans have become the dominant driver of planetary change, requiring that we revisit historic ecological paradigms to ensure they capture the dynamics of contemporary ecological systems. Analysing >5,500 in-water reef fish surveys between 0 and 30 m depth on reef slopes of 35 islands across the Pacific, we assess whether a depth gradient consistently predicts variation in reef fish biomass. We reveal predictable ecological organization at unpopulated locations, with increased biomass of planktivores and piscivores and decreased primary consumer biomass with increasing depth. Bathymetric steepness also had a striking influence on biomass patterns, primarily for planktivores, emphasizing potential links between local hydrodynamics and the upslope propagation of pelagic subsidies to the shallows. However, signals of resource-driven change in fish biomass with depth were altered or lost for populated islands, probably due to depleted fish biomass baselines. While principles of depth zonation broadly held, our findings expose limitations of the paradigm for predicting ecological dynamics where human impacts confound connections between ecological communities and their surrounding environment.

Ecological paradigms inform the understanding and management of natural systems but are limited by two fundamental issues. First, biophysical processes governing ecological organization often occur at regional and continental scales[1–3], inherently introducing scale-dependent patterns and heterogeneity in observed local community structure[4–6]. To understand ecological generality, a large enough lens across a landscape or seascape is required to encompasses these processes[7]. However, many influential paradigms were developed from single-point studies in the mid-twentieth century (for example, refs. [6–8]). Until recently our ability to test generalizable predictions on ecological organization in hierarchically structured ecosystems has been limited by a lack of spatially comprehensive data and accessible

[1]School of Ocean Sciences, Bangor University, Menai Bridge, UK. [2]School for Marine Science & Technology, University of Massachusetts Dartmouth, Dartmouth, MA, USA. [3]Dragonfly Data Science, Wellington, New Zealand. [4]Pacific Islands Fisheries Science Center, National Oceanic Atmospheric Administration, Honolulu, HI, USA. [5]IBSS Corporation, Silver Spring, MD, USA. [6]Department of Environmental Studies, Linfield University, McMinnville, OR, USA. ✉e-mail: l.richardson@bangor.ac.uk

statistical tools[7,9]. Second, escalating anthropogenic impacts confound natural drivers of ecological organization[10–12] such that humans are now considered the dominant force of planetary change[13]. Where theories are founded on a premise that ecological organization occurs in isolation of anthropogenic forcing[14,15], the predictive capacity of historical paradigms can breakdown (for example, island biogeography theory[16]), requiring thorough reassessment of their applicability in this era of rapid change[14,17].

Ecological zonation—the distribution of organisms across space—represents one of the oldest ecological concepts[8,18,19]. Here, we revisit this basic principle in the context of resource-driven depth zonation of tropical coral reef communities—the distribution of reef fish biomass among distinct trophic groups. Depth was recognized as a fundamental structuring force over six decades ago[20–23] and recently shown to be an important predictor of fish diversity[24]. Ocean-facing reefs are formed along a bathymetric depth gradient with covarying gradients in availability of sunlight, water temperature, surface wave energy and nutrients[21,25–27]. With increasing depth, there are predictable changes in energetic resource supply. Light for primary producers diminishes[25] but particulate foods and nutrients derived from deeper water that support higher trophic levels such as planktivorous predators, secondary consumers and piscivores[28–30], increase with depth with increased proximity to sources of upwelling[26,31,32]. The degree to which upwelling can boost shallow-water (<30 m depth[33]) primary production is, in turn, determined by the steepness of the reef slope—either facilitating or impeding the upslope propagation of deeper nutrient-rich waters to the shallows[26,31,32]. Where bathymetry mediates local hydrodynamics around islands, upwelling processes can concentrate in specific areas[34,35], creating intra-island variation in pelagic resource supply[36]. At larger spatial scales, cross-regional gradients in primary production[3] probably underscore background levels of local depth-dependent resource supply. However, despite these important structuring forces and a general acceptance of an effect of depth, we have maintained a limited understanding of resource-driven depth zonation on shallow coral reefs. Early observations were born of single-location point studies and to date the applicability of the theory remains untested across broad geographies, limiting our understanding of how this ubiquitous physical characteristic of tropical reefs influences natural ecological biomass baselines[37]. Modern-day island reefs span vast ocean expanses and are among some of the most biodiverse, socio-economically important but also human-impacted ecosystems on our planet[17,38]. Traversing numerous biophysical gradients that mediate ecological organization across scales[6], they provide a unique focal system to explicitly test early ecological theories across scales and assess whether classic paradigm-based science and management of contemporary coral reefs should be adapted[14,17,39]. Reefs are exposed to ocean warming and acidification and a suite of local human impacts that deplete biological communities and degrade habitats[14,17]. Some of these human activities are also stratified by depth. Fishing, for example, tends to concentrate in shallower depths and disproportionately targets distinct trophic groups of reef fishes such as large-bodied piscivores, herbivores and planktivores[40–42]. With human impacts on coral reefs globally widespread[38], it is unknown how anthropogenic forcing interacts with natural energetic resource supply across depth and therefore whether a classic depth zonation model is effective for predicting the ecological organization of modern reefs.

Establishing meaningful baselines from which to effectively measure change requires spatially comprehensive replication at the unimpacted end of the intact-to-degraded spectrum[37]. Using a standardized Pacific-wide set of reef fish surveys[43], composite data on bathymetric steepness and hierarchical statistical models, we test whether depth zonation patterns in fish biomass are generalizable on tropical coral reefs across broad geographies and compare patterns in locations with and without local human populations. To explicitly assess generality, we isolate the study focus to test a framework of a priori defined proposals of the effects of depth, bathymetric steepness and human population status on the biomass of reef fishes across a broad spatial extent characterized by known environmental and anthropogenic variation[3,44]. In doing so, we intentionally exclude other known influential biophysical and anthropogenic covariates on reef fish biomass (for example, refs. 44,45) to test the predictive capacity of depth at an ocean-basin scale on the biomass of fishes grouped by their major dietary sources[46]—primary consumers, planktivores, secondary consumers and piscivores. We link data from 5,525 visual surveys of 35 islands and atolls (hereafter 'islands') across five distinct ecoregions[47] spanning ~4,600 km latitude and 6,800 km longitude in the Pacific (Supplementary Table 1) with prior information on unfished biomass baseline estimates to integrate existing global-scale research[48]. We do this to (1) quantify gradients in fish biomass across shallow reef depths (1–30 m) and steepness; (2) compare depth zonation patterns at unpopulated versus human-populated locations; and (3) estimate the scale-dependency of observed patterns by quantifying variability in fish biomass across reefs, islands and ecoregions.

## Results

Hierarchical Bayesian regression estimates showed that for unpopulated islands there was evident depth zonation in the biomass of all trophic groups of reef fish across or within the 0–30 m depth range (Figs. 1 and 2, Table 1, Extended Data Fig. 1 and Supplementary Tables 4 and 5). However, the effect of depth on fish varied by trophic group (Fig. 2), evidenced by differences in estimated effect sizes ($\beta$, a model vector of population-level regression coefficients) (Fig. 1 and Supplementary Table 4) and probability ($P$) from model posterior draws (Table 1 and Supplementary Table 5). The biomass of planktivores and piscivores increased across the 0–30 m depth gradient with high probability ($P(\beta$ depth $> 0) = 0.98$, both), while primary consumer biomass decreased with increasing depth from 0 to 30 m ($P(\beta$ depth $> 0) = 0.93$; Fig. 2, Table 1 and Supplementary Table 4). The biomass of secondary consumers increased from 0 to 10 m depth, plateaued from 10 to 20 m and then decreased from 20 to 30 m (Table 1 and Fig. 2). Combining all trophic groups, total fish biomass increased from 0 to 20 m depth ($P(\beta$ depth $> 0) = 0.75$; Supplementary Table 5) and plateaued from 20 to 30 m (Table 1). Overall, human population status had a negative effect on the biomass of all trophic groups (all: $P(\beta$ population status $< 0) = 1.00$; Fig. 1 and Supplementary Table 5), with posterior estimates of fish biomass of populated islands consistently lower than for unpopulated islands across 0–30 m depth (Fig. 2). The greatest negative effect of human population status was on the biomass of piscivore reef fish (Figs. 1 and 2 and Supplementary Table 4).

After establishing the direction of change in fish biomass per trophic group over an increasing depth range (Fig. 2, Table 1 and Supplementary Table 4), we next sought to determine the magnitude of change in fish biomass across the study depth range, across nominally shallow (0–10 m), mid-depth (10–20 m) and deep (20–30 m) sites. We examined the density distributions from the model posteriors of predicted biomass changes across each depth bin and for each trophic group (Fig. 3a) and compared the difference in depth zonation measured as absolute change in biomass at populated versus unpopulated islands (Fig. 3b). For populated islands, the magnitude of change in fish biomass across depth was reduced (total biomass, planktivores, piscivores and secondary consumers) or not observed (primary consumers) relative to patterns observed at unpopulated islands (Figs. 2 and 3a,b and Supplementary Table 7). Total biomass increase was predominantly lower at populated locations across 0–20 m, piscivore and planktivore biomass increases were lower across 0–30 m and secondary consumer biomass increase was reduced within the shallow 0–10 m range (Fig. 3a,b and Supplementary Table 7). We observed little change in primary consumer biomass across depth for populated islands (Fig. 3a,b and Supplementary Table 7). Examining zonation as a function of proportionate change in biomass across depth, there was greater

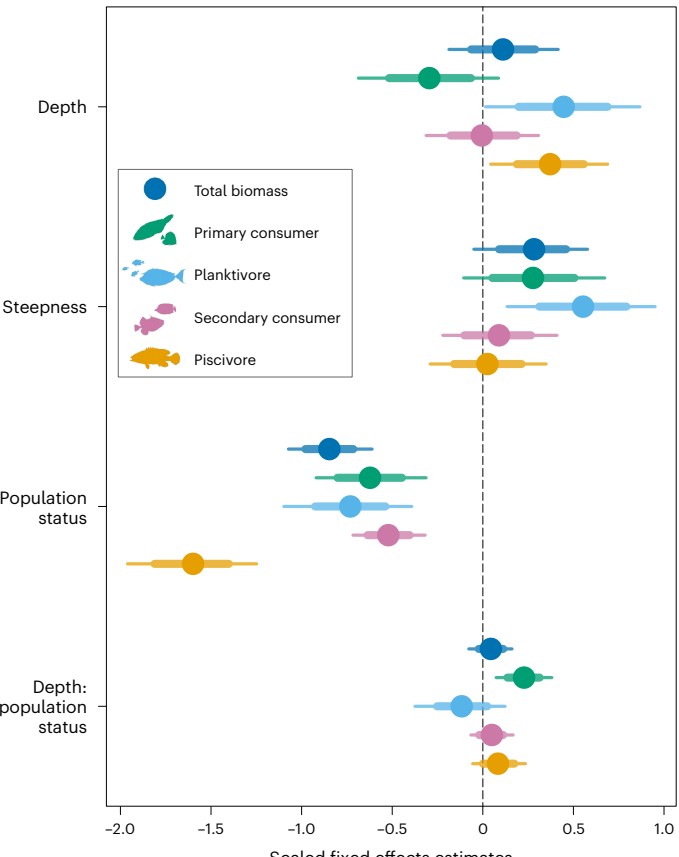

**Fig. 1 | Relationships between coral reef fish biomass of distinct trophic groups and overall effects of shallow reef depth, bathymetric steepness and human population status of islands.** Population status indicates the effect of populated by humans versus unpopulated. Effect sizes are scaled and include the interaction of depth with population status (depth:population status). Points represent posterior median estimates from Bayesian hierarchical models testing for an effect of each explanatory variable on reef fish biomass, with 75% (thick lines) and 95% (thin lines) percentiles. Explanatory variables were mean-centred and scaled by 1 s.d. to facilitate comparisons of effect sizes among them. For estimates of hurdle components (piscivore and planktivore models), see Extended Data Fig. 1 and Supplementary Table 4. Unadjusted Bayesian conditional $R^2$ values and 95% CI: total fish biomass, 0.55 (0.44–0.67); primary consumers, 0.54 (0.51–0.57); planktivores, 0.48 (0.24–0.67); secondary consumers, 0.37 (0.31–0.47); and piscivores, 0.52 (0.38–0.62) (Supplementary Table 6 for marginal unadjusted $R^2$ estimates). Total $n$ = 5,525 SPC surveys (across 2,253 forereef sites, 35 islands and 5 ecoregions).

observed depth zonation for populated islands in the biomass of secondary consumers and piscivores than for unpopulated islands (Fig. 3c, Extended Data Fig. 2 and Supplementary Table 8). These inverse trends in zonation, indicating greater proportionate change with depth for populated islands, were probably driven by lower biomass baselines and higher incidences of zero-count observations across all fish groups; but most notably for piscivores at populated islands and in shallower depths than at unpopulated islands (Supplementary Table 9).

Incorporating site-level derived estimates of mean bathymetric steepness (°) from within a 400 m buffer radius into fish biomass models revealed confounding and variable effects of forereef steepness on the biomass of planktivores, primary consumers and total fish biomass ($P(\beta$ steepness > 0) ≥ 0.92; Fig. 4; Extended Data Fig. 1 and Supplementary Tables 4 and 5). Increased reef steepness from 0° to an initial threshold of approximately 5–10° strongly correlated with a twofold increase in biomass of planktivores, 25% increase in biomass

of primary consumers and 50% increase in total fish biomass (Fig. 4). Planktivore biomass increased between 0° and 30° with the sharpest increase between 0° and 5–10° and plateauing around 30°. Conversely, total biomass and primary consumer biomass plateaued around 10°, then declined from 30° (Fig. 4).

The proportion of variation in fish biomass explained by each spatial scale, quantified by extracting the posterior standard deviations of these modelled random effects, varied among trophic fish groups (Fig. 5 and Supplementary Tables 10 and 11; see Extended Data Fig. 3 for variation in island-level depth effects among ecoregions). For all trophic groups, except secondary consumers, there was greater variation in biomass at the site scale (25–52% variance) and ecoregion scale (25–66%) than at the island scale (8–22%), suggesting that intra-island and inter-regional processes are more influential on fish biomass distributions than those occurring among islands (Fig. 5 and Supplementary Table 10). The biomass of secondary consumers was most variable at the site scale (63%), relative to the island and ecoregion scales (20% and 17%, respectively; Fig. 5 and Supplementary Table 10). There were high probabilities across all fish groups that variation was greater at the site scale than island scale ($P$(sdsSITE > sdsISLAND) ≥ 0.99; Supplementary Table 11). For planktivores, variation in biomass was proportionately greatest at the ecoregion scale (66% variance). For all other groups, except planktivores and piscivores, site-scale variance was greater than at the ecoregion scale ($P$(sdsSITE > sdsECOREGION) ≥ 0.85) (Fig. 5 and Supplementary Tables 10 and 11).

## Discussion

While the structuring force of depth on reef ecology featured among the earliest descriptions of tropical coral reefs (for example, zonation in species composition)[20,22,23], these observations were restricted to single-point locations. To date, the generality of resource-driven depth zonation in fish biomass remains untested across broad geographies. Here, we show that in the absence of local human populations there are predictable changes in tropical fish biomass with depth that track expected gradients in energetic resource supply to reefs[25,28]. These patterns hold true across the study area which spans distinct biogeographic regions, with high spatial consistency across islands and ecoregions (Extended Data Fig. 3) despite varying spatial heterogeneity in fish biomass among trophic groups, suggesting the role of distinct scale-dependent drivers. Recent work details declining patterns of reef fish diversity with increasing depth from the shallows to the mesophotic zone (maximum 150 m depth)[24]. We build on these findings by revealing a common degree of ecological organization in relation to both depth and bathymetric steepness across geographically distinct reefs. However, while there was evidence of resource-driven depth zonation in some groups at human-populated islands, the absolute change in biomass with depth relative to unpopulated islands was much reduced (planktivores, secondary consumers and piscivores) or was absent (primary consumers). Where humans have fundamentally reset standing biomass baselines[48], changes to naturally observed zonation may signal biological depletion confounding the predictive capacity of depth-dependent gradients in resource supply. These findings support calls for revisiting and potentially updating twentieth century ecological paradigms (for example, island biogeography theory[16]) that may no longer capture ecological patterns and processes in a human-dominated world[14].

At geographically distinct unpopulated islands, we show that reef fish biomass of all broad trophic groups correlated predictably and relatively consistently across depth, despite underlying variation in biophysical drivers known to affect standing reef fish biomass[3,29,45]. Focussing on the shallowest 0–30 m, we show that secondary consumer biomass increased between 0 and 15 m then plateaued. This diverse trophic group includes macro and sessile invertivores and omnivores whose biomass can vary differentially with depth at local scales[33,49]. However, broad energetic pathways are governed by primary

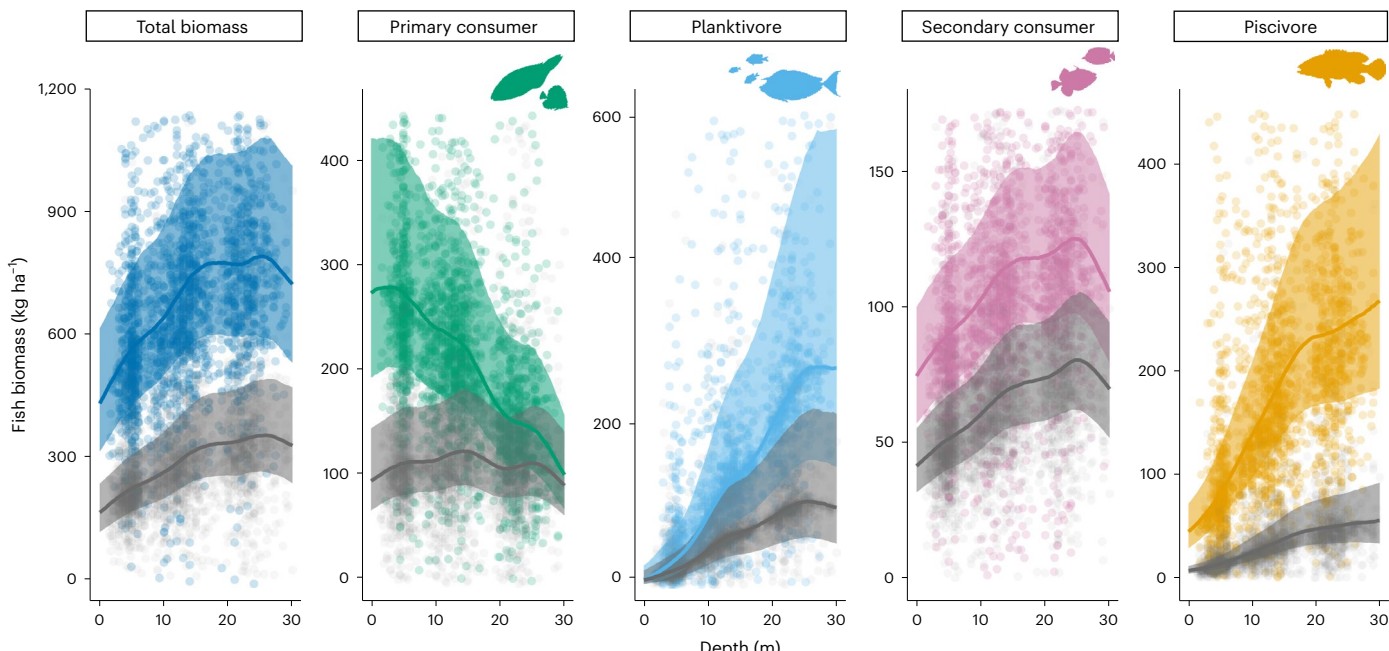

**Fig. 2 | Coral reef fish biomass across a shallow depth gradient at unpopulated (colour) and populated (grey) islands.** Estimates represent conditional posterior medians (lines), 75% percentiles (shaded areas) and partial residuals (points) at the study mean value of bathymetric steepness. The *y* axis is limited to 1.05× the maximum value of the 75% CI so partial residuals exceeding axis limits are not displayed. Total *n* = 5,525 SPC surveys (across 2,253 forereef sites, 35 islands and 5 ecoregions).

**Table 1 | Probabilities of an increase in fish biomass across specified depths at unpopulated (U) and populated (P) islands**

| Depth (m) | Population status | Total biomass | Primary consumer | Planktivore | Secondary consumer | Piscivore |
|---|---|---|---|---|---|---|
| 0 versus 10 | U | **0.95** | <u>**0.81**</u> | **1.00** | **0.87** | **1.00** |
|  | P | **0.96** | 0.65 | **1.00** | **0.90** | **1.00** |
| 10 versus 20 | U | **0.95** | <u>**0.92**</u> | **1.00** | 0.63 | **1.00** |
|  | P | **0.98** | 0.58 | **0.98** | 0.73 | **1.00** |
| 20 versus 30 | U | 0.41 | <u>**0.95**</u> | **0.86** | <u>**0.81**</u> | **0.84** |
|  | P | 0.50 | 0.29 | 0.71 | 0.29 | **0.78** |

Probability estimates are derived from posterior model distributions comparing biomass at one fixed depth versus a deeper depth (0 versus 10 m depth), with steepness held at the study mean value. Probabilities ≥75% highlighted in bold. Probabilities ≥75% of the inverse difference (that is, a high probability of a decrease in biomass with increasing depth) are underlined.

productivity, generally with nutrient limitations in the shallows[32] but greater productivity with depth, which at broader scales may cause the more consistent zonation pattern of this group[25,50]. Planktivore and piscivore biomass increased across 0–30 m depth, probably reflecting the increased proximity to pelagic energetic subsidies[25,26,32] delivered by upwelling that support the growth of planktonic prey for planktivorous fishes[51,52]. These planktivorous fishes are in turn prey for higher trophic level piscivores[28], such that the distribution of piscivores can be indirectly governed by the availability of pelagic energetic subsidies[29,44,53]. Notably, we observed an approximate twofold increase in the biomass of planktivores and piscivores between 0 and 10 m depth. If the biomass of these groups tracks the availability of their dietary targets, then this trend might be driven by the limited subsidies reaching the shallows. The upslope delivery of pelagic subsidies from deeper waters can be highly variable and upwelled waters are often depth-restricted to below 10 m depth[25]. This can be due to friction caused by the reef topography slowing the propagation of these nutrient-rich waters up the reef slope[54] and limiting their positive benefits on the concentration of zooplankton to greater depths[25]. In contrast, primary consumer biomass decreased with increasing depth, probably limited by the rapid

attenuation of light available for photosynthetic algal production with depth[25]. Across latitudinal gradients, primary consumer biomass is higher in areas of greater irradiance[29]. Their distribution across depth is therefore also probably driven by the enhanced benthic primary production that occurs in shallower well-lit waters.

Reef fish biomass also exhibited a striking and varied relationship with bathymetric steepness, primarily in planktivores and primary consumers. For this study, we measured average steepness at the site scale using a 400 m radial buffer. The correlation between steepness at this scale and fish biomass may reflect the role of localized hydrodynamics[34] and physical hydrodynamic interactions with the benthos[54] in determining the delivery of nutrient-rich subthermocline water up onto shallow reefs[26]. While the presence of a depth gradient is a fundamental physical feature of every tropical coral reef island and atoll in the world, the steepness of this gradient can vary. As such, the two can combine to determine the influx of pelagic subsidies to otherwise nutrient-poor tropical waters[31] and probably set natural limits on the distribution and productivity of reef fishes[55–58]. Previous studies using estimates of reef steepness derived at broader spatial scales (for example, 10 km site buffer radius) have found inconsistent effects of reef steepness on

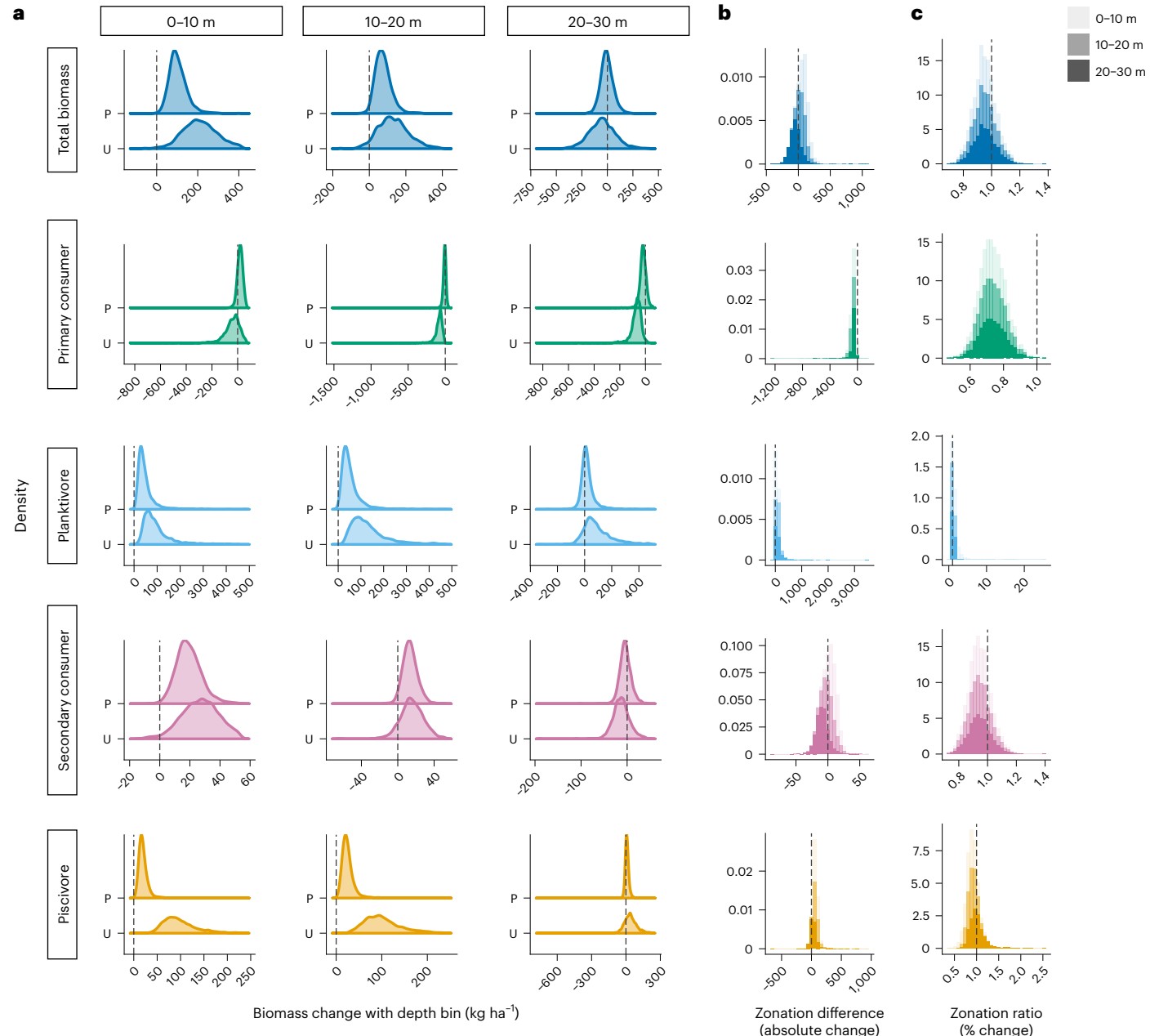

**Fig. 3 | Changes in reef fish biomass across shallow reef depth. a**, Conditional posterior distribution of changes in biomass of each trophic group of fish (rows labelled on left) with each 10 m depth bin (columns labelled at top), for populated (P) and unpopulated (U) islands. **b**, Conditional posterior distributions of differences in zonation, measured as differences in absolute increase or decrease in biomass within each depth bin for populated versus unpopulated islands. **c**, Conditional posterior distributions of zonation ratios, measured as the ratio of percentage change in biomass in each depth bin for populated versus unpopulated islands. For example, **a** and **b** show a greater increase in absolute biomass of piscivores across depth bins for unpopulated islands than for populated islands (further right of the dotted line) but **c** shows that the zonation ratio of percentage change in biomass is greater for populated islands in 0–20 m, spanning two bins (left of dotted line). All plots display change in biomass with depth standardized at the study mean value of bathymetric steepness.

fish productivity[52], possibly highlighting critical scale-dependency in localized upwelling processes created by physical features like internal waves[36]. Planktivores and primary consumers are strongly influenced by energetic subsidies to coral reefs[29,44,59] and their biomass is naturally higher in areas of higher oceanic primary production[45,52]. Our results show that planktivore biomass increased between 0° and 30° with the sharpest increase observed within the initial 0° to 5–10°. Indeed, an increase in reef steepness from just 0° to 5–10° yielded twice the biomass of planktivores and a 25% increase in the biomass of primary consumers. Notably, a threshold of approximately 0–10° steepness is

required for enhanced nearshore primary production around these islands and atolls[31]. At steepness levels of 30–44°, planktivore biomass plateaued and primary consumer biomass declined. This may indicate a threshold of critical slope steepness where internal waves rich in deep-water pelagic nutrients become more likely to be reflected back offshore than to propagate upslope and/or break at steeper topography[35,60,61]. We thereby provide ocean-basin scale evidence quantifying the influence of local-scale reef steepness on patterns of reef fish biomass. Combined, the results suggest the existence of lower and upper local-scale thresholds in critical reef steepness in mediating

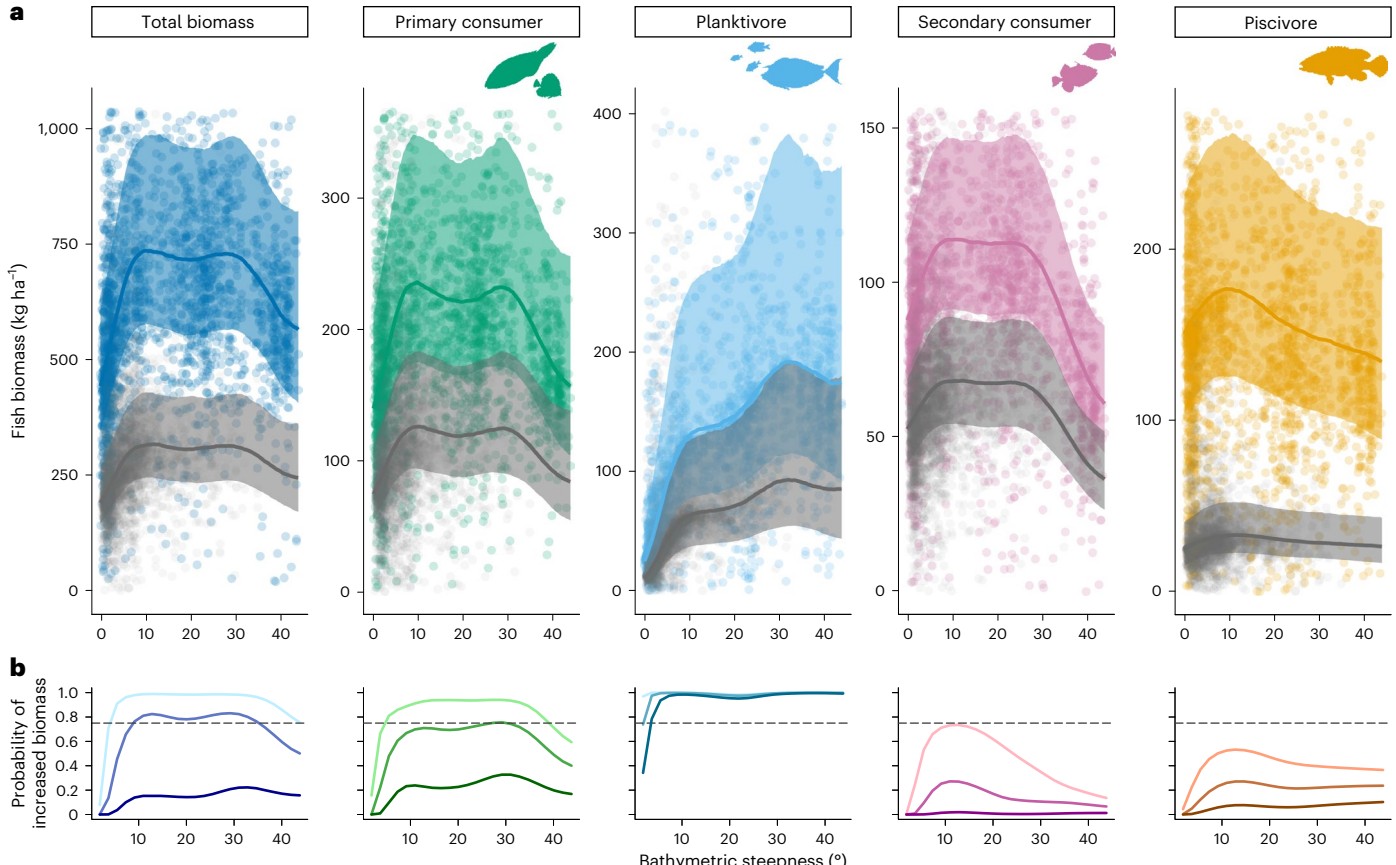

**Fig. 4 | Coral reef fish biomass across a gradient of reef bathymetric steepness at unpopulated (colour) and populated (grey) islands. a**, Estimates represent marginal (integrated over depths from 0 to 30 m) posterior medians (lines), 75% percentiles (shaded areas) and partial residuals (points) at the study mean value of depth. The *y* axis is limited to the maximum value of the 75% CI so partial residuals exceeding axis limits are not displayed). Total *n* = 5,525 SPC surveys (across 2,253 forereef sites, 35 islands and 5 ecoregions). **b**, Probability of increased fish biomass with increasing bathymetric steepness from 0° to 44°. Coloured lines show the marginal posterior distribution derived probabilities of proportionate increases in reef fish biomass with increasing bathymetric steepness (°) among trophic groups. Shading of coloured lines represents probabilities of biomass increase by 25% (light), 50% (medium) and 100% (dark). Grey dotted line highlights probability threshold of 0.75.

delivery of allochthonous subsidies into the shallows and that these effects propagate through to determine the natural carrying capacity of specific trophic groups of reef fishes. Previous studies document variable peaks in planktivorous and piscivorous fishes at mesophotic depths beyond the 30 m limit of this study[49,62]. These variable peaks may be indicative of spatial variation in upwelling, potentially linked to—among other oceanographic factors and associated changes in benthic composition[33]—differences in local bathymetric steepness among those study locations.

Despite marked bathymetric gradients in fish biomass of unpopulated islands, we show that depth-related changes in biomass were altered by depleted biomass baselines at islands inhabited by people. There was overall lower fish biomass across the depth gradient for all trophic groups of populated locations. Further, the change in absolute biomass of planktivores, piscivores and secondary consumers across depth was substantially reduced for populated islands and depth zonation in primary consumers was lost. Conversely, when measured as percentage change in biomass, depth zonation was greater on populated islands for secondary consumers and especially piscivores. However, for populated islands overall lower biomass baselines of all groups and frequent absence of piscivores in shallow-water surveys (that is, zero-count survey observations) probably served to artificially inflate proportionate change across depth. These findings add to mounting global evidence of humans changing fundamental ecological

organization on tropical reefs[14,17]. Human-driven declines in reef fish biomass even at relatively low levels of human exploitation are well documented in the Pacific[44,63]. Fishing reduces the overall standing biomass of reef fishes across trophic groups[44], often with marked losses of piscivores and herbivores[42,64–66]. Our findings of diminished biomass of primary consumers from shallow depths and piscivores and planktivores between 0 and 30 m on populated reefs does not exclude the possibility of mesophotic refugia for depth generalists[33,42,49,67]. It does, however, underscore the vulnerability of herbivorous fishes that are largely restricted to shallow reef zones[68]. We note that human impacts on reef fish assemblages are not limited to the effects of fishing[14,63]. Global warming interacts with local threats such as land-use related sedimentation and nutrient loading into watersheds, dredging, plastic pollution and invasive species[14,69]. As a result of these multiple stressors, underlying relationships between reef organisms and their surrounding environmental settings have been blurred across the region[12,39]. Our findings show that by using human population presence/absence as a simple binary predictor of these impacts, natural zonation signals of absolute fish biomass change across depth are substantially reduced or are lost at populated islands, with variable responses among trophic groups. Such evidence emphasizes the critical need for greater protection for reef ecosystems from a suite of human impacts[14,38] and in particular for depth-constrained trophic groups that perform distinct and important functions.

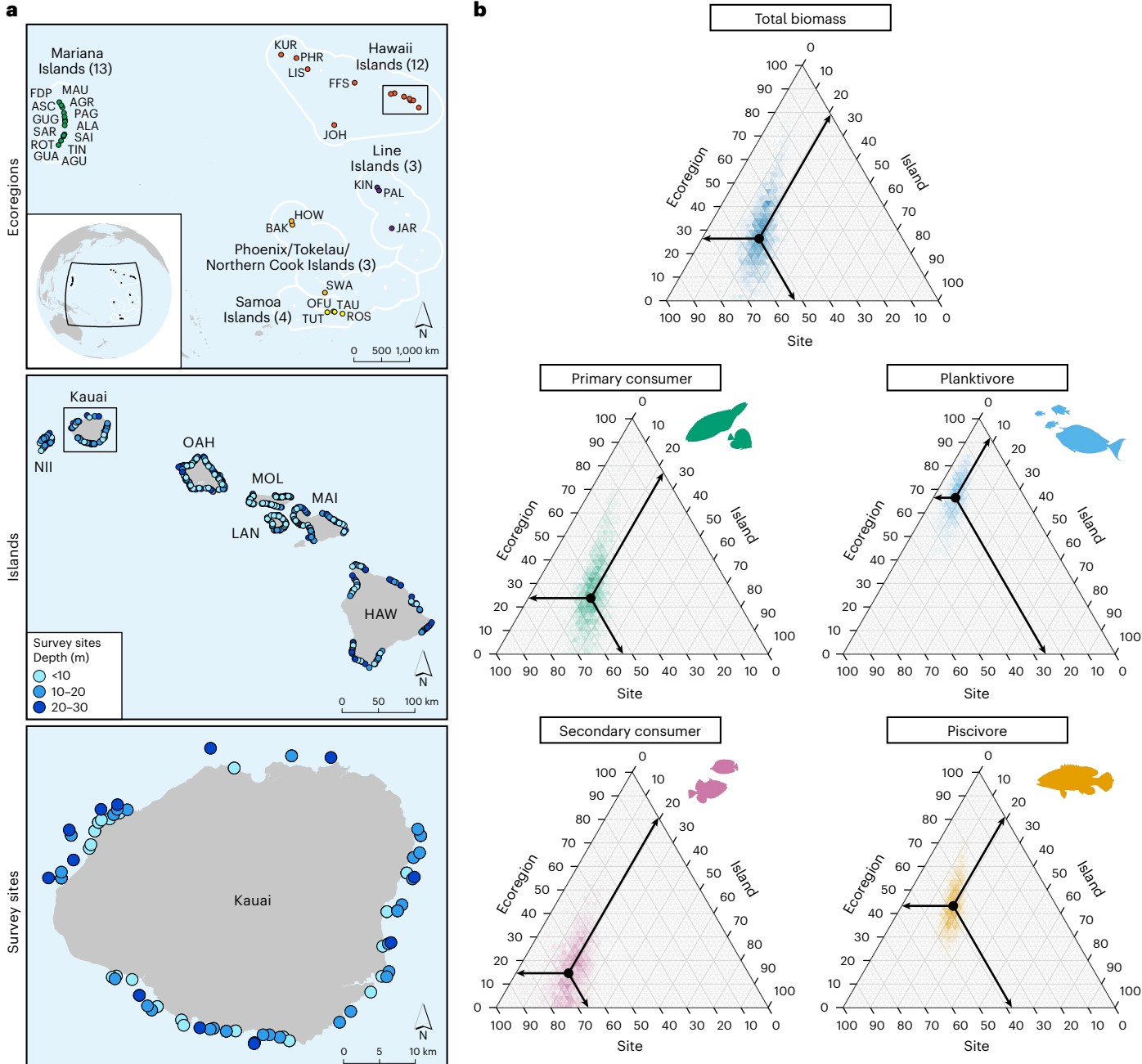

**Fig. 5 | The proportion of residual variation in coral reef fish biomass explained by the hierarchical structure of site, island and ecoregion spatial scales across the central and western Pacific. a**, Maps illustrate the spatial scales (from top to bottom): ecoregions (full island names and associated abbreviations in Supplementary Table 1), example of islands within ecoregions (main Hawaiian Islands shown) and example of sites within islands (Kauai shown). **b**, Ternary plots of the relative posterior standard deviations explained by the spatial scales for total biomass and each trophic group. Black arrows indicate geometric mean percentage of standard deviations at each nested spatial scale (median s.d. estimates and CIs in Supplementary Table 10).

Natural resource management is most effective when applied at scales aligning with (or broader than) scales of ecological variance[70]. This is because ecological communities exist in relation to their environment at spatial scales where structuring biophysical processes dominate to limit or promote the abundance of competitive organisms[1,2,50]. For example, intra-island gradients in surface wave energy and localized upwelling can determine the abundance and spatial clustering of benthic organisms on coral reefs[36,71]. For reef fish assemblages, inter-island variability in oceanic productivity and island geomorphology can mediate levels of species richness and functional redundancy[5].

Conversely, finer-scale habitat quality (that is, complexity and cover) can be more influential in determining other aspects of fish assemblage diversity and the abundance of particular groups and species[5,6]. As a result, variation in these biophysical processes through space can create inherent ecological heterogeneity across those spatial scales[4,70].

While there was minimal observed deviation from the global depth effect across the study islands and ecoregions, our results show that spatial variation in fish biomass—across site, island and ecoregion scales—was differentially and unevenly distributed among trophic groups, underscoring the importance of scale of observation in

ecological enquiry[1,72]. The 35 islands and atolls in this study span much of the western and central Pacific and encompass substantial biophysical gradients that influence ecological organization—ranging from local variation in live coral habitat availability among reefs, for example, to regional disparities in temperature, irradiance and primary production[3,12,29,44]. These scale-dependent gradients can influence the biomass of herbivorous, planktivorous and piscivorous reef fishes observed across the region[44,45,52]. We found that the greatest spatial variance was at the site scale for primary and secondary consumers, piscivores and total biomass. We note that unmeasured temporal stochasticity at the site-level due to factors such as fish recruitment, mobility or behaviour can influence small-scale single time-point observations and their associated variability at that scale[6]. Nonetheless, the importance of site-scale characteristics, indicated by this intra-island heterogeneity, supports numerous studies that identify habitat availability[73], local hydrodynamics[5] and local disturbances[14,63,74] as predominant mediators of the biomass of those groups[6]. Spatial variance at the site-level was particularly high (63%) for secondary consumers, emphasizing trends in location-specific variability in their biomass–depth relationships described in previous studies[33,49]. Conversely, spatial variance in planktivore biomass was greatest at the ecoregion scale, pointing towards regional disparities in primary production[3] and the availability of pelagic subsidies as a primary correlate in the distribution of planktivorous fish biomass[30,52]. These findings align with previous studies that describe habitat composition at the site-level to be the likely dominant driver of reef fish metacommunity structure, including diversity and the biomass of most trophic groups, while attributing greater prevalence of planktivores to larger-scale gradients in overall ocean productivity[6]. That we observed lower variation at the island scale than site and ecoregion scales may be due, in part, to a dominant influence of local variation in habitat, hydrodynamics or disturbances and variable background levels of productivity across ecoregions, over processes occurring at the island scale. In this context, our findings suggest that management of primary consumers, piscivores and especially secondary consumers might achieve satisfactory outcomes at local within-island scales with no-take areas[75] (assuming a source of larval supply), habitat restoration or better regulated destructive human activities[74]. Local management of planktivores is no doubt also important[30] but, given the potential influence of regional-scale drivers on planktivore biomass production and the importance of this group as the prey base for higher trophic levels[28], more nuanced, region-specific targets for recovery[76] or catch of planktivores may be advisable in areas of naturally lower primary production.

In revisiting one of the oldest ecological theories—energetic resource-driven depth zonation of tropical coral reef communities, to date untested at scale[20–23]—we provide evidence of generalizable depth zonation spanning islands across the Pacific. While the principle of resource-driven depth zonation held at both unpopulated and populated islands for some trophic groups (for example, direction of change for planktivores, piscivores and secondary consumers), their magnitude of change with depth (that is, absolute biomass) was substantially reduced for populated islands. For functionally important primary consumers, the depth zonation signal was conspicuously absent for populated islands. Therefore, while we broadly evidence sustained depth zonation in these contemporary reef systems, our findings expose limitations of the energetic resource-driven zonation paradigm for predicting ecological dynamics where human impacts increasingly confound connections between ecological communities and their surrounding environment[10,12,39].

## Methods
### Study location
To examine the fish zonation across depths and investigate how humans may impact natural zonation on coral reefs, we used monitoring data from a standardized dataset that spans the central and western Pacific[43]. Specifically, 5,525 distinct surveys from 2,253 forereef sites (≤30 m depth) conducted on 35 US and US-affiliated islands and atolls across 42° of latitude (14° S to 28° N) and 62° of longitude (178° W to 145° E). The data were collected between 2010 and 2014 for the National Oceanic and Atmospheric Administration (NOAA) Pacific Reef Assessment and Monitoring Program (RAMP; 2010–2012) and NOAA's National Coral Reef Monitoring Program (NCRMP; 2013–2019), conducted by the Ecosystem Sciences Division (ESD) of NOAA's Pacific Islands Fisheries Science Center (PIFSC)[43]. We classified sites around islands and atolls (hereafter, 'islands') as 'populated' or 'unpopulated' on the basis of unpopulated islands having <50 residents and located >100 km from the nearest larger human settlement using the 2010 US census (www.census.gov/2010census) (sensu refs. 12,63). Of the 35 study islands, 21 were classified as unpopulated (n = 2,321 surveys, across 923 sites) and 14 as populated (n = 3,204 surveys, across 1,330 sites) (Supplementary Table 1). Islands were also classified by their location within ecoregions: Hawaii Islands, Line Islands, Mariana Islands, Phoenix, Tokelau, Northern Cook Islands and Samoa Islands[47]. The location of each replicate site was preselected by randomized stratified design per sampling unit of the Pacific RAMP/NCRMP protocol (island, group of small islands or subsections of larger islands). The target sampling domain was hard-bottom substrate, with sampling effort stratified by reef zone and depth (0–6, 6–18 and 18–30 m)[43]. We constrained the dataset to forereef habitat only to remove any possible confounding effects of habitat type on reef fish assemblages. Reef depth (m) was recorded by divers in situ at survey sites. An online map viewer of the study sites is available: http://noaa.maps.arcgis.com/apps/webappviewer/index.html?id=da5c18ea60d049588fca5feecb82fe07.

### Reef fish survey data
The abundance and body-size of all diurnal, non-cryptic reef fishes were estimated using stationary point count (SPC) surveys (sensu refs. 5,29,43,44,77,78). At each site, divers conducted simultaneous visual fish counts within one to four adjacent, visually estimated 15 m diameter cylindrical plots, extending from the substrate to the limit of vertical visibility[43]. First, divers compiled lists of all species observed within the survey area over a 5 min period, then counted and estimated the size (total length, TL, to the nearest cm) of listed species present within the cylinder over ~30 min. Surveys were conducted by multiple observers across the study ecoregions and years. NOAA uses extensive training and technical validation protocols to ensure consistency and avoid bias in survey technique, fish species identification and size estimation[43]. Full details on SPC survey methods and technical validation steps are available in ref. 43. To further mitigate any confounding effect of observer bias among fish surveys, we included 'diver identity' as a random intercept in all statistical models (described below) (sensu ref. 48). We calculated individual species biomass from the SPC counts using the allometric conversion as $W = aL^b$, where $W$ is the biomass for individual fishes, $L$ is the estimated body length of each individual and parameters $a$ and $b$ are constants for each species (from ref. 79). Fish biomass (g m$^{-2}$) was pooled into total fish biomass and four trophic groups: 'primary consumers' (herbivores and detritivores), 'planktivores', 'secondary consumers' (omnivores and benthic invertivores) and 'piscivores'[46]. Taxa that are not typically reef-associated were excluded from the analyses, including tuna, bonito and milkfish (families Chanidae, Myliobatidae and Scombridae; Supplementary Table 12). Sixteen species of shark, jack and barracuda (families Carcharhinidae, Carangidae and Sphyrnidae) were also excluded from the analyses as these highly mobile, large-bodied, roving piscivores are known to be affected by the presence of stationary divers, typically resulting in systematic overinflation of visual survey density estimates[80] (sensu refs. 44,48; Supplementary Table 12). Further, the presence of divers among study locations also probably introduces a potential source of differential bias of biomass estimates of these fishes, with 'mobbing' behaviour by jacks, sharks and barracuda known to occur

particularly in remote, unpopulated areas such as the northwestern Hawaiian Islands[44,45]. Zonation patterns in piscivore biomass were comparable with and without this filtering approach. This suggests that the reported patterns were not an artefact of the data handling choice to exclude some species known to be affected and systematically overestimated by divers (Extended Data Fig. 4). However, model outputs of population-level effects of depth and bathymetric steepness showed much greater biomass estimates at unpopulated than populated islands, indicating that conservative exclusion of these species mitigated potential systematic bias associated with the survey method among locations (Extended Data Fig. 4).

## Bathymetric reef steepness

We derived site-level estimates of bathymetric steepness (°) from depth mosaics created from multibeam SONAR, bathymetric LiDAR and imagery derived depths in ArcGIS Pro v.2.7 using the 'Slope' tool (Spatial Analyst) (Supplementary Table 2). Resulting mosaics ranged in spatial resolution from 5 to 50 m. Steepness estimates were calculated by averaging steepness values within a 400 m radial buffer around each replicate SPC site and manually excluding backreef and lagoon areas and landmass elevation using NOAA PIFSC ESD habitat map information. All analyses were done in the appropriate Universal Transverse Mercator zone for each island. A radial buffer size of 400 m was selected to encompass depths that would capture the propensity for pulsed delivery of nutrient-rich subthermocline water by upwelling[81,82] and for this upwelling to propagate onto shallow reef habitats at depths ≤30 m (ref. 31) (maximum depth range within slope buffers: 596.2 m). Buffer-size extent was also selected to cover a reliable gradient in steepness while avoiding buffer sizes that would easily extend across small island-masses to include bathymetry on opposite sides of the island. All sites were visually cross-checked for island overlap and those including >5% radial-buffer bathymetry on the opposite site of a landmass were excluded from all analyses.

## Statistical analysis

To determine cross-spatial-scale depth zonation patterns in reef fish biomass, we fitted multilevel Bayesian regression models with brms[83]. Fish biomass (biomass density; g m$^{-2}$) was modelled separately for each trophic group and total fish biomass, using the following general model:

$$y_i \sim \gamma(\mu_j, \varsigma)$$

$$\log(\mu_{ji}) = \alpha + \boldsymbol{\beta} X_i + Z_i \gamma$$

where $\alpha$ is the trophic group (or total biomass) specific intercept, $\boldsymbol{\beta}$ is a vector of population-level regression coefficients relating covariates $X_i$ for observation $i$ to the log of the expected biomass density $\mu_j$. Group-level coefficients $\gamma$ are estimated for random effects encoded in design matrix $Z$.

We used depth and bathymetric steepness here as proxies for biophysical processes which influence coral reef fish assemblage structure. Population-level effects therefore included forereef depth (1.3–30.0 m), bathymetric steepness (0.01–43.78°; mean 10.53°) and the interactions of depth with each of bathymetric steepness and human population status. Near-island primary production can decrease exponentially with increasing island bathymetric steepness[31]. Therefore, potential nonlinear effects of bathymetric predictors on fish biomass was included in the interaction of steepness with depth by fitting it as a random effect with a cubic-basis spline[30].

To account for confounding effects of local human impacts on biomass density, we also included a population-level effect for human population status (populated or unpopulated). Temporal variability in reef fish survey estimates can be introduced by observers and can also reflect dynamic processes such as those determining interannual

variation in larval recruitment or nutrient availability across the region[84,85]. We therefore included group-level effects for observation year (5 years; 2010–2014) and year nested within both ecoregion and island in our models to account for this potential variation and avoid potential sampling bias. A group-level random intercept for 'diver identity' was included to account for the potential effect of individual observer bias. By assuming an inherent non-independence within divers and their observations that might affect the estimated means and associated errors of fish biomass (sensu ref. 48), we were then able to estimate isolated population-level effects (depth, human population status and bathymetric steepness) (sensu ref. 48). More broadly, by controlling these potential sources of variability, we can more accurately test a priori proposals about ecological zonation occurring across spatial scales and with greater inferential strength[6].

To understand whether the ecological organization of coral fish assemblages in relation to these biophysical processes holds true across varying spatial scales, we first accounted for the inherent hierarchical structure in the data by including random intercepts for ecoregions, islands within ecoregions and sites within islands (sensu ref. 4). We suggested that patterns of fish biomass across bathymetric gradients may track scale-dependent biophysical drivers that regulate energetic resource supply to shallow coral reefs[25,36,86]. For example, regional-scale oceanographic currents and sea surface temperatures drive regional differences in primary production and net resource availability[3,86]. These net gradients in availability can be modified at smaller spatial scales by oceanographic features interacting with local bathymetry[31] such that, depending on the prevailing direction of internal tidal energy, upwelling processes can drive strong intra-island gradients in nutrient and planktonic resource supply among sites[25]. As potential indicators of these scale-dependent processes, we then measured the variation in reef fish biomass at these three hierarchical scales (sites within islands within ecoregions) and compared the proportion of the total variation explained by those scales (sensu ref. 4). We quantified variation by extracting posterior standard deviations of random effects at these distinct geographical scales in the fish biomass models and compared them across trophic groups. We further included a random slope term for depth-within-island to account for potentially variable depth zonation of biomass across islands.

All models were fitted with a Gamma response distribution, using a log-link function as biomass was positive, continuous and overdispersed[87]. Fish of each trophic group were not observed in every SPC survey. To account for these zero-count observations, hurdle models were used, first fitting the presence–absence of fish biomass as a function of the predictors described above, with a binomial distribution and logit function and then fitting the non-zero biomass data with the Gamma multilevel model outlined above. Where the proportion of zeros was too low to effectively estimate effect sizes in the presence–absence component (that is, an insufficient contrast between the number of zeros and ones), the use of a hurdle structure affected model convergence and only added noise. This occurred for primary consumers (1.09% zeros) and secondary consumers (0.05% zeros), so for these groups the zero biomass replicates were removed from the analysis and the Gamma model detailed above was fitted.

This study builds on existing knowledge established in previous research that estimated a global baseline of total resident reef fish biomass in the absence of fishing[48]. We integrate this prior information by using their published posterior biomass estimate (1,013 kg ha$^{-1}$) as the mean of the prior for log of total biomass ($\alpha$; converted to g m$^{-2}$) (with standard deviation set at 1):

$$\alpha \approx N(\log(101.3), 1)$$

The intercept prior for each trophic group was estimated as a proportion of this total unfished global biomass estimate as approximated in ref. 48. The grouping of secondary consumers as defined in

this study (a coarse group based on diverse diet items typically targeted by species including invertivores, corallivores and omnivores[46]) differed from those used in ref. 48. Therefore, the intercept prior for this group was determined by the proportion of secondary consumers in the total biomass from the present study data, applied to the unfished biomass estimate in ref. 48. Our study and ref. 48 use comparable data (in situ counts of diurnally active, non-cryptic reef fish on forereef slopes, excluding sharks and semipelagics such as jacks). However, to account for potential differentiating factors between the studies, such as species filtering approaches, census method or geographical representativeness, we inflated the prior standard deviation in the intercepts for our models by an order of magnitude. Model priors are detailed in Supplementary Table 3 and plotted with unpopulated posterior intercept estimations in Extended Data Fig. 5. Marginal posterior distributions for model parameters were estimated by Hamiltonian Monte Carlo sampling, using 10,000 iterations across four chains, with a warm-up of 2,000 iterations and a thinning factor of four. To ensure unbiased parameter estimates (that is, absence of divergent transitions), we set adapt delta to 0.995 and a maximum tree-depth of 12. Model fits and convergence were assessed with graphical posterior predictive checks and via trace and effective sample size plots, the Gelman–Rubin R-hat diagnostic and Bayesian adaptation of $R^2$ (ref. 88). An effective sample size of >1,000 was chosen to determine stable parameter estimates[89]. Medians of posterior distributions were calculated to obtain a single-point estimate and 75% and 95% credible intervals (CIs) were calculated from the respective quantiles of the posterior distributions of all metrics presented. Non-independence of population-level predictors was assessed by plotting bivariate correlations between the posterior samples (MCMC draws) of predictor coefficients and quantifying Pearson correlation coefficients between paired samples (Supplementary Fig. 1)[88]. Correlation coefficients were all <5%, bar one: a single pairwise correlation coefficient for hurdle components depth and steepness in the planktivore model which was still relatively low at 28%.

All analyses were conducted in R v.4.2.1 (ref. 90). Bayesian hierarchical models were implemented in cmdstanr using brms v.2.17.0 (ref. 89); probability of covariate effect direction was estimated with bayestestR v.0.10.0 (ref. 91); model information for querying posterior predictions was extracted with tidybayes v.3.0.2 (ref. 92); cross-spatial model variance was plotted with TernaryPlot in Ternary v.1.2.3 (ref. 93); model fits assessed using r2_bayes in performance v.0.9.2 (ref. 94) and independence of model predictors assessed with ggpairs in GGally v.2.1.2 (ref. 95). Fish symbols used in figures were created with fishualize 0.2.0 (ref. 96).

### Reporting summary

Further information on research design is available in the Nature Portfolio Reporting Summary linked to this article.

## Data availability

All data and R code used in this study are available at an open-source repository (https://github.com/LauraERichardson/Depth-Fish).

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

## Acknowledgements

We thank I. Williams and the staff at the Pacific Island Fisheries Science Center (NOAA) for extensive survey data collection. NOAA's Coral Reef Conservation Program (CRCP) supported and funded the National Coral Reef Monitoring Program (NCRMP); CRCP project no. 743. This research was funded by the European Commission supported by a Marie Skłodowska-Curie Sêr Cymru II COFUND Fellowship (no. BU191) and a Marie Skłodowska-Curie European Fellowship (no. 844213—FISHSCALE) awarded to L.E.R. We thank S. Bond and A. Lawrence for useful discussions, A. Feather and S. Sannassy Pilly for technical code assistance, A. Merritt for contributing to figure preparation, J. Hiddink and an anonymous internal NOAA reviewer for critical feedback that contributed to preparation of the manuscript.

## Author contributions

L.E.R., A.H. and G.J.W. conceived this study. L.E.R., A.H., G.J.W., J.M.G., J.L. and J.A.M.G. designed the methodology. A.H. contributed to survey data collection. T.L.K. provided NOAA's fish survey data. J.L. and J.M.G. computed satellite-derived bathymetric slope steepness estimates. L.E.R. and P.N. conducted the analyses with input from A.J.D. L.E.R. led manuscript writing with input from A.H., G.J.W., J.L., P.N., A.J.D., T.L.K. and K.E.I. All authors contributed substantially to the drafts and approved the final version for publication.

## Competing interests

The authors declare no competing interests.

## Additional information

**Extended data** is available for this paper at https://doi.org/10.1038/s41559-023-02201-x.

**Correspondence and requests for materials** should be addressed to Laura E. Richardson.

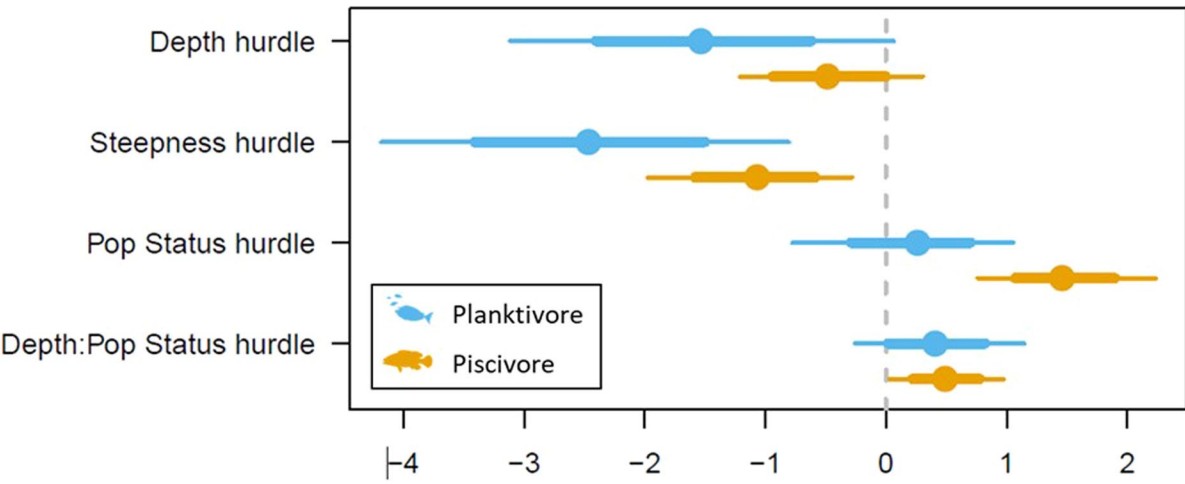

Scaled fixed effects estimates

**Extended Data Fig. 1 | Hurdle model component (presence–absence) effect estimates of shallow reef depth, bathymetric steepness and human population status of islands (*Pop Status* indicates the effect of 'populated' by humans versus 'unpopulated') on planktivore and piscivore reef fish biomass.** Effect sizes are scaled and include interactions of depth with population status (*Depth:Pop Status*) on reef fish biomass. Points represent posterior median estimates from Bayesian hierarchical models testing for an effect of each explanatory variable on reef fish biomass, with 75% (thick lines) and 95% (thin lines) percentiles. Explanatory variables were mean-centred and scaled by one standard deviation to facilitate comparisons of effect sizes among them. $N = 5,525$ stationary point count (SPC) surveys (across 2,253 forereef sites, 35 islands, five ecoregions).

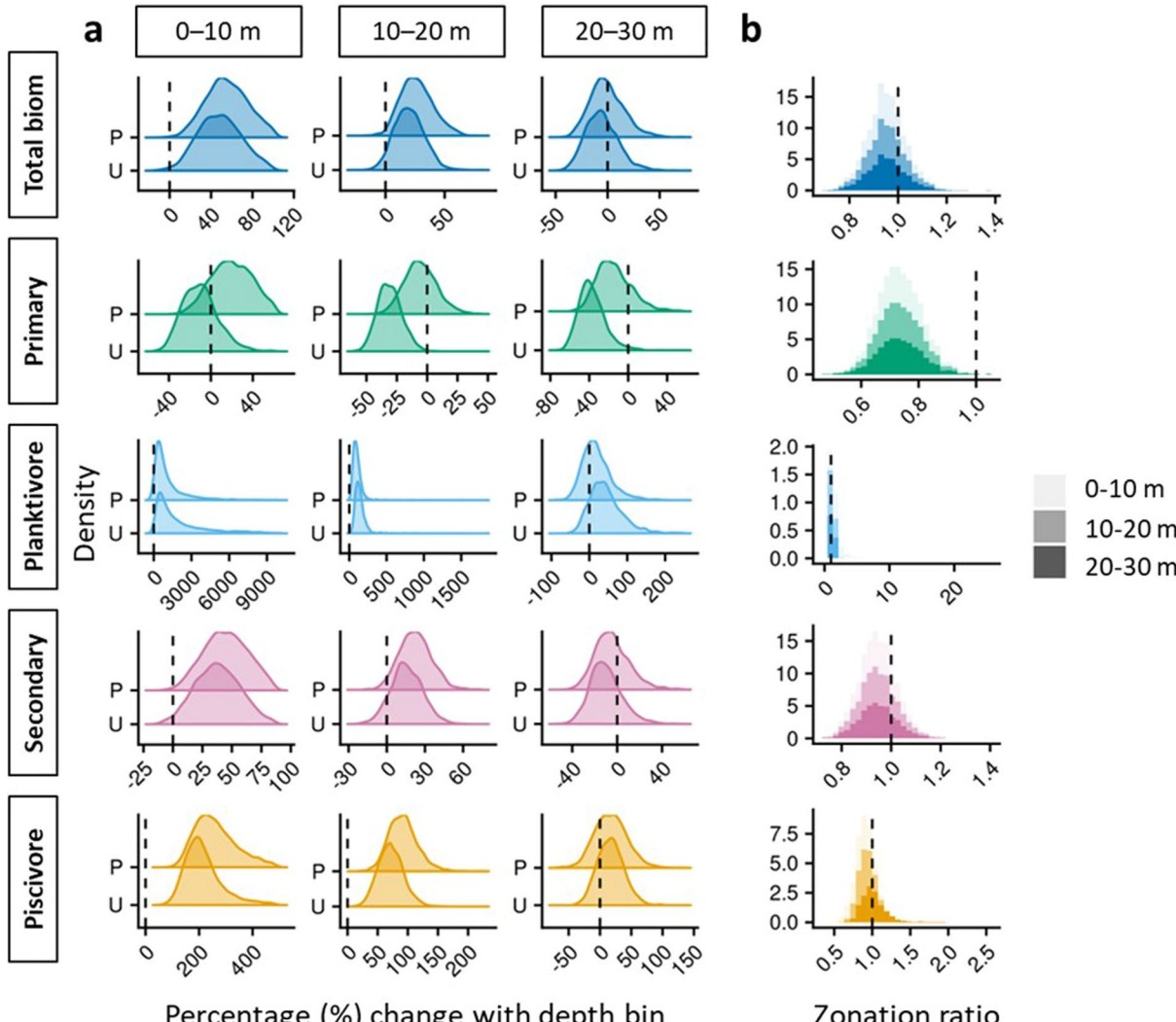

**Extended Data Fig. 2 | Percentage change in reef fish biomass across shallow reef depth.** Density distributions of posterior predicted percentage change in biomass of each trophic group (rows labelled on left) with each 10 m depth bin (columns labelled at top), at populated (P) and unpopulated (U) islands. B) Posterior predicted distributions of zonation ratios of populated versus unpopulated islands in each 10 m depth bin.

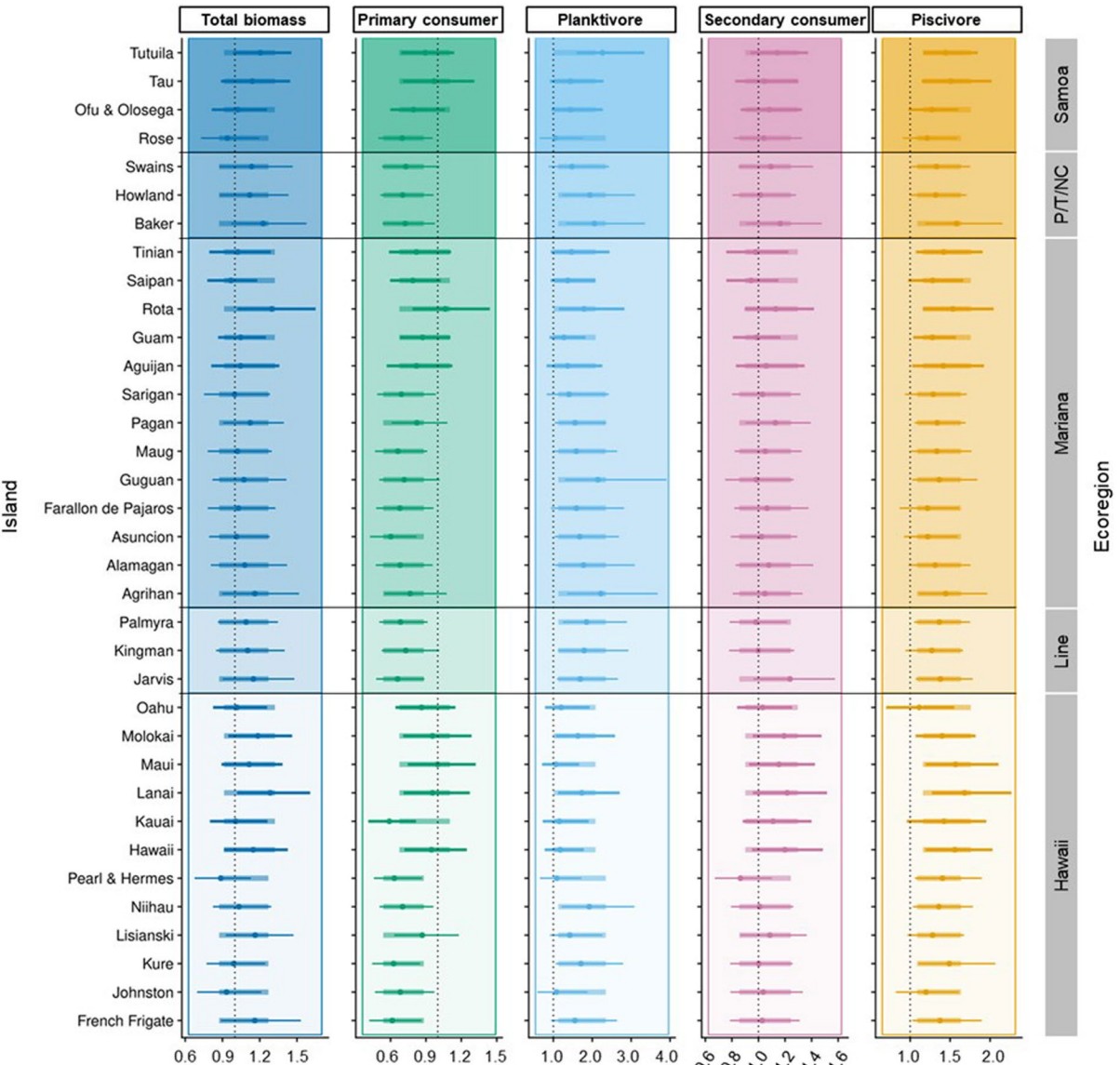

**Extended Data Fig. 3 | Island-level depth effects on reef fish biomass among distinct study ecoregions (Samoa Islands; Phoenix/Tokelau/Northern Cook islands (P/T/NC); Mariana Islands; Line Islands; Hawaii Islands).** Point estimates and associated 95% credible interval (CI) lines indicate the effect of increasing depth on fish biomass (proportional change) at each island (increase if >1, decrease with depth if <1). The population status of islands is indicated by CI line thickness (thick CI, populated; thin CI, unpopulated). Pale shaded boxes underlying point estimates and CIs represent the average proportionate global effect of depth on biomass given the human population status and average bathymetric steepness and are provided to aid visual assessment of spatial consistency in depth effects across the study. For example, at islands where depth effects on fish biomass follow the overall global depth trend for a given trophic group, point estimates overlap the pale shaded boxes. Conversely, at islands where depth effects differ from the global effects, point estimates are positioned outside of the shaded boxes (for example, in Kauai, there is a greater observed decrease in the biomass density of primary consumers with increasing depth than the over-all effect of depth for that group at populated islands with similar bathymetric slope steepness; similarly, there is a greater proportionate increase in biomass of planktivores at Tutuila with increasing depth relative to the global depth effect for that group at populated islands with comparable reef steepness). $N = 5,525$ stationary point count (SPC) surveys (across 2,253 forereef sites, 35 islands, five ecoregions).

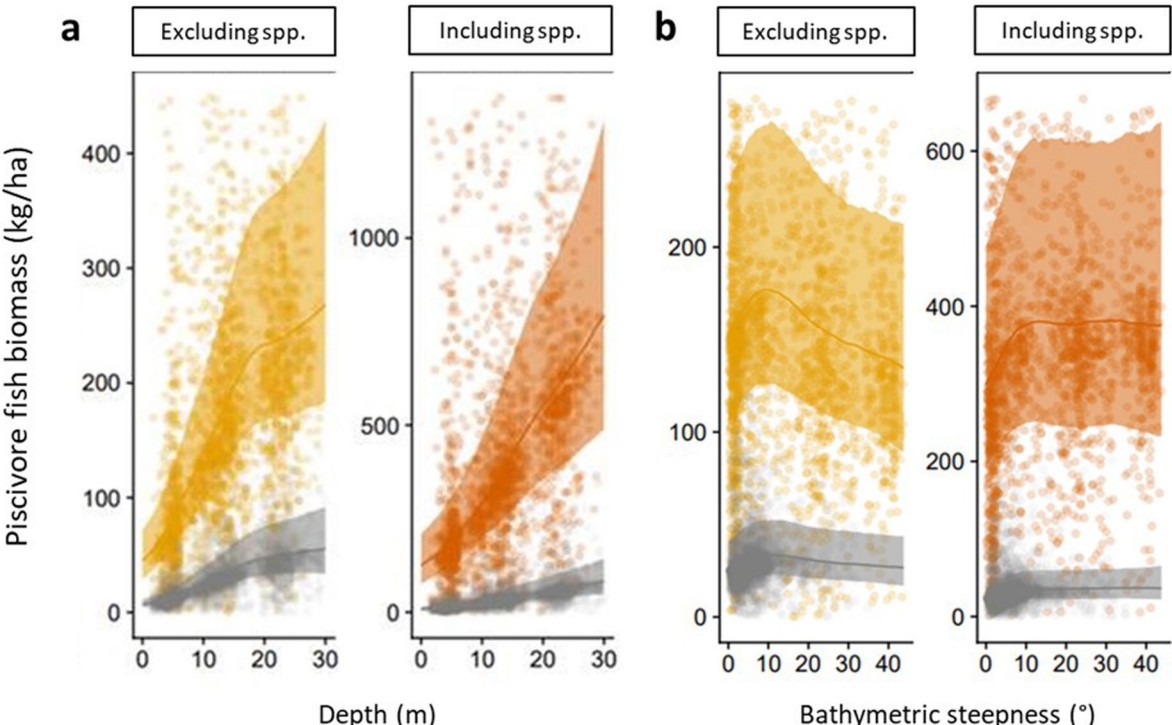

**Extended Data Fig. 4 | Piscivore fish biomass, excluding and including the biomass of *Carcharhinidae*, *Carangidae* and *Sphyrnidae* (see Supplementary Table 12), across gradients of depth (a) and bathymetric steepness (b) at unpopulated (colour) and populated (grey) islands.** Estimates represent conditional posterior medians (lines), 75% percentiles (shaded areas) and partial residuals (points) at the study mean values of bathymetric steepness (panel a) and depth (panel b). The y axis is limited to 1.05x the maximum value of the 75% CI so partial residuals exceeding axis limits are not displayed. *N* = 5,525 stationary point count (SPC) surveys (across 2,253 forereef sites, 35 islands, five ecoregions).

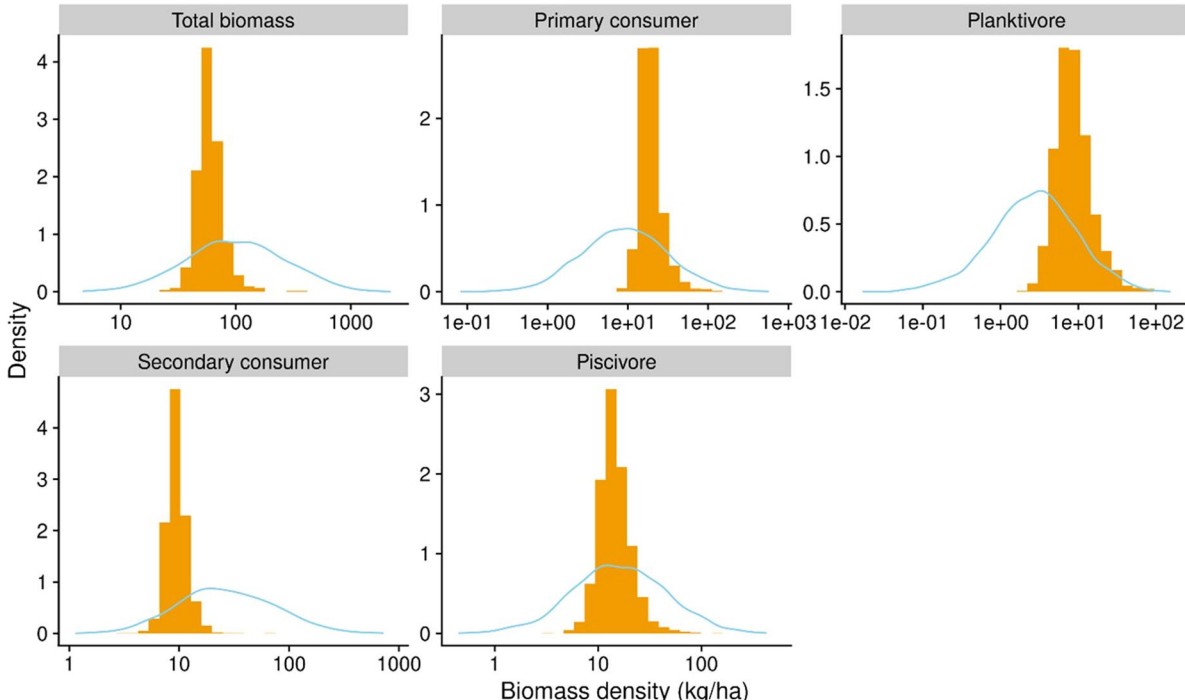

**Extended Data Fig. 5 | Biomass model prior distributions and unpopulated posterior intercept estimations for each trophic group of reef fish.** Specified prior distributions for each biomass group are shown with the blue line. Posterior intercept distributions are shown in orange.

# Reporting Summary

## Statistics

For all statistical analyses, confirm that the following items are present in the figure legend, table legend, main text, or Methods section.

| n/a | Confirmed | |
|---|---|---|
| ☐ | ☒ | The exact sample size (*n*) for each experimental group/condition, given as a discrete number and unit of measurement |
| ☐ | ☒ | A statement on whether measurements were taken from distinct samples or whether the same sample was measured repeatedly |
| ☐ | ☒ | The statistical test(s) used AND whether they are one- or two-sided<br>*Only common tests should be described solely by name; describe more complex techniques in the Methods section.* |
| ☐ | ☒ | A description of all covariates tested |
| ☐ | ☒ | A description of any assumptions or corrections, such as tests of normality and adjustment for multiple comparisons |
| ☐ | ☒ | A full description of the statistical parameters including central tendency (e.g. means) or other basic estimates (e.g. regression coefficient) AND variation (e.g. standard deviation) or associated estimates of uncertainty (e.g. confidence intervals) |
| ☒ | ☐ | For null hypothesis testing, the test statistic (e.g. *F*, *t*, *r*) with confidence intervals, effect sizes, degrees of freedom and *P* value noted<br>*Give P values as exact values whenever suitable.* |
| ☐ | ☒ | For Bayesian analysis, information on the choice of priors and Markov chain Monte Carlo settings |
| ☐ | ☒ | For hierarchical and complex designs, identification of the appropriate level for tests and full reporting of outcomes |
| ☐ | ☒ | Estimates of effect sizes (e.g. Cohen's *d*, Pearson's *r*), indicating how they were calculated |

*Our web collection on statistics for biologists contains articles on many of the points above.*

## Software and code

Policy information about availability of computer code

| Data collection | We derived site-level estimates of bathymetric steepness (°) from depth mosaics created from multibeam SONAR, bathymetric LiDAR, and imagery derived depths in ArcGIS Pro v2.7 using the 'Slope' tool (Spatial Analyst). |
|---|---|
| Data analysis | All analyses were conducted in R 4.2.1. Bayesian hierarchical models were implemented in cmdstanr using brms 2.17.0; probability of covariate effect direction was estimated with bayestestR 0.10.0; model information for querying posterior predictions was extracted with tidybayes 3.0.2; cross-spatial model variance was plotted with TernaryPlot in Ternary 1.2.3; and model fits assessed using r2_bayes in performance 0.9.2. All data and R code used in this study are available at an open-source repository (https://github.com/LauraERichardson/Depth-Fish). |

For manuscripts utilizing custom algorithms or software that are central to the research but not yet described in published literature, software must be made available to editors and reviewers. We strongly encourage code deposition in a community repository (e.g. GitHub). See the Nature Portfolio guidelines for submitting code & software for further information.

## Data

Policy information about availability of data

All manuscripts must include a data availability statement. This statement should provide the following information, where applicable:
- Accession codes, unique identifiers, or web links for publicly available datasets
- A description of any restrictions on data availability
- For clinical datasets or third party data, please ensure that the statement adheres to our policy

> The dataset generated during and/or analysed during the current study are available on Github (https://github.com/LauraERichardson/Depth-Fish).

## Research involving human participants, their data, or biological material

Policy information about studies with human participants or human data. See also policy information about sex, gender (identity/presentation), and sexual orientation and race, ethnicity and racism.

| | |
|---|---|
| Reporting on sex and gender | *Use the terms sex (biological attribute) and gender (shaped by social and cultural circumstances) carefully in order to avoid confusing both terms. Indicate if findings apply to only one sex or gender; describe whether sex and gender were considered in study design; whether sex and/or gender was determined based on self-reporting or assigned and methods used.*<br>*Provide in the source data disaggregated sex and gender data, where this information has been collected, and if consent has been obtained for sharing of individual-level data; provide overall numbers in this Reporting Summary. Please state if this information has not been collected.*<br>*Report sex- and gender-based analyses where performed, justify reasons for lack of sex- and gender-based analysis.* |
| Reporting on race, ethnicity, or other socially relevant groupings | *Please specify the socially constructed or socially relevant categorization variable(s) used in your manuscript and explain why they were used. Please note that such variables should not be used as proxies for other socially constructed/relevant variables (for example, race or ethnicity should not be used as a proxy for socioeconomic status).*<br>*Provide clear definitions of the relevant terms used, how they were provided (by the participants/respondents, the researchers, or third parties), and the method(s) used to classify people into the different categories (e.g. self-report, census or administrative data, social media data, etc.)*<br>*Please provide details about how you controlled for confounding variables in your analyses.* |
| Population characteristics | *Describe the covariate-relevant population characteristics of the human research participants (e.g. age, genotypic information, past and current diagnosis and treatment categories). If you filled out the behavioural & social sciences study design questions and have nothing to add here, write "See above."* |
| Recruitment | *Describe how participants were recruited. Outline any potential self-selection bias or other biases that may be present and how these are likely to impact results.* |
| Ethics oversight | *Identify the organization(s) that approved the study protocol.* |

Note that full information on the approval of the study protocol must also be provided in the manuscript.

# Field-specific reporting

Please select the one below that is the best fit for your research. If you are not sure, read the appropriate sections before making your selection.

☐ Life sciences ☐ Behavioural & social sciences ☒ Ecological, evolutionary & environmental sciences

For a reference copy of the document with all sections, see nature.com/documents/nr-reporting-summary-flat.pdf

# Ecological, evolutionary & environmental sciences study design

All studies must disclose on these points even when the disclosure is negative.

| | |
|---|---|
| Study description | To examine the fish zonation across depths and investigate how humans may impact natural zonation on coral reefs, we used monitoring data from a standardized dataset of underwater visual fish surveys spanning the central and western Pacific (Heenan et al. 2017; https://www.nature.com/articles/sdata2017176). Surveys (n=5525) were carried out at sites around islands and atolls (hereafter 'islands') classified as 'populated' or 'unpopulated' based on unpopulated islands having <50 residents and located >100 km from the nearest larger human settlement using the 2010 US census (www.census.gov/2010census). Of the 35 study islands, 21 were classified as unpopulated (n = 2,321 distinct surveys, across 923 sites) and 14 as populated (n = 3,204 distinct surveys, across 1,330 sites) (Table S1). Islands were also classified by their location within ecoregions: Hawaii Islands; Line Islands; Mariana Islands; Phoenix, Tokelau, Northern Cook Islands; and Samoa Islands. The location of each unique site was pre-selected by randomised stratified design per sampling units of the Pacific RAMP/NCRMP protocol (island, group of small islands, or subsections of larger islands). The target sampling domain was hard-bottom substrate, with sampling effort stratified by reef zone and depth (0–6 m; 6–18 m; 18–30 m). |
| Research sample | Underwater visual surveys were conducted at distinct sites using the stationary point count method, recording the species, abundance, and body-size of coral-reef fish observed within 15-m diameter cylindrical plots. This is the standard monitoring method |

| | |
|---|---|
| | for the National Oceanic and Atmospheric Administration (NOAA) Pacific Reef Assessment and Monitoring Program (RAMP; 2010-2012) and NOAA's National Coral Reef Monitoring Program (NCRMP; 2013-2019). |
| Sampling strategy | The number of ecoregions (n=5), islands and atolls (n = 35), sites (n=2253), SPC surveys (n=5525) was chosen to maximize the study sample size based on the availability of standardized monitoring data. |
| Data collection | he abundance and body-size of all diurnal, non-cryptic reef fishes were estimated using stationary point count (SPC) surveys. At each site, divers conducted simultaneous visual fish counts within 1–4 adjacent, visually-estimated 15-m diameter cylindrical plots, extending from the substrate to the limit of vertical visibility. First, divers compiled lists of all species observed within the survey area over a 5-min period, then counted and estimated the size (total length, TL, to the nearest cm) of listed species present within the cylinder over approximately 30-mins. Full details on SPC survey methods are available in the published data source by Heenan et al. 2017 [https://www.nature.com/articles/sdata2017176]. |
| Timing and spatial scale | Data were collected using underwater visual census stationary point count (SPC) surveys (n=5,525) from 2,253 forereef sites (≤ 30 m depth), conducted at 35 US and US-affiliated islands and atolls in the Pacific Ocean, across 42 degrees (°) of latitude (14°S to 28°N), and 62° of longitude (178° W to 145°E). The data were collected between 2010–2014 for the National Oceanic and Atmospheric Administration (NOAA) Pacific Reef Assessment and Monitoring Program (RAMP; 2010-2012) and NOAA's National Coral Reef Monitoring Program (NCRMP; 2013-2019), conducted by the Ecosystem Sciences Division (ESD) of NOAA's Pacific Islands Fisheries Science Center (PIFSC). Each site was visited a single time and their location was pre-selected by randomised stratified design per sampling units of the Pacific RAMP/NCRMP protocol (island, group of small islands, or subsections of larger islands) (see Heenan et al. 2017 for details; https://www.nature.com/articles/sdata2017176#Sec2). The timing of surveys were determined by the schedule of research expeditions across the region. |
| Data exclusions | We constrained the dataset to forereef habitat only to remove any possible confounding effects of habitat type on reef fish assemblages. Taxa that are not typically reef-associated were excluded from the analyses, including tuna, bonito, and milkfish (families Chanidae, Myliobatidae, Scombridae; Table S12 in the Supplemental Information). Sixteen species of shark, jack, and barracuda (families Carcharhinidae, Carangidae, Sphyrnidae) were also excluded from the analyses as these highly mobile, large-bodied, roving piscivores are known to be affected by the presence of stationary divers, typically resulting in systematic over-inflation of visual survey density estimates. |
| Reproducibility | Detailed description of the methods are provided, and all code and data necessary to reproduce the findings (including figures and tables) are freely available on GitHub. |
| Randomization | Prior to field data collection, the location of survey sites (latitude and longitude coordinates) was selected from sampling strata via a randomized depth-stratified design, with the goal of surveying reefs as widely as possible around and across islands and atolls, on hard-bottom substrate in water shallower than 30m. |
| Blinding | Blinding was used during data collection by preselecting distinct survey locations via a randomized depth-stratified design (detailed in Heenan et al. 2017; https://www.nature.com/articles/sdata2017176#Sec2). |

Did the study involve field work? ☒ Yes ☐ No

# Field work, collection and transport

| | |
|---|---|
| Field conditions | Surveys were conducted when seas were calm and underwater visibility was clear. Long-term mean SST in the region is approximately 27° C (Gove et al. 2013; https://doi.org/10.1371/journal.pone.0061974). |
| Location | Data were collected from 5,525 surveys on 2,253 forereef sites (≤ 30 m depth) conducted on 35 US and US-affiliated islands and atolls across 42 degrees (°) of latitude (14°S to 28°N), and 62° of longitude (178° W to 145°E). |
| Access & import/export | Data used are acquired from fishery-independent coral reef surveys as part of the National Oceanic and Atmospheric Administration (NOAA) Pacific Reef Assessment and Monitoring Program (RAMP; 2010-2012) and NOAA's National Coral Reef Monitoring Program (NCRMP; 2013-2019). These data are made publicly available and can be accessed for all NCRMP jurisdictions online. In this instance, the study data was provided on request by Dr Tye Kindinger in NOAA's Pacific Islands Fisheries Science Center. |
| Disturbance | Only observational data were collected in this study, thus there was minimal disturbance caused. |

# Reporting for specific materials, systems and methods

We require information from authors about some types of materials, experimental systems and methods used in many studies. Here, indicate whether each material, system or method listed is relevant to your study. If you are not sure if a list item applies to your research, read the appropriate section before selecting a response.

## Materials & experimental systems

| n/a | Involved in the study |
|-----|----------------------|
| ☒ | Antibodies |
| ☒ | Eukaryotic cell lines |
| ☒ | Palaeontology and archaeology |
| ☒ | Animals and other organisms |
| ☒ | Clinical data |
| ☒ | Dual use research of concern |
| ☒ | Plants |

## Methods

| n/a | Involved in the study |
|-----|----------------------|
| ☒ | ChIP-seq |
| ☒ | Flow cytometry |
| ☒ | MRI-based neuroimaging |

