## [Peer Review File · Nature Ecology & Evolution]

Peer Review Information

Journal: Nature Ecology & Evolution

Manuscript Title: Local human impacts disrupt depth-dependent zonation of tropical reef fish communities

Corresponding author name(s): Laura E. Richardson

Editorial Notes:

Reviewer Comments & Decisions:Decision Letter, initial version:

9th January 2023

Dear Dr Richardson,

I am writing to you in the temporary absence of my colleague, **[REDACTED]**. Your Article, "Re-visiting the paradigm of coral reef depth zonation on contemporary reefs" has now been seen by 3 reviewers. You will see from their comments copied below that while they find your work of considerable potential interest, they have raised quite substantial concerns that must be addressed. In light of these comments, we cannot accept the manuscript for publication, but would be very interested in considering a revised version that addresses these serious concerns.

We hope you will find the reviewers' comments useful as you decide how to proceed. If you wish to submit a substantially revised manuscript, please bear in mind that we will be reluctant to approach the reviewers again in the absence of major revisions.

If you choose to revise your manuscript taking into account all reviewer and editor comments, please highlight all changes in the manuscript text file.

* Include a "Response to reviewers" document detailing, point-by-point, how you addressed each referee comment. If no action was taken to address a point, you must provide a compelling argument. This response will be sent back to the referees along with the revised manuscript.

* If you have not done so already we suggest that you begin to revise your manuscript so that it conforms to our Article format instructions at <http://www.nature.com/natecolevol/info/final-submission>. Refer also to any guidelines provided in this letter.

[REDACTED]

If you wish to submit a suitably revised manuscript we would hope to receive it within 6 months. If you cannot send it within this time, please let us know. We will be happy to consider your revision so long as nothing similar has been accepted for publication at Nature Ecology & Evolution or published elsewhere.

Nature Ecology & Evolution is committed to improving transparency in authorship. As part of our

efforts in this direction, we are now requesting that all authors identified as 'corresponding author' on published papers create and link their Open Researcher and Contributor Identifier (ORCID) with their account on the Manuscript Tracking System (MTS), prior to acceptance. This applies to primary research papers only. ORCID helps the scientific community achieve unambiguous attribution of all scholarly contributions. You can create and link your ORCID from the home page of the MTS by clicking on 'Modify my Springer Nature account'. For more information please visit www.springernature.com/orcid.

Thank you for the opportunity to review your work.

[REDACTED]

Reviewers' comments:

Reviewer #1 (Remarks to the Author):

This is a well-structured and well-written study that provides basin-scale patterns of reef fish trophic group and biomass zonation across different spatial scales. It is interesting to know how these differ across different groups and how these have been linked to ecological mechanisms such as upwelling.

It was also interesting to find out about thresholds of bathymetric steepness below which fish biomass is enhanced.

Overall, this a good study that would benefit from some edits / clarifications.

Below some overall comments, and more specific comments per section with line numbers.

Overall comments to address

- More methodological information, especially in relation to the survey method and how consistency in fish identification and fish length measurements was achieved.
- Bayesian modelling is not something that I have myself used before (all lines between 444-510), therefore, I cannot comment on if it has been applied correctly or not. I would advise having someone familiar with this method to review the statistical methodology used.
- Depth zonation patterns: I think most people seeing this term would instinctively think of zonation because of changes in community composition. I think it is important to clarify even in the title that this concerns zonation in biomass and/or trophic groups.

Introduction

Line 67. Define deep-water (e.g. >200m) or change to "particulate foods and nutrients derived from deeper water..."

Line 69. Increase instead of increases

Line 70. Define shallow-water

Line 82. span vast spaces of the ocean. This is because it has been argued to refer to one ocean (<https://doi.org/10.1002/aqc.3512>)

Results

Line 116: I would remove "predictably" as it's the results section

Table 1: I would change underlining negative associations to adding a - in front of numbers. e.g. -0.75. That is more intuitive to me.

Fig. S1: I would merge it with Fig. 1 as they both show important information. However, I am not wedded to the suggestion.

Lines 203-214. This would fit better in the materials and methods or introduction section. If something needs to be kept in the Results section I would only do that if absolutely necessary.

Line 224: "exceeding island-scale variance (9%; $P(\text{sdsSITE} > \text{sdsISLAND}) = 0.95$)". Is that not already captured in lines 261-218?

Line 225: Why does the site-scale variance here (38-64%) differ from that at line 25-52%? The same applies for ecoregion scale.

I think it has to do with the way they have been estimated - looking at Table S11 you talk about probabilities, so I would mention that in the main text as well.

Discussion

Line 239: Some of those studies, e.g. Goreau, 1959 talk about zonation focusing on species composition rather than changes in biomass.

Line 245: I would rephrase to "To our knowledge, this is the first time..."

Lines 245-246: However,

Line 255: Only unpopulated?

Lines 304-306: Can you expand a little bit more here? Do you mean that below 30m steepness has been found to have contrasting effects because the steepness in the studies sites was different?

Line 307: For most part the depth-related changes in biomass remained the for trophic groups, it's just the absolute values that were reduced, no?

Line 333-334: True, but you have not focused on species in this study. Would mentioning specific trophic groups that perform distinct functions be more accurate?

Line 377: Again refs 6 and 25 concern zonation because of changes in species and not trophic group biomass.

Lines 381-383: The direction of change was the same though. I know you mention it, but the way it's presented both here and in the discussion, is like it is very different.

The difference lies mostly in the absolute numbers. Maybe it's not that important, and I am getting too caught up in it.

Lines 386-388. Can you add examples of ecological paradigms that are included in the refs you cite, here?

General Discussion Comment: Something that was not explored. Why was variation between islands less than between sites or between ecoregions?

If site-scale variance is greater than island or ecoregion scale for most groups, then does this mean that conservation efforts should focus on site-level information more compared to information at higher spatial scales?

Methods

Line 413 onwards (Reef fish survey data section).

Divers: Where all the surveys conducted by the same team of divers? If not, then authors must acknowledge the variability in species identifications.

SPC curves: Then there is the question of the survey method itself. Can the authors cite other studies that have used this method?

Lengths: From what I understand actual fish lengths have been estimated (to the nearest cm) rather than fish size classes. I am struggling to understand how this could be achieved underwater without a significant amount of bias given that some fish could be as far away as 15m.

Can you explain a bit more on how consistency of length measurements was assured between divers across all the different surveys? Even if that information exists in another reference, it is worth mentioning it in this manuscript, given the large dataset that is being used.

Lines 440-442: Does this mean that 400m was cross-checked against all surveyed sites to make sure it does not include opposite sides of the island? What about encompassing the reef flat and reef crest zones?

Lines 444-451: I am not familiar with this method to be able to evaluate if it has been applied correctly. Can you cite 1-2 refs for the type of model you ended up using?

Line 480: What criterion was used to determine if the proportion of zeros was low enough?

Reviewer #2 (Remarks to the Author):

The study by Richardson and coauthors entitled: "Re-visiting the paradigm of coral reef depth zonation on contemporary reefs" (NATECOLEVOL-221117855) sets out to explore depth zonation patterns of coral reef fishes across gradients of biophysical processes and human disturbance. This study builds upon a broad body of work focused on contributing to our understanding of how habitat, depth, and other important biophysical drivers influence the structure and distribution of fishes. To date, most contributions to the literature have focused on describing these patterns at single locations or groups of sites across a limited spatial scale. This study complements these efforts by assembling a large and comprehensive dataset including over 5,000 in situ reef fish surveys spanning across 35 islands to explore these ecological patterns. Overall, I found the study to be well designed, scientifically interesting, and well written. I think the results provide valuable insights into how coral reef fish communities are structured in a changing world. However, I feel that there are a couple areas that should be considered before the manuscript is published. I have outlined these broad recommendations along with a couple more detailed comments below. Once these items are addressed, I feel that this manuscript will make a valuable contribution to the field of Ecology.

First, I would like the authors to consider how they filtered and selected the reef fishes for this study. While I appreciate the complexity and variability of coral reef fish data, I am concerned about the exclusion of ecologically important species (Sharks and Jacks). I am familiar with the literature identifying potential biases associated with large-bodied and mobile species as well as the behavioral effects associated with diver presence. Many of these biases or behavioral effects could be observed for other large-bodied mobile species within important groups such as parrotfishes (Scaridae), wrasses (Labridae) and snappers (Lutjanidae). I would like the authors to consider the potential biases associated with choosing to remove ecologically important species. For example, in most coral reef communities Jacks (Carangidae) are voracious predators exerting strong top-down effects on prey species. This is particularly evident with species such as *Caranx ignobilis*, *C. melampygus*, and *C. lugubris* that can be observed actively hunting across coral reef habitats preying on fishes and other invertebrates. This predatory behavior has been well documented in ecoregions included in this study (Hawaiian Islands and Line Islands). This is also the case with reef sharks (Carcharhinidae) excluded from this study (e.g., *Carcharhinus melanopterus*, *C. amblyrhynchos*, and *Triaenodon obesus*). I ask that the authors reconsider removing the entire group of sharks and jacks and be more selective in their filtering to factor in the ecological relevance of their choices.

Second, the authors take advantage of a previously published dataset (MacNeil et al. 2015) that provides an estimate of the total unfished global fish biomass to serve as the mean for analysis in this study. The intercept value was used for all trophic groups included in this study with the exception of the secondary consumers. While I do not disagree with this approach, I have concerns over the comparability of datasets. Specifically, the previous study uses a filtering process to exclude certain species. It does not seem like the authors in this study used the same filtering process to exclude the same species. Therefore, this would introduce potential biases in the analysis and statistical modeling outputs. Again, I recommend that the authors consider the ecological implications of the species filtering process and provide a justification for how they selected species or groups to exclude. I think the MacNeil work is great but maybe the authors from this study should consider using the approach they used for the secondary consumers when data were not comparable between studies.

Lastly, I found the dataset included in this study to be impressive and I appreciate the efforts involved with synthesizing data from across 5 ecoregions. As the authors point out, each of the 5 ecoregions included in this study are exposed to different biophysical forces and levels of human disturbance. I understand the major aim of this study was to compare patterns of reef fish biomass across depth gradients for 5 ecoregions. However, I think it would be interesting to see if these patterns hold up within each ecoregion. I found the results from figure 5 to be interesting and wanted explore these results more. I am not suggesting that the authors create additional figures etc. However, I would recommend that the authors provide some additional discussion to expand on their findings. Are the patterns consistent within each ecoregion?

Line 445 – (biomass density; g-m²) do you mean g m⁻²

Figures – In general I found the figures to be great. However, I am not sure if it was the download process but the figures appear to be low resolution and slightly blurry. I know a lot of work went into creating the figures and it would be good to confirm the quality for publication.

Table S1 – Aguijan categorized as “Populated”? I assume this is due to its proximity to Tinian and Saipan rather than its small resident population?

Reviewer #3 (Remarks to the Author):

This well-written paper uses a uniquely large data set to examine the effects of depth on reef fish assemblages, and how population pressure influences this pattern. The paper uses modern analytical techniques to provide a convincing story of depth-related gradients for both major trophic groups and total biomass. The paper also includes an interesting finding of the role of reef bathymetry on reef fishes – the influence of reef steepness on fish assemblages intuitively might be expected, but has rarely been demonstrated at this scale and with such good bathymetry data.

My issue with the paper is whether the story is sufficiently interesting for such a high-profile journal for two reasons: 1) the interest in revisiting basic depth gradients and examining homogenization by human impacts and 2) the lack of detailed covariates. The depth-related gradient of zones on reefs is indeed a well-described and accepted paradigm, although I would argue we now have enough descriptions of changes with depth that the ability to generalize is stronger than indicated here (first line of abstract, L77-80). The homogenization of assemblages because of human impacts is also well known in a range of ecosystems (including on reefs). Therefore, it isn't particularly surprising that the zonation is less clear near highly populated islands. I agree this hasn't been demonstrated as clearly as in this study and it needs to be quantified, but I question whether this elevates the paper to one that should be published in *Nature Ecology & Evolution*. More fundamentally, I'm not sure how this would change our approach to studying or conserving reefs – the underlying ecological paradigm is clearly still true given the significance of the variables that change with increasing depth, and management is rarely zone specific. For example, any meaningful study of patterns of fish assemblages already controls for variable fishing pressure. Therefore, does this study really suggest that revisiting existing paradigms would change our approach to studying or managing ecosystems?

More significantly this study feels like it misses opportunities to consider depth in the context of other variables, which potentially either excludes the chance of examining some interesting nuances in the data or possibly confounds the analyses. Firstly, human population pressure simply places reefs into one of two categories (populated or unpopulated) which seems overly simplified – why not consider population pressure as a continuous variable and look at more general trends and thresholds? Secondly, the analyses pool all sites from 'reef slopes', which standardizes the geomorphological zone but ignores any differences in benthic assemblages. Thus the reader is left wondering, for example, whether populated islands have more homogenous benthic assemblages across depth gradients that limits fish zonation. I'm sure the divers collected some benthic data, so why not include a basic metric like coral cover (or even habitat type) in the analyses to better isolate the depth signal? I appreciate that the aim was to examine a basic biophysical gradient (depth), but I feel the paper would have been more interesting to consider depth in the context of all the other variables that we know affect reef fishes. To me, one of the most interesting parts of this paper was examining the role of steepness on zonation and I kept wanting to know which other factors were critical in explaining the deviance in the data set. One of the benefits of modern ecology is the access to large-scale, reliable data layers of a range of covariates. However, this study doesn't use these rich sources of information to fill out the models of fish abundance. This is particularly problematic when considering spatial scales of variation (regions / islands / sites) – because the analyses only uses spatial scale in the model the Discussion has to be quite

speculative about the actual drivers of these patterns. But had the models included covariates of e.g. net primary productivity, temperature, or wave exposure (all of which are relatively straightforward to obtain) – or island type (see Taylor et al, 2018, *Ecography* 38:520-530 for why this variable might be critical) - then we could have seen actual correlations in the data rather than have to guess at what might be varying from island to island (“potential indicators”, L210).

In summary, had I been reviewing this for a general marine journal I would have suggested revisiting the analyses with some key additional covariates and perhaps human pressure as a continuous variable, but recommended the paper eventually be published as examining depth-related patterns with a data set of this size and examining the homogenizing role of human populations is worth quantifying. Plus the importance of steepness and the variation among trophic groups are interesting. However, the threshold for this journal is higher and I am not convinced that the underlying question is sufficiently interesting, especially given the relatively cursory treatment of other important covariates that limits understanding of which variables are critical at large scales.

Author Rebuttal to Initial comments

Reviewer #1 (Remarks to the Author):

R1.1: *This is a well-structured and well-written study that provides basin-scale patterns of reef fish trophic group and biomass zonation across different spatial scales. It is interesting to know how these differ across different groups and how these have been linked to ecological mechanisms such as upwelling. It was also interesting to find out about thresholds of bathymetric steepness below which fish biomass is enhanced. Overall, this a good study that would benefit from some edits / clarifications. Below some overall comments, and more specific comments per section with line numbers.*

We thank Reviewer #1 for their careful and considered evaluation of our manuscript. We are glad they found the results interesting, and the article well-structured and well-written. We have addressed the areas in the text that required clarification, revision, or expansion as suggested, taking into account the points they have raised.

Overall comments to address

R1.2: *More methodological information, especially in relation to the survey method and how consistency in fish identification and fish length measurements was achieved.
- Bayesian modelling is not something that I have myself used before (all lines between 444-510), therefore, I cannot comment on if it has been applied correctly or not. I would advise having someone familiar with this method to review the statistical methodology used.*

Thank-you. See our detailed responses to **R1.25–R1.27**, and **R1.29**.

R1.3: *Depth zonation patters: I think most people seeing this term would instinctively think of zonation because of changes in community composition. I think it is important to clarify even in the title that this concerns zonation in biomass and/or trophic groups*

Thanks for this. For now, and to favour brevity in the title for the journal, we have not incorporated this suggestion into the title, but we have revised it to make it more concise: ‘*Re-visiting the paradigm*

of depth zonation on contemporary coral reefs'. We agree with the Reviewer that further clarification of the term zonation is needed early in the introduction. In addition to the existing definition of ecological zonation as "the distribution of organisms across space" (L58), we have added a more detailed definition of 'depth zonation' at first mention as follows:

L67: "Here we revisit this basic principle in the context of depth zonation of tropical coral reef communities—the distribution of reef fish biomass among distinct trophic groups."

Introduction

R1.4: *Line 67. Define deep-water (e.g. >200m) or change to "particulate foods and nutrients dervied from deeper water..."*

Thanks. We have changed this to "particulate foods and nutrients derived from deeper water" as suggested (L73).

R1.5: *Line 69. Increase instead of increases. Changed.*

R1.6: *Line 70. Define shallow-water.*

Changed to "shallow-water (<30 m depth³⁷)" (L76).

- Stefanoudis et al. (2019). Depth-dependent structuring of reef fish assemblages from the shallows to the rariphotic zone. *Front Mar Sci* 6, 307.

R1.7: *Line 82. span vast spaces of the ocean. This is because it has been argued to refer to one ocean (<https://doi.org/10.1002/aqc.3512>)*

Thanks for this suggestion. To avoid the plural term "oceans", the text is changed to: "Modern-day island reefs span vast ocean expanses..." (L88).

Results

R1.8: *Line 116: I would remove "predictably" as it's the results section.*

Thanks, removed.

R1.9: *Table 1: I would change underlining negative associations to adding a - in front of numbers. e.g. -0.75. That is more intuitive to me.*

Thanks for this suggestion. We would agree that adding a "-" in front of the numbers to indicate decrease would be intuitive if the table presented depth and population status effect sizes on fish biomass. Unfortunately, adding "-" might indicate negative probability estimates. Classical probabilities range from 0–1, such that a negative probability would indicate that an event or observation is less likely than 'impossible'. Indeed, negative probability is discussed in relation to quantum mechanics, and is centred on the concept of event 'cancelation' (e.g. Feynman 1987). If the Reviewer agreed, an alternative option for presenting probabilities associated with a negative relationship between biomass and depth in this Table could be to change underlining negative associations to colouring the text in red (or an alternative colour), which we will gladly do on request.

- Feynman. (1987). Negative probability. *Quantum implications: essays in honour of David Bohm*, 235-248.

R1.10: *Fig. S1: I would merge it with Fig. 1 as they both show important information. However, I am not wedded to the suggestion.*

To aid clearer visualisation of the scaled effects of depth, bathymetric steepness, and human population status on the positive reef fish biomass data, the effect estimates of hurdle components (presence-absence) from the piscivore and planktivore models were not included in Fig. 1. Including them, as presented in Fig. S1, clusters the effect estimates and their percentiles such that the reader may be less able to effectively interpret relative effect size. However, we will gladly exchange Fig. 1 for Fig. S1 on request.

R1.11: *Lines 203-214. This would fit better in the materials and methods or introduction section. If something needs to be kept in the Results section I would only do that if absolutely necessary.*

Thanks. We have moved these lines as suggested to the methods (L522-534), and revised the new opening statement as follows:

L230: “The proportion of variation in fish biomass explained by each spatial scale, quantified by extracting the posterior standard deviations of these modelled random effects, varied among trophic fish groups (Fig. 5; Table S10 and S11; see Fig. S3 for variation in island-level depth effects among ecoregions).”

R1.12: *Line 224: "exceeding island-scale variance (9%; $P(sdsSITE > sdsISLAND) = 0.95$)". Is that not already captured in lines 261-218?*

Deleted this detail to streamline results, thank you.

R1.13: *Line 225: Why does the site-scale variance here (38-64%) differ from that at line 25-52%? The same applies for ecoregion scale. I think it has to do with the way they have been estimated - looking at Table S11 you talk about probabilities, so I would mention that in the main text as well.*

Thanks for highlighting this area of potential confusion. Line 217 presented the range in variance explained for all fish groups except secondary consumers (i.e. planktivores, piscivores, primary consumers and total fish biomass). L225 described the range in variance explained for all groups except planktivores (i.e. piscivores, primary consumers, secondary consumers, and total fish biomass). To avoid this potential confusion, we have removed the percentage ranges previously provided in L225 and retained reference to Fig. 5 and the complete summary in Table S10.

Discussion

R1.14: *Line 239: Some of those studies, e.g. Goreau, 1959 talk about zonation focusing on species composition rather than changes in biomass.*

This is correct. To make this explicit, we have revised this sentence as follows:

L256: “However, while the structuring force of depth on reef ecology featured among the earliest descriptions of tropical coral reefs (for example, zonation in species composition)^{7,27,28}, these observations were restricted to single-point locations and the generality of a depth zonation paradigm remains untested across broad geographies.”

R1.15: *Line 245: I would rephrase to "To our knowledge, this is the first time..."*

Changed as recommended, thanks (L265).

R1.16: *Lines 245-246: However,*

Apologies, we are unclear what the Reviewer is suggesting here. We will be glad to address this point if we can request some additional direction.

R1.17: *Line 255: Only unpopulated?*

This paragraph focusses on depth patterns at just unpopulated islands. However, we see that the original wording might have suggested that depth zonation was only observed at unpopulated reefs. We have revised as follows:

L275: “At geographically distinct unpopulated islands, we show that reef fish biomass of all broad trophic groups correlated predictably and relatively consistently across depth despite underlying variation in biophysical drivers known to affect standing reef fish biomass^{33,41,50}.”

R1.18: *Lines 304-306: Can you expand a little bit more here? Do you mean that below 30m steepness has been found to have contrasting effects because the steepness in the studies sites was different?*

We see that this sentence was potentially unclear. Studies looking at depth zonation beyond the 0–30 m limit in our study report contrasting peaks in abundance of piscivores and planktivores. For example, at Linden Bank, a submerged shoal on the outer shelf of the Great Barrier Reef, reported dominance of piscivores (and mobile invertivores) between 50–70 m depth (Scott et al. 2022). In contrast, the proportion of planktivores on forereefs at Enewetak Atoll in the Marshall Islands increased from 50% at 30 m to >90% from 90–200 m, which the authors suggest may relate to upwelling processes increasing plankton and shallow reef productivity (Thresher and Colin 1986). In L304-306, we suggest that these variable peaks in trophic group biomass at mesophotic depths are potentially indicative of spatial variation in upwelling, which could be linked—among other oceanographic factors—to variable local bathymetric steepness among those study locations. To make this clearer, we have revised the manuscript as follows:

L325: “Previous studies document variable peaks in planktivorous and piscivorous fishes at mesophotic depths beyond the 30 m limit of this study^{54,67}, which may be indicative of spatial variation in upwelling, potentially linked to—among other oceanographic factors—variable local bathymetric steepness among those study locations.”

- Thresher and Colin. (1986). Trophic structure, diversity and abundance of fishes of the deep reef (30–300m) at Enewetak, Marshall Islands. *Bull Mar Sci* 38, 253–272.
- Scott et al. (2022). Variation in abundance, diversity and composition of coral reef fishes with increasing depth at a submerged shoal in the northern Great Barrier Reef. *Rev Fish Biol Fish* 32, 941–962.

R1.19: *Line 307: For most part the depth-related changes in biomass remained the for trophic groups, it's just the absolute values that were reduced, no?*

This is a useful point, highlighting that better clarity in the text might avoid potential confusion. We have amended the sentence as follows:

L332: “Despite marked bathymetric gradients in fish biomass at unpopulated islands, we show that depth related changes in biomass were altered by depleted biomass baselines at islands inhabited by people. There was overall lower fish biomass across the depth gradient

for all trophic groups at populated locations. Further, the change in absolute biomass of planktivores, piscivores, and secondary consumers across depth was substantially reduced at populated islands, and depth zonation in primary consumers was lost.”

R1.20: *Line 333-334: True, but you have not focused on species in this study. Would mentioning specific trophic groups that perform distinct functions be more accurate?*

Thanks. We have changed this statement from discussing functionally important species to the need for protection of “...depth-constrained trophic groups that perform distinct and important functions” (L359).

R1.21: *Line 377: Again refs 6 and 25 concern zonation because of changes in species and not trophic group biomass.*

This is correct. As defined in the introduction, the term zonation refers to “the distribution of organisms across space” (L58), and we have clarified our use of the term ‘depth zonation’ at first mention, to include (though not be limited to) “the distribution of reef fish biomass among distinct trophic groups” with depth (L68; see **R1.3**). To remove any potential confusion, we have revised L377 in the original manuscript as follows:

L409: “We revisited ecological depth zonation— recognised as a fundamental structuring force of tropical coral reef communities over six decades ago^{7,26-28}—with two purposes: first, to test the generality of depth zonation of reef fish biomass for the first time across an ocean-basin scale; and second, to assess whether a zonation paradigm holds on reefs exposed to direct local human impacts.”

R1.22: *Lines 381-383: The direction of change was the same though. I know you mention it, but the way it's presented both here and in the discussion, is like it is very different. The difference lies mostly in the absolute numbers. Maybe it's not that important, and I am getting too caught up in it.*

Thanks for highlighting this. We have revised the statement as follows to clarify the exact result:

L415: “However, we found that while the principle of resource-driven depth zonation held at populated islands for some trophic groups (e.g. direction of change for planktivores, piscivores, secondary consumers), their magnitude of change with depth (i.e. absolute biomass) was substantially reduced by human depletion.”

R1.23: *Lines 386-388. Can you add examples of ecological paradigms that are included in the refs you cite, here?*

Thanks for highlighting potential confusion here. We had provided references to recent papers that discuss the need to revisit classic ecological paradigms for understanding contemporary coral reef ecology, including Hughes et al. 2017 *Nature*, and Williams et al. 2019 *Functional Ecology*. These articles emphasise how human impacts are increasingly transforming reefs into new configurations unlike anything observed previously by humans, requiring radical changes in how we study and manage them. The articles do not assess specific paradigms. However, the reference to Helmus et al. 2014 *Nature* offers an example of island biogeography theory no longer being an effective predictor of species richness in the Anthropocene. To bring in this example earlier we have added it to the introduction, reiterated the example in the opening discussion paragraph, and revised the final conclusion sentence for clarity, as follows:

L58: “Human impacts confound natural drivers of ecological organisation in most contemporary ecosystems^{20–22}, and historical paradigms are now failing to capture ecological patterns where there is anthropogenic forcing (e.g. island biogeography theory¹⁸).”

L272: “These findings support calls for revisiting and potentially updating twentieth century ecological paradigms (e.g. island biogeography theory¹⁸) that may no longer capture ecological patterns and processes in a human-dominated world²³.”

L420: “However, where the influence of a physical feature as fundamental as depth on coral reef ecological organisation is being disrupted, we reiterate recent calls to revisit classic and influential ecological paradigms to determine their relevance in this era of rapid change^{18,23,24}.”

- Helmus et al. (2014). Island biogeography of the Anthropocene. *Nature* 513(7519), 543-546.
- Hughes et al. (2017). Coral reefs in the Anthropocene. *Nature* 546(7656), 82-90.
- Williams et al. (2019). Coral reef ecology in the Anthropocene. *Funct Ecol* 33(6), 1014-1022.

R1.24: *General Discussion Comment: Something that was not explored. Why was variation between islands less than between sites or between ecoregions? If site-scale variance is greater than island or ecoregion scale for most groups, then does this mean that conservation efforts should focus on site-level information more compared to information at higher spatial scales?*

Thanks to the Reviewer for highlighting where further discussion on cross-scale spatial variation in fish biomass patterns would improve the manuscript. Coral reefs are considered to be highly hierarchical in structure, determined by processes occurring at multiple spatial and temporal scales (Hughes et al. 1999, MacNeil et al. 2009). As the Reviewer rightly highlights, within this hierarchical context, management and governance efforts are considered most effective when carried out at scales aligning with scales of ecological heterogeneity (Cumming and Dobbs 2020). We describe this in detail in the paragraph L335-345, and we link our results to scaled management and conservation efforts in L361-368 in the original manuscript. However, we now provide additional discussion exploring the relative differences in variation observed across scales, notably discussing the relatively lower variation observed at the island-scale as follows:

L389: “These findings align with previous studies that describe habitat composition at the site-level to be the likely dominant driver of reef fish metacommunity structure, including diversity and the biomass of most trophic groups, while attributing greater prevalence of planktivores to larger-scale gradients in overall ocean productivity⁴⁴. That we observed lower variation at the island-scale than site and ecoregion scales may be due to a dominant influence of habitat and background levels of productivity, over processes occurring at the island-scale. In this context, our findings suggest that management of primary consumers, piscivores, total standing biomass, and especially secondary consumers might achieve satisfactory outcomes at local within-island scales with no-take areas⁸⁰, habitat restoration, or better regulated destructive human activities⁷⁹. Local management of planktivores is no doubt also important³⁴. But, given the potential influence of regional-scale drivers on planktivore biomass production and the importance of this group as the prey base for higher trophic levels³², more nuanced, region-specific targets for recovery⁸¹ or take of planktivores may be advisable in areas of naturally lower primary production.”

- Cumming and Dobbs. (2020). Quantifying social-ecological scale mismatches suggests people should be managed at broader scales than ecosystems. *One Earth* 3(2), 251-259.
- Hughes et al. (1999). Patterns of recruitment and abundance of corals along the Great Barrier Reef. *Nature* 397:59–63
- MacNeil et al. (2009). Hierarchical drivers of reef-fish metacommunity structure. *Ecology* 90:252–264.

In response to a similar question of spatial scale by Reviewer #2 (R2.4), we have also now created an additional figure (Fig. S3 below) for the reader to visualise the observed within- and among-island scale variation in depth zonation (island-level depth effects, overlying global depth for comparison, across each trophic group; within-island variation is indicated by 95% credible intervals associated with island point estimates).

Methods

Line 413 onwards (Reef fish survey data section).

R1.25: *Divers: Where all the surveys conducted by the same team of divers? If not, then authors must acknowledge the variability in species identifications.*

Thanks for this important consideration. The National Oceanic and Atmospheric Administration (NOAA) has multiple trained fish survey divers who conduct *in situ* observations across sampling cruises, across distinct ecoregions and years. Divers must have a minimum of 30 underwater visual fish survey census dives conducted prior to joining a monitoring cruise. In addition, to ensure consistency in observer species identification and size estimation, NOAA employs extensive training, testing, and technical validation protocols both outside of the cruise period and during (detailed in Heenan et al. 2017). These protocols address the potential for intra- and inter-diver variability in two ways:

1. New and experienced divers receive full training in fish identification and survey protocol, in classroom and in-water sessions; prior to each survey cruise, all divers must accurately identify >90% of regional-specific fish species in a test that is specifically weighted towards rare species and those that have conspecifics with similar appearance.
2. For the timeframe of the data used here, between cruises divers conducted in-water training exercises to practice survey protocol, fish identification and fish size estimation (see response to comment R1.27 for detail on size estimation protocol and technical validation).

During the cruises, there are typically 4-10 fish survey divers who routinely discuss and compare species identification and sizes immediately after a survey, and during data entry. Diver pairs are continually rotated, and diver performance is estimated as the difference between the estimates of each diver and those of their dive partner at each site, calculated for total fish biomass, species richness, and body-size distributions of commonly observed species. As divers survey adjacent cylinders on the reef (i.e. not identical areas of the reef), real differences between diver observations are expected. But the diver performance estimate is measured to detect potential consistent bias made by each diver (i.e. if there is no consistent bias, the median difference with their buddy partner should be close to zero). Diver performance is measured every few days during survey cruises to allow for early detection of observer error (Heenan et al. 2017).

In addition to these technical validation protocols, we included a group-level random intercept for 'diver identity' in all of our statistical models of fish biomass to account for any remaining effect of observer bias. By assuming inherent correlations among divers and their observations that affect the estimated means and associated errors, we were then able to estimate isolated population level effects (i.e. depth, human population status, bathymetric steepness) (*sensu* Macneil et al. 2015 Nature).

- Heenan et al. (2017). Long-term monitoring of coral reef fish assemblages in the Western central pacific. *Sci Data* 4(1), 1-12.
- MacNeil et al. (2015). Recovery potential of the world's coral reef fishes. *Nature* 520(7547), 341-344.

We have revised the manuscript as follows to provide the required additional information:

L457: “Surveys were conducted by multiple observers across the study ecoregions and years. NOAA employs extensive training and technical validation protocols to ensure consistency and avoid bias in survey technique, fish species identification, and size estimation⁴⁹. Full details on SPC survey methods and technical validation steps are available in [⁴⁹]. To further mitigate any confounding effect of observer bias among fish surveys, we included ‘diver identity’ as a random intercept in all statistical models (described below) (*sensu*⁵³).”

L514: “A random intercept for ‘diver identity’ was included to account for any specific effect of observer bias by assuming an inherent correlation structure among divers and their observations (*sensu*⁵³).”

R1.26: *SPC surveys: Then there is the question of the survey method itself. Can the authors cite other studies that have used this method?*

Yes – the stationary point count is one of the three main methods used to survey coral reef fish assemblages globally (Caldwell et al. 2016). This survey method has been used since 2010 by the United States government agency, the National Oceanic and Atmospheric Administration (NOAA) in their Pacific monitoring program (Towle et al. 2022). The full details of which are available in the data descriptor paper (Heenan et al. 2017). The dataset has been used in numerous peer-reviewed scientific publications, some of which are cited in the manuscript (now referenced in L453 of the revised manuscript):

- Caldwell et al. (2016). Reef fish survey techniques: Assessing the potential for standardizing methodologies. *PLoS One* 11(4): e0153066.
- Heenan et al. (2019). Natural variation in coral reef trophic structure across environmental gradients. *Front Ecol Environ* 18(2), 69-75.
- Towle et al. (2022). A National Status Report on United States Coral Reefs Based on 2012–2018 Data From National Oceanic and Atmospheric Administration’s National Coral Reef Monitoring Program. *Front Mar Sci*, 8.
- Williams et al. (2015). Human, oceanographic and habitat drivers of central and western Pacific coral reef fish assemblages. *PLoS One* 10(4), e0120516.
- Yeager et al. (2017). Scale dependence of environmental controls on the functional diversity of coral reef fish communities. *Glob Ecol Biogeogr* 26(10), 1177-1189.

L453: “The abundance and body-size of all diurnal, non-cryptic reef fishes were estimated using stationary point count (SPC) surveys (*sensu*^{8,9,15,49,82,83}).”

R1.27: *Lengths: From what I understand actual fish lengths have been estimated (to the nearest cm) rather than fish size classes. I am struggling to understand how this could be achieved underwater without a significant amount of bias given that some fish could be as far away as 15m. Can you explain a bit more on how consistency of length measurements was assured between divers across all the different surveys? Even if that information exists in another reference, it is worth mentioning it in this manuscript, given the large dataset that is being used.*

This is another important question. As in response to **R1.25**, NOAA employs extensive training and technical validation protocols to ensure consistency in sampling. This includes practice size calibration dives during the out of cruise season. These calibration surveys involve divers visually estimating the size of anchored wooden cut-outs of fish with known body lengths, placed throughout the full extent of the survey cylinder (**Fig. R1a**). While it is true that there is a degree of variability in novice divers, or those new to the training program (**Fig. R1c**), the experienced trained staff are reasonably consistent (**Fig. R1b**) with no systematic tendency to over or underestimate size. It is also important to note that as divers remain at the centre of their cylinder, except towards then end of the survey when the size of smaller benthic site-attached species are estimated, typically the furthest distance the diver will be from any fish that is being estimated in length is 7.5 m, not 15 m.

Figure R1: Taken from Heenan et al. (2017) *Scientific Data*: During training dives, observer accuracy is assessed by divers estimating the size of wooden fish models of known lengths (a), which are haphazardly distributed throughout a mock SPC cylinder. Example mean difference (+—standard error) between actual and estimated length of model fishes by trained staff (b—open circles) and by trainee survey divers (c—closed circles) during size estimation training trials between 2014–2016. The closer the difference between size estimates and actual model sizes is to zero, the more accurate the sizing. Trainee fish survey divers, which includes novices in the middle of the training program, people who have done fish surveys but not the SPC method, as well as people who are taking an SPC refresher tend to towards more variable size estimates compared to core staff. Typically new fish surveyors are required to have conducted a minimum of 30 survey dives prior to joining a RAMP cruise.

L458 has been revised to highlight the technical validation protocols designed to ensure consistency in length measurements, with reference to the data descriptor paper (Heenan et al. 2017): “NOAA employs extensive training and technical validation protocols to ensure consistency and avoid bias in survey technique, fish species identification, and size estimation⁴⁹. Full details on SPC survey methods and technical validation steps are available in [49].”

R1.28: Lines 440-442: *Does this mean that 400m was cross-checked against all surveyed sites to make sure it does not include opposite sides of the island? What about encompassing the reef flat and reef crest zones?*

We selected an automated process to estimate mean bathymetric steepness within a 400-m radial buffer around each site to remove potential subjectivity in the method. We confirm that we visually cross-checked all 2,262 survey sites to see whether these buffers overlapped landmass (N.B. landmass areas were excluded as standard). While there was no observed overlap of the buffers across opposite

sides of the primary 35 study islands, we found a total of 15 sites where some degree of overlap of an adjoining islet or peninsula was observed (<1% of sites). Ten of these cases occurred where the buffer encompassed the end of an islet or a protruding peninsula, which would require drawing an arbitrary line in the buffer from the islet out to the buffer circumference in order to differentiate one area of ocean from the other (e.g. **Fig. R2a,b**). Given that the hydrodynamics at each site are predominantly unknown (i.e. the likely direction of currents and upwelling), we selected not to introduce this bias and accepted a level of uncertainty introduced by our *a priori* determination of radial buffer extent (L436-439 original manuscript: “A radial buffer-size of 400 m was selected to encompass depths that would capture the propensity for pulsed delivery of nutrient-rich sub-thermocline water by upwelling^{86,87} and for this upwelling to propagate onto shallow reef habitats at depths ≤ 30 m³⁵.”). The five remaining cases (representing 0.2% of the total sites) included landmass where the buffer was split into two areas of sea (e.g. A1 and A3 in **Fig. R2c**). We calculated the percentage of forereef area included from opposite sides of the landmass that contributed to the estimate of mean bathymetric steepness for these sites as: 1.0%, 2.1%, 5.0%, 24.1%, and 43.1% (**Table R1**).

We selected this automated method of estimating coarse-scale bathymetric steepness to mitigate potential subjectivity in a site-by-site assessment. However, prompted by the Reviewer’s question and to ensure the highest levels of confidence in our data, we have subsequently decided to remove two sites where the percentage of the buffer area that was located on an opposite side of a landmass contributed >5% to the bathymetry estimates (i.e. sites TUT-00376 and TUT-00516; **Table R1**; **Fig. R2c**). We have also removed an additional 7 sites that are officially classified as ‘forereef’ in NOAA’s National Coral Reef Monitoring Program (Heenan et al. 2017) but could also be classified as reef crest or reef flat. We have re-run all models and adjusted the minor details of results (e.g. precise probability estimates) throughout, though we note that the broad study results remain unchanged from the original submission.

To clarify our cross-checking process and site filtering approach, we have revised the methods section as follows:

L490: “All sites were visually cross-checked for island overlap, and those including >5% radial-buffer bathymetry on the opposite site of a landmass were excluded from all analyses.”

Figure R2: Example 400-m radial buffers used to estimate site-level mean bathymetric steepness where the buffers overlapped islets or peninsula landmass. a, b: Examples of cases where buffers encompassed landmass and would require drawing a line from land to the buffer circumference to differentiate areas to include/exclude. c: an example of where the landmass divided the buffer into two areas (A1 and A3).

Table R1: Proportion of 400-m radial buffer area on the opposite side of a landmass area contributing to site-level average bathymetric steepness estimates. Site names include the three-letter island reference codes detailed in Table S1.

Site name	A1: Site side (m ²)	A3: Opposite side (m ²)	Percentage of area on opposite side contributing to site average estimate (%)
PAG-00379	340,157	7,268	2.1
TUT-00376	257,666	195,131	43.1
TUT-00516	343,405	109,209	24.1
OFU-00437	176,259	1,865	1.0
OAH-00300	356,730	18,672	5.0

R1.29: Lines 444-451: I am not familiar with this method to be able to evaluate if it has been applied correctly. Can you cite 1-2 refs for the type of model you ended up using?

We have added a reference to the peer-reviewed paper by Bürkner (2017) which provides a detailed description of the Bayesian multi-level statistical model approach using the R software package `brms` as used in our study. The paper includes description of the multi-level regression models we have applied, the use of priors and selection of population- or group-level specific family parameters, and a worked example:

- Bürkner (2017). `brms`: An R Package for Bayesian Multilevel Models Using Stan. *J Stat Softw* 80, 1–28.

R1.30: *Line 480: What criterion was used to determine if the proportion of zeros was low enough?*

We tested initial biomass models with a hurdle structure and found that the hurdle component only affected model results when the number of zeros were sufficient (i.e. >5%). To accommodate zero-count observations by fitting a hurdle model, a sufficient contrast (i.e. a sufficient number of zeros and ones) is required to be able to estimate effects in the presence-absence component of the model. If nearly all values are >0, then there are only a few datapoints to inform the model about factors that lead to a zero outcome. In those cases, the probability of positives will be close to one (particularly with a dataset as large as in this study since there are 1000s of positive data observations), and the hurdle component will only add noise. In the case of primary and secondary consumer biomass where the percentage of zeros was 1.09% and 0.05% respectively, in our dataset, the low contrast for the hurdle components affected model convergence and only added noise, so those were deemed not useful. To clarify this, we have provided additional description in the methods as follows:

L542: “Where the proportion of zeros was too low to effectively estimate effect sizes in the presence-absence component (i.e. an insufficient contrast between the number of zeros and ones), the use of a hurdle structure affected model convergence and only added noise. This occurred for primary consumers (1.09% zeros) and secondary consumers (0.05% zeros), so for these groups the zero biomass replicates were removed from the analysis and the Gamma model detailed above was fitted.”

Reviewer #2 (Remarks to the Author):

R2.1: *The study by Richardson and coauthors entitled: “Re-visiting the paradigm of coral reef depth zonation on contemporary reefs” (NATECOLEVOL-221117855) sets out to explore depth zonation patterns of coral reef fishes across gradients of biophysical processes and human disturbance. This study builds upon a broad body of work focused on contributing to our understanding of how habitat, depth, and other important biophysical drivers influence the structure and distribution of fishes. To date, most contributions to the literature have focused on describing these patterns at single locations or groups of sites across a limited spatial scale. This study complements these efforts by assembling a large and comprehensive dataset including over 5,000 in situ reef fish surveys spanning across 35 islands to explore these ecological patterns. Overall, I found the study to be well designed, scientifically interesting, and well written. I think the results provide valuable insights into how coral reef fish communities are structured in a changing world. However, I feel that there are a couple areas that should be considered before the manuscript is published. I have outlined these broad recommendations along with a couple more detailed comments below. Once these items are addressed, I feel that this manuscript will make a valuable contribution to the field of Ecology.*

Thanks to Reviewer #2 for their positive comments and thoughtful evaluation of our manuscript. We are pleased they found the article to be interesting, valuable to the field of Ecology, well-designed, and well-written. We have addressed the broad and detailed recommendations made, editing, or adding the required detail in the revised submission.

R2.2: *First, I would like the authors to consider how they filtered and selected the reef fishes for this study. While I appreciate the complexity and variability of coral reef fish data, I am concerned about the exclusion of ecologically important species (Sharks and Jacks). I am familiar with the literature identifying potential biases associated with large-bodied and mobile species as well as the behavioral effects associated with diver presence. Many of these biases or behavioral effects could be observed for other large-bodied mobile species within important groups such as parrotfishes (Scaridae), wrasses (Labridae) and snappers (Lutjanidae). I would like the authors to consider the potential biases associated with choosing to remove ecologically important species. For example, in most coral reef communities Jacks (Carangidae) are voracious predators exerting strong top-down effects on*

prey species. This is particularly evident with species such as *Caranx ignobilis*, *C. melampyngus*, and *C. lugubris* that can be observed actively hunting across coral reef habitats preying on fishes and other invertebrates. This predatory behavior has been well documented in ecoregions included in this study (Hawaiian Islands and Line Islands). This is also the case with reef sharks (Carcharhinidae) excluded from this study (e.g., *Carcharhinus melanopterus*, *C. amblyrhynchos*, and *Triaenodon obesus*). I ask that the authors reconsider removing the entire group of sharks and jacks and be more selective in their filtering to factor in the ecological relevance of their choices.

We thank the Reviewer for raising this. We agree that it would be interesting to explore piscivore biomass trends including sharks and jacks (despite their known daily vertical migrations up and down the reef slope; Meyer et al. 2007, Afonso et al. 2009, Vianna et al. 2013) if we had robust, unbiased estimates of these larger predators. As the Reviewer notes, piscivorous species of jacks and sharks are critically important to trophic dynamics and ecological organisation on coral reefs via top-down predation processes. However, as the Reviewer also highlights, published evidence shows that the body-size, mobility, curiosity, and trophic position of sharks and jacks influences their detectability and can introduce substantial sampling bias (Kulbicki 1998), particularly in surveys involving stationary divers (as in our study; Heenan et al. 2019). Common ‘mobbing’ behaviour of survey divers by sharks and jacks can lead to systematic overestimates in density (Parrish and Boland 2004), particularly in remote unpopulated areas where mobile roving piscivores can remain curious and unafraid of humans such as in the northwestern Hawaiian Islands (Williams et al. 2015). To mitigate this method bias, and in particular likely spatial variation in sampling bias among populated vs unpopulated locations, sharks and jacks are routinely excluded from reef fish survey analyses (*sensu* MacNeil et al. 2015; **Table R2**).

Table R2 Examples of studies globally that exclude sharks and other large semi-pelagic (associated with both shallow reef and pelagic environments) piscivorous fish such as jacks.

Geographic study region	Excludes sharks	Excludes sharks and semi-pelagics
Global	D’Agata et al. 2016 Roy Soc Proc B	MacNeil et al. 2015 Nature Cinner et al. 2016 Nature
Caribbean		Hawkins and Roberts 2004 Cons Biol
Pacific	D’Agata et al. 2016 Nat Comms Gray et al. 2016 PLoS ONE	Williams et al. 2015 PLoS ONE Yeager et al. 2017 Glob Ecol Biogeog
Indian Ocean	McClanahan et al. 2009 MEPS	McClanahan et al. 2020 Aqu Conserv Samoilys et al. 2018 PLoS ONE Cowburn et al. 2018 Mar Pol Bull

Unlike jacks and sharks, ‘mobbing’ behaviour towards divers by other large-bodied mobile species of reef fish such as parrotfishes, wrasses, and snappers is less routinely observed, as they may be less strongly affected by diver-presence (Longo and Floeter 2012). Our study authors have observed some hair-raising instances of diver-mobbing by the snapper, *Lutjanus bohar*, in the southern Line Islands (a geography not included in our study) where they are dominant in abundance over sharks (G. J. Williams, *pers. comms.*). However, such occurrences appear specific to that geography and are less commonly reported than the diver-positive behaviour documented for sharks and jacks, specifically in our study region. Indeed, efforts to quantify the effect of diver-presence on detectability of reef fishes across distinct families have described these other large-bodied groups as either diver-neutral (no detected effect on observed abundance, e.g. Lutjanidae), or diver-negative (where diver-disturbance reduces rather than increases detectability, e.g. Scaridae, Labridae, Serranidae) (Emslie et al. 2018).

- Afonso et al. (2009). Multi-scale patterns of habitat use in a highly mobile reef fish, the white trevally *Pseudocaranx dentex*, and their implications for marine reserve design. *Mar Ecol Prog Ser* 381, 273-286.
- Emslie et al. (2018). Reef fish communities are spooked by scuba surveys and may take hours to recover. *PeerJ* 6, e4886.
- Heenan et al. (2019). Natural variation in coral reef trophic structure across environmental gradients. *Front Ecol Env* 18(2), 69-75.
- Kulbicki. (1998). How the acquired behaviour of commercial reef fishes may influence the results obtained from visual censuses. *J Exp Mar Biol Ecol* 222(1-2), 11-30.
- Longo and Floeter. (2012). Comparison of remote video and diver's direct observations to quantify reef fishes feeding on benthos in coral and rocky reefs. *J Fish Biol* 81(5), 1773-1780.
- Meyer et al. (2007). Seasonal and diel movements of giant trevally *Caranx ignobilis* at remote Hawaiian atolls: implications for the design of marine protected areas. *Mar Ecol Prog Ser* 333, 13-25.
- Parrish and Boland. (2004). Habitat and reef-fish assemblages of banks in the northwestern Hawaiian Islands. *Mar Biol* 144: 1065–73.
- Vianna et al. (2013). Environmental influences on patterns of vertical movement and site fidelity of grey reef sharks (*Carcharhinus amblyrhynchos*) at aggregation sites. *PLoS ONE* 8(4), e60331.
- Williams et al. (2015). Human, oceanographic and habitat drivers of central and western Pacific coral reef fish assemblages. *PLoS ONE* 10: e0120516.

Therefore, while we appreciate the Reviewer's point and reiterate that we would consider exploring trends with sharks and jacks *if* we had unbiased estimates, we believe that including them from this data source where there is known systematic method bias would be misleading. Further, we are keen to ensure comparability between our study and existing publications in the literature that quantify unfished biomass baselines and drivers of reef fish assemblage structure (e.g. **Table R2**). However, we recognise that more detailed explanation for our filtering method would better equip the reader, which we have now added to the revised manuscript as follows:

L469: “Sharks and jacks (families Carcharhinidae, Carangidae, Sphyrnidae) were also excluded as these highly mobile, large-bodied, roving piscivores are known to be affected by the presence of stationary divers, typically resulting in systematic over-inflation of visual survey density estimates⁸⁵ (*sensu*^{8,53}). The presence of divers was also a potential source of differential bias of biomass estimates of these fishes among study locations, with ‘mobbing’ behaviour by jacks and sharks known to occur particularly in remote, unpopulated areas such as the northwestern Hawaiian Islands^{8,50}.”

R2.3: *Second, the authors take advantage of a previously published dataset (MacNeil et al. 2015) that provides an estimate of the total unfished global fish biomass to serve as the mean for analysis in this study. The intercept value was used for all trophic groups included in this study with the exception of the secondary consumers. While I do not disagree with this approach, I have concerns over the comparability of datasets. Specifically, the previous study uses a filtering process to exclude certain species. It does not seem like the authors in this study used the same filtering process to exclude the same species. Therefore, this would introduce potential biases in the analysis and statistical modeling outputs. Again, I recommend that the authors consider the ecological implications of the species filtering process and provide a justification for how they selected species or groups to exclude. I think the MacNeil work is great but maybe the authors from this study should consider using the approach they used for the secondary consumers when data were not comparable between studies.*

We are glad the Reviewer broadly agrees with our approach for setting priors (“*the mean for analysis in this study*”) for our models of fish biomass, albeit with reservations about the potential incomparability of datasets between our study and that by MacNeil et al. (2015). By using ‘prior’

information on global unfished biomass estimates from MacNeil et al., we provide the models with a ‘real world’ starting point to begin exploring the basis of the given conjectures (e.g. whether or not there is a depth effect). Incorporating prior information is the defining characteristic of Bayesian philosophy, allowing us to directly integrate information about the scientific context and remove the implicit equal weighting of possibilities that is more akin to frequentist analyses (Lemione 2019, McElreath 2020). It is worth noting that, in general, “the prior data and new data can be of different types” (McElreath 2020).

Nevertheless, while not identical, our approach to filtering jacks and reef-sharks is very close to that of MacNeil et al., who “*retained counts of diurnally active, non-cryptic reef fish that are resident on the reef slope, excluding sharks and semi-pelagics*”. We explicitly inflated the prior standard deviation in the intercepts for our models, which is based on the MacNeil et al. study, to account for factors that may limit comparability. These factors include species filtering approaches, but also geographical representativeness, and census method. In other words, we selected the MacNeil et al. study as the most relevant information to base a prior mean on (i.e., to determine the order of magnitude we might expect in unfished biomass on a reef), we do not think that the precision in their estimates would be useful for our prior. We therefore inflated our prior standard deviation considerably (by an order of magnitude, given the prior is on the log-scale). Given the large amounts of data that inform our models (previously $n=5,549$, now $n=5,525$ stationary point count fish surveys), the prior had a negligible impact, and estimated mean densities were often significantly shifted from the prior. We have added clarification about our prior specification to the methods (detailed below), and have added an additional figure to the Supplemental Information showing the prior and posterior intercept for unpopulated islands to illustrate how the data moves our posterior estimates from the prior (Fig. S4 below).

- Lemoine (2019). Moving beyond noninformative priors: why and how to choose weakly informative priors in Bayesian analyses. *Oikos*, 128(7), 912-928.
- McElreath (2020). *Statistical rethinking: A Bayesian course with examples in R and Stan*. 2nd Edition, Chapman and Hall/CRC, New York, pp 612.

L549: “This study builds on existing knowledge established in previous research that estimated a global baseline of total resident reef fish biomass in the absence of fishing⁵³. We integrate this prior information by using their published posterior biomass estimate (1,013 kg ha⁻¹) as the mean of the prior for log of total biomass (α ; converted to g m⁻²) (with standard deviation set at 1).”

L560: “MacNeil et al.⁵³ and our study employ comparable data (i.e. *in situ* counts of diurnally active, non-cryptic reef fish on forereef slopes, excluding sharks and semi-pelagics such as jacks). However, to account for potential differentiating factors between the studies, such as species filtering approaches, census method, or geographical representativeness, we inflated the prior standard deviation in the intercepts for our models by an order of magnitude. Model priors are detailed in Table S3 and plotted with unpopulated posterior intercept estimations in Fig. S4.”

Figure S4 Biomass model prior distributions and unpopulated posterior intercept estimations for each trophic group of reef fish. Specified prior distributions for each biomass group are shown with the blue line. Posterior intercept distributions are shown in orange.

R2.4: *Lastly, I found the dataset included in this study to be impressive and I appreciate the efforts involved with synthesizing data from across 5 ecoregions. As the authors point out, each of the 5 ecoregions included in this study are exposed to different biophysical forces and levels of human disturbance. I understand the major aim of this study was to compare patterns of reef fish biomass across depth gradients for 5 ecoregions. However, I think it would be interesting to see if these patterns hold up within each ecoregion. I found the results from figure 5 to be interesting and wanted explore these results more. I am not suggesting that the authors create additional figures etc. However, I would recommend that the authors provide some additional discussion to expand on their findings. Are the patterns consistent within each ecoregion?*

We thank Reviewer #2 for this helpful suggestion – we agree that the consistency of the patterns among sites, islands, and ecoregions is interesting and warrants further discussion (also per **R1.24**). While our study primarily strives to quantify generality in depth effects across the dataset (L98 original manuscript; response to **R3.3** and revised text L106-113), we also emphasise the role of scale-dependent biophysical gradients that likely influence the spatial variation in fish biomass trends we observed across the Pacific (manuscript Fig.5, Tables S10 and S11). We have added discussion on the spatial heterogeneity observed across scales (detailed below) and the consistency of fish biomass depth zonation patterns across scales, including ecoregion (detailed below). The Reviewer’s comment also inspired us to create an additional figure for the Supplemental Information (Fig. S3 below) to help the reader visualise spatial consistency, by showing depth effects at each of the study islands (within each ecoregion) and where depth zonation patterns deviated from the global depth effects. Broadly, few islands across the study ecoregions deviated from the global depth effect, indicating spatial consistency in the observed patterns. Though we note high levels of within-island variation in these estimates (indicated by the 95% credible intervals for each island), consistent with the high levels of observed site-level variation in the study (Fig.5, Table S10).

Results:

L230: “The proportion of variation in fish biomass explained by each spatial scale, quantified by extracting the posterior standard deviations of these modelled random effects, varied among trophic fish groups (Fig. 5; Table S10 and S11; see Fig. S3 for variation in island-level depth effects among ecoregions).”

Discussion:

L262: “These patterns hold true across the study area which spans distinct biogeographic regions, with high spatial consistency across islands and ecoregions (Fig. S3) despite varying spatial heterogeneity in fish biomass observed among trophic groups suggesting the role of distinct scale-dependent drivers.”

L275: “At geographically distinct unpopulated islands, we show that reef fish biomass of all broad trophic groups correlated predictably and relatively consistently across depth despite underlying variation in biophysical drivers known to affect standing reef fish biomass^{33,41,50}.”

L372: “While there was minimal observed deviation from the global depth effect across the study islands and ecoregions, our results show that spatial variation in fish biomass—across site, island, and ecoregion scales—was differentially and unevenly distributed among trophic groups, underscoring the importance of scale of observation in ecological enquiry^{10,77}.”

L389: “These findings align with previous studies that describe habitat composition at the site-level to be the likely dominant driver of reef fish metacommunity structure, including diversity and the biomass of most trophic groups, while attributing greater prevalence of planktivores to larger-scale gradients in overall ocean productivity⁴⁴. That we observed lower variation at the island-scale than site and ecoregion scales may be due to a dominant influence of habitat and background levels of productivity, over processes occurring at the island-scale.”

Figure S3 Island-level depth effects on reef fish biomass among distinct study ecoregions (Samoa Islands; Phoenix/Tokelau/Northern Cook islands (P/T/NC); Mariana Islands; Line Islands; Hawaii Islands). Point estimates and associated 95% credible interval (CI) lines indicate the effect of increasing depth on fish biomass (proportional change) at each island (increase if >1 , decrease with depth if <1). The population status of islands is indicated by CI line thickness (thick CI, populated; thin CI, unpopulated). Pale shaded boxes underlying point estimates and CIs represent the average proportionate global effect of depth on biomass given the human population status and average bathymetric steepness, and are provided to aid visual assessment of spatial consistency in depth effects across the study. For example, at islands where depth effects on fish biomass follow the over-all global depth trend for a given trophic group, point estimates are positioned on top of the pale shaded boxes. Conversely, at islands where the effect of depth differs from the global effect, point estimates are positioned outside the shaded box (e.g., in Kauai, there is a greater observed decrease in the biomass density of primary consumers with increasing depth than the over-all effect of depth for that group at populated islands; in Tutuila there is a greater proportionate increase in planktivore biomass with increasing depth relative to the global depth effect for that group at populated islands).

R2.5: Line 445 – (biomass density; $g\ m^{-2}$) do you mean $g\ m^{-2}$?

Corrected, thanks.

R2.6: *Figures – In general I found the figures to be great. However, I am not sure if it was the download process but the figures appear to be low resolution and slightly blurry. I know a lot of work went into creating the figures and it would be good to confirm the quality for publication.*

We confirm that that all the figures have a minimum resolution of 300 dpi and saved at a maximum width of 180 mm per the journal requirements. The figures presented in the original submission were lower resolution to reduce the manuscript file size (apologies).

R2.7: *Table S1 – Aguijan categorized as “Populated”? I assume this is due to its proximity to Tinian and Saipan rather than its small resident population?*

This is correct. We classified islands as ‘populated’ based on having >50 residents or located <100 km from the nearest larger human settlement (*sensu* Williams et al. 2011; Williams et al. 2015), per L400-403 (in original submission).

- Williams et al. (2011). Differences in reef fish assemblages between populated and remote reefs spanning multiple archipelagos across the Central and Western Pacific. *J Mar Biol* 2011:826234.
- Williams et al. (2015). Local human impacts decouple natural biophysical relationships on Pacific coral reefs. *Ecography* 38(8), 751-761.

Reviewer #3 (Remarks to the Author):

R3.1:

This well-written paper uses a uniquely large data set to examine the effects of depth on reef fish assemblages, and how population pressure influences this pattern. The paper uses modern analytical techniques to provide a convincing story of depth-related gradients for both major trophic groups and total biomass.

The paper also includes an interesting finding of the role of reef bathymetry on reef fishes – the influence of reef steepness on fish assemblages intuitively might be expected, but has rarely been demonstrated at this scale and with such good bathymetry data.

We thank Reviewer #3 for their positive comments and specific consideration of the overall position of our study in the context of the scope of *Nature Ecology & Evolution*. We are pleased they consider the manuscript to be well-written, employing a “uniquely large” and high-resolution dataset and “modern analyses” to provide a convincing story, with interesting results that have “rarely been demonstrated at this scale”. We have addressed the concerns raised, providing detailed explanation where possible and added clarification in this response document and in the revised manuscript, accordingly.

R3.2: *My issue with the paper is whether the story is sufficiently interesting for such a high-profile journal for two reasons: 1) the interest in revisiting basic depth gradients and examining homogenization by human impacts and 2) the lack of detailed covariates. The depth-related gradient of zones on reefs is indeed a well-described and accepted paradigm, although I would argue we now have enough descriptions of changes with depth that the ability to generalize is stronger than indicated here (first line of abstract, L77-80). The homogenization of assemblages because of human impacts is also well known in a range of ecosystems (including on reefs). Therefore, it isn't particularly surprising that the zonation is less clear near highly populated islands. I agree this hasn't been demonstrated as clearly as in this study and it needs to be quantified, but I question*

whether this elevates the paper to one that should be published in Nature Ecology & Evolution. More fundamentally, I'm not sure how this would change our approach to studying or conserving reefs – the underlying ecological paradigm is clearly still true given the significance of the variables that change with increasing depth, and management is rarely zone specific. For example, any meaningful study of patterns of fish assemblages already controls for variable fishing pressure. Therefore, does this study really suggest that revisiting existing paradigms would change our approach to studying or managing ecosystems?

We are disappointed (and sorry) that Reviewer #3 did not find the study as interesting as Reviewers #1 and #2. We are encouraged though that they found the demonstrated role of reef steepness on reef fishes interesting and novel in the scale at which it is demonstrated and that they recognise the uniquely large dataset and modern analytical techniques used. We are also pleased that the Reviewer recognises that while depth zonation is an accepted paradigm, as is the increasing propensity for ecological homogenisation under mounting human impacts, that these things need to be quantified and that we offer the clearest empirical demonstration of these to date.

Nonetheless, the Reviewer's comment has highlighted that the broader context and importance of our enquiry could be made clearer for the reader. As described by Underwood et al. (2000), we cannot make progress on understanding processes in ecology until we have understood the patterns of taxa distribution across scales relevant to (and as large as) biophysical gradients that structure communities. This requires that for testing principle ecological theories, widely accepted or otherwise, observations must be made across large-scales (in time, or space, or both) (Underwood et al. 2000, Marquet 2009). Until recently, testing ecological theory at (and across) large enough scales simply has not been possible due to a lack of spatially comprehensive, standardised ecological data, and accessible statistical tools (Marquet 2009, Farley et al. 2018). The result is a potential bias in ecology, and coastal marine ecology in particular, against the development of understanding ecological generality (Underwood et al. 2000). In our study, we have tested the theory of ecological depth zonation for the first time across an ocean-basin scale, via access to standardised, high-resolution ecological and environmental data and a Bayesian hierarchical modelling approach. The results do not indicate that the management of reefs requires an overhaul in respect to identified patterns of depth zonation (nor does the discussion). Instead, we emphasise that many existing widely accepted ecological paradigms that have informed the management of natural systems, to date remain largely untested at sufficient scale.

We thank the Reviewer for prompting us to revisit how we have described this fundamental premise for the paper. To better set-up this context for the reader, we have added to the introduction and discussion sections as follows:

Introduction:

L43: “However, two fundamental issues limit the application of these paradigms. First, scale and predictability². We cannot understand ecological generality without observations of patterns spanning scales that are large enough to encompass relevant environmental gradients². Biophysical processes governing ecological organisation often occur across broad geographies (e.g. regional or continental¹⁰⁻¹³), mediating local community structure¹, and introducing inherent scale-dependent spatial heterogeneity in observed ecological patterns¹⁴⁻¹⁶. To assess generality of ecological theories, a large enough lens is required to capture processes occurring across a land- or seascape of local environmental variation^{1,2}. Without it, perceptual bias is introduced where observed ecological patterns have a unique set of scale-specific causes and consequences^{10,13}, and an understanding of ecological generality is left without empirical grounding².”

L102: “Using a standardised Pacific-wide set of reef fish surveys⁴⁹, composite data on bathymetric steepness, and hierarchical statistical models, we test for the first time whether

depth zonation patterns are generalisable on tropical coral reefs across broad geographies, and compare patterns in locations with and without local human populations.”

Discussion:

L409: “We revisited ecological depth zonation— recognised as a fundamental structuring force of tropical coral reef communities over six decades ago but remains untested at scale^{7,26–28}— with two purposes: first, to test the generality of depth zonation of reef fish biomass for the first time across an ocean-basin scale; and second, to assess whether a zonation paradigm holds on reefs exposed to direct local human impacts.”

- Farley et al. (2018). Situating ecology as a big-data science: Current advances, challenges, and solutions. *Bioscience* 68, 563–576.
- Marquet. (2009). Macroecological perspectives on communities and ecosystems. in *The Princeton guide to ecology* (ed. Levin et al.), pp 386.
- Underwood et al. (2000). Observations in ecology: you can’t make progress on processes without understanding the patterns. *J Exp Mar Biol Ecol* 250(1-2), 97-115.

R3.3: *More significantly this study feels like it misses opportunities to consider depth in the context of other variables, which potentially either excludes the chance of examining some interesting nuances in the data or possibly confounds the analyses. Firstly, human population pressure simply places reefs into one of two categories (populated or unpopulated) which seems overly simplified – why not consider population pressure as a continuous variable and look at more general trends and thresholds? Secondly, the analyses pool all sites from ‘reef slopes’, which standardizes the geomorphological zone but ignores any differences in benthic assemblages. Thus the reader is left wondering, for example, whether populated islands have more homogenous benthic assemblages across depth gradients that limits fish zonation. I’m sure the divers collected some benthic data, so why not include a basic metric like coral cover (or even habitat type) in the analyses to better isolate the depth signal? I appreciate that the aim was to examine a basic biophysical gradient (depth), but I feel the paper would have been more interesting to consider depth in the context of all the other variables that we know affect reef fishes. To me, one of the most interesting parts of this paper was examining the role of steepness on zonation and I kept wanting to know which other factors were critical in explaining the deviance in the data set. One of the benefits of modern ecology is the access to large-scale, reliable data layers of a range of covariates. However, this study doesn’t use these rich sources of information to fill out the models of fish abundance. This is particularly problematic when considering spatial scales of variation (regions / islands / sites) – because the analyses only uses spatial scale in the model the Discussion has to be quite speculative about the actual drivers of these patterns. But had the models included covariates of e.g. net primary productivity, temperature, or wave exposure (all of which are relatively straightforward to obtain) – or island type (see Taylor et al, 2018, *Ecography* 38:520-530 for why this variable might be critical) - then we could have seen actual correlations in the data rather than have to guess at what might be varying from island to island (“potential indicators”, L210). In summary, had I been reviewing this for a general marine journal I would have suggested revisiting the analyses with some key additional covariates and perhaps human pressure as a continuous variable, but recommended the paper eventually be published as examining depth-related patterns with a data set of this size and examining the homogenizing role of human populations is worth quantifying. Plus the importance of steepness and the variation among trophic groups are interesting. However, the threshold for this journal is higher and I am not convinced that the underlying question is sufficiently interesting, especially given the relatively cursory treatment of other important covariates that limits understanding of which variables are critical at large scales.*

We agree - coral reefs are highly complex socio-ecological systems, structured by numerous socioeconomic and biogeophysical factors and processes (Cinner et al. 2016; Hughes et al. 2017; Williams et al. 2019). Well-described interacting natural biophysical processes and human impacts

occur across a range of temporal and spatial scales to determine contemporary ecological organisation of organisms on reefs (Hughes et al. 1999; MacNeil et al. 2009; Cinner et al. 2013; Williams et al. 2015b; Richardson et al. 2018). For example, the Reviewer correctly highlights the critical role of live coral cover in habitat provisioning at small scales (from the coral colony scale to site scale; Richardson et al. 2017) for determining the distribution of reef fishes (Graham and Nash 2013; Williams et al. 2015b; Heenan et al. 2016). Similarly, numerous indices can be used as proxies to estimate the impacts of humans on reefs, including human population density, distance to market, ‘gravity’ (combining human population size and accessibility to reefs), fishing intensity, or a simple binary human population status index (Nadon et al. 2012; Williams et al. 2015b, Williams et al. 2015a; Cinner et al. 2016, 2018; Heenan et al. 2016). Biophysical factors such as temperature, wave energy, primary productivity, and island morphology are also known correlates with reef fish community structure (Williams et al. 2015b; Yeager et al. 2017; Heenan et al. 2019). Indeed, many of these covariates have been identified as critical structuring forces of reef fish assemblages across the Pacific study region (e.g. Williams et al. 2011, Williams et al. 2015b, Yeager et al. 2017, Heenan et al. 2019).

Nonetheless, we *a-priori* designed our study to test a principle ecological theory which until now has been widely assumed but never tested at scale: The generality and predictive capacity of depth on reef fish biomass across an ocean-basin scale, in the presence vs absence of local human populations. We intentionally strove for simplicity by testing our hypotheses (with covariates: depth, steepness, binary ‘humans/no humans’) across a spatial extent characterised by known environmental and anthropogenic variation (Gove et al. 2013; Williams et al. 2015b). In doing so, we were able to quantify and attribute variation (*sensu* MacNeil et al. 2009) in our models to hierarchical spatial scales corresponding to known differential drivers of central and western Pacific reef fish assemblages (Williams et al. 2015b; Heenan et al. 2019). We agree with the Reviewer that it would be an interesting exercise to make use of the rich sources of detailed environmental and anthropogenic information available to explain as much variance in the fish biomass as possible. However, while unquestionably interesting, especially to us coral reef ecologists, this approach would achieve something quite different to an intentionally parsimonious test of ecological theory, with coral reefs as just a model system. Adding a range of known influential covariates would not help to reveal the generality of a depth effect. Rather, it would instead move the study towards using statistics to describe the ‘whole system’, mopping up unexplained noise in the data through the creation a complex ‘causal salad’ (McElreath 2020). Instead, we used a logically constructed framework of *a-priori* defined hypotheses with theoretical basis, with the primary goal of advancing conceptual understanding of an established ecological theory (*sensu* Underwood et al. 2000).

Perhaps our overall intentions in terms of study design and goal were not clear enough, and so we have rephrased the introduction as follows to make our overarching goal and study objectives clearer (thanks for the prompt):

L106: “To explicitly assess generality, we isolate the study focus to test a framework of *a-priori* defined hypotheses of the effects of depth, bathymetric steepness, and human population status on the biomass of reef fishes across a broad spatial extent characterised by known environmental and anthropogenic variation^{8,41}. In doing so, we intentionally exclude other known influential biophysical and anthropogenic covariates on reef fish biomass (e.g. ^{8,50}) to test the predictive capacity of depth at an ocean-basin scale on the biomass of fishes grouped by their major dietary sources⁵¹ – primary consumers, planktivores, secondary consumers, and piscivores.”

- Cinner et al. (2013). Global effects of local human population density and distance to markets on the condition of coral reef fisheries. *Conserv Biol* 27:453–458.
- Cinner et al. (2016). Bright spots among the world’s coral reefs. *Nature* 535(7612), 416-419.
- Cinner et al. (2018). Gravity of human impacts mediates coral reef conservation gains. *Proc Natl Acad Sci* 115:E6116–E6125.
- Fox et al. (2018). Gradients in primary production predict trophic strategies of mixotrophic corals across spatial scales. *Curr Biol* 28:3355-3363.e4.
- Gove et al. (2013). Quantifying climatological ranges and anomalies for Pacific coral reef ecosystems. *PLoS One* 8:e61974.
- Graham and Nash (2013). The importance of structural complexity in coral reef ecosystems. *Coral Reefs* 32:315–326.
- Heenan et al. (2016). Natural bounds on herbivorous coral reef fishes. *Proc R Soc B Biol Sci* 283:20161716.
- Heenan et al. (2019). Natural variation in coral reef trophic structure across environmental gradients. *Front Ecol Environ* 18:69–75.
- Hughes et al. (1999). Patterns of recruitment and abundance of corals along the Great Barrier Reef. *Nature* 397:59–63.
- Hughes et al. (2017). Coral reefs in the Anthropocene. *Nature* 546(7656), 82-90.
- MacNeil et al. (2009). Hierarchical drivers of reef-fish metacommunity structure. *Ecology* 90:252–264.
- McElreath. (2020). *Statistical rethinking: A Bayesian course with examples in R and Stan*. Chapman and Hall/CRC.
- Nadon et al. (2012). Re-creating missing population baselines for Pacific reef sharks. *Conserv Biol* 26:493–503.
- Richardson et al. (2017). Cross-scale habitat structure driven by coral species composition on tropical reefs. *Sci Rep* 7(1), 7557.
- Richardson et al. (2018) Mass coral bleaching causes biotic homogenization of reef fish assemblages. *Glob Chang Biol* 24:3117–3129.
- Underwood et al. (2000). Observations in ecology: you can’t make progress on processes without understanding the patterns. *J Exp Mar Biol Ecol* 250(1-2), 97-115.
- Williams et al. (2011). Differences in reef fish assemblages between populated and remote reefs spanning multiple archipelagos across the Central and Western Pacific. *J Mar Biol* 2011:826234.
- Williams et al. (2015a). Local human impacts decouple natural biophysical relationships on Pacific coral reefs. *Ecography* 38:751–761.
- Williams et al. (2015b) Human, oceanographic and habitat drivers of Central and Western Pacific coral reef fish assemblages. *PLoS One* 10:e0120516.
- Williams et al. (2019). Coral reef ecology in the Anthropocene. *Funct Ecol* 33(6), 1014-1022.
- Yeager et al. (2017). Scale dependence of environmental controls on the functional diversity of coral reef fish communities. *Glob Ecol Biogeogr* 26:1177–1189.

Decision Letter, first revision:

14th April 2023

Dear Dr Richardson,

Your manuscript entitled "Re-visiting the paradigm of depth zonation on contemporary coral reefs"

has now been seen again by the same by 3 reviewers, whose comments are attached. As you can see from the reports, although the reviewers acknowledge the effort to revise the manuscript in response to their previous concerns, they continue to highlight a number of issues which will need to be addressed before we can offer publication in Nature Ecology & Evolution. We will therefore need to once again see your responses to the latest comment and to some editorial concerns, along with a revised manuscript, before we can reach a final decision regarding publication.

In particular, we expect to only need to send your revised version back to Reviewer 2, and we hope that some additional explanation in the revised paper of your data choices and filtering, and exploration of potential covariance, will be satisfactory to that reviewer.

We therefore invite you to revise your manuscript taking into account all reviewer and editor comments. Please highlight all changes in the manuscript text file in Microsoft Word format.

- * If you have not done so already please begin to revise your manuscript so that it conforms to our Article format instructions at <http://www.nature.com/natecolevol/info/final-submission>. Refer also to any guidelines provided in this letter.

[REDACTED]

Nature Ecology & Evolution is committed to improving transparency in authorship. As part of our efforts in this direction, we are now requesting that all authors identified as 'corresponding author' on published papers create and link their Open Researcher and Contributor Identifier (ORCID) with their account on the Manuscript Tracking System (MTS), prior to acceptance. ORCID helps the scientific community achieve unambiguous attribution of all scholarly contributions. You can create and link your ORCID from the home page of the MTS by clicking on 'Modify my Springer Nature account'. For more information please visit www.springernature.com/orcid.

We look forward to seeing the revised manuscript and thank you for the opportunity to review

your work.

[REDACTED]

Reviewers' comments:

Reviewer #1 (Remarks to the Author):

Dear Authors,

I would like to thank you for the thorough revision you undertook to address mine and the other reviewers' comments.

I have only small follow-up questions / suggestions to make based on your revisions.

To better track of those, I am copying below my original comment, your responses, followed by my new response (it will be indicated by saying "Follow-up comment" in front of it).

Alternatively, I have also attached a word doc where I am using a different colour font (dark blue) instead to mark my new comments.

R1.3: Depth zonation patterns: I think most people seeing this term would instinctively think of zonation because of changes in community composition. I think it is important to clarify even in the title that this concerns zonation in biomass and/or trophic groups

Thanks for this. For now, and to favour brevity in the title for the journal, we have not incorporated this suggestion into the title, but we have revised it to make it more concise: 'Re-visiting the paradigm of depth zonation on contemporary coral reefs'. We agree with the Reviewer that further clarification of the term zonation is needed early in the introduction. In addition to the existing definition of ecological zonation as "the distribution of organisms across space" (L58), we have added a more detailed definition of 'depth zonation' at first mention as follows:

L67: "Here we revisit this basic principle in the context of depth zonation of tropical coral reef communities—the distribution of reef fish biomass among distinct trophic groups."

Follow-up comment: I still think that it would be better if the zonation refers to trophic groups, and specifically that of reef fish, so the title could be changed to: "Re-visiting the paradigm of trophic group depth zonation on coral reef fish".

However, no strong feelings there, as I know that papers in high-impact journals tend to favour generic all-encompassing titles.

Results

R1.10: Fig. S1: I would merge it with Fig. 1 as they both show important information. However, I am not wedded to the suggestion.

To aid clearer visualisation of the scaled effects of depth, bathymetric steepness, and human population status on the positive reef fish biomass data, the effect estimates of hurdle components (presence-absence) from the piscivore and planktivore models were not included in Fig. 1. Including them, as presented in Fig. S1, clusters the effect estimates and their percentiles such that the reader may be less able to effectively interpret relative effect size. However, we will gladly exchange Fig. 1 for Fig. S1 on request.

Follow-up comment: I think it's fine to keep Fig. 1 and Fig. S1 separate, but you should only report the effect estimates of hurdle components in Fig. S1, since the top half is a repetition of Fig. 1. Unless it's required for interpreting the figure.

Discussion

R1.18: Lines 304-306: Can you expand a little bit more here? Do you mean that below 30m steepness has been found to have contrasting effects because the steepness in the studies sites was different?

We see that this sentence was potentially unclear. Studies looking at depth zonation beyond the 0–30 m limit in our study report contrasting peaks in abundance of piscivores and planktivores. For example, at Linden Bank, a submerged shoal on the outer shelf of the Great Barrier Reef, reported dominance of piscivores (and mobile invertivores) between 50–70 m depth (Scott et al. 2022). In contrast, the proportion of planktivores on forereefs at Enewetak Atoll in the Marshall Islands increased from 50% at 30 m to >90% from 90–200 m, which the authors suggest may relate to upwelling processes increasing plankton and shallow reef productivity (Thresher and Colin 1986). In L304-306, we suggest that these variable peaks in trophic group biomass at mesophotic depths are potentially indicative of spatial variation in upwelling, which could be linked—among other oceanographic factors—to variable local bathymetric steepness among those study locations. To make this clearer, we have revised the manuscript as follows:

L325: “Previous studies document variable peaks in planktivorous and piscivorous fishes at mesophotic depths beyond the 30 m limit of this study^{54,67}, which may be indicative of spatial variation in upwelling, potentially linked to—among other oceanographic factors—variable local bathymetric steepness among those study locations.”

- Thresher and Colin. (1986). Trophic structure, diversity and abundance of fishes of the deep reef (30–300m) at Enewetak, Marshall Islands. *Bull Mar Sci* 38, 253–272.
- Scott et al. (2022). Variation in abundance, diversity and composition of coral reef fishes with increasing depth at a submerged shoal in the northern Great Barrier Reef. *Rev Fish Biol Fish* 32, 941–962.

Follow-up comment: I would perhaps also touch on the effect of changes in benthic community composition in reef fish trophic structure, similar to some of the comments of Reviewer 3 who asked for the role of benthos in explaining reef-fish patterns. For example, see loss of concurrent loss of herbivorous fish and hard corals in Bermuda (Stefanoudis et al. 2019 – paper you already cite); or the one from Russ et al. 2021 showing how coral cover affects fish trophic structuring in the shallows

Russ, G.R., Rizzari, J.R., Abesamis, R.A. and Alcala, A.C., 2021. Coral cover a stronger driver of reef fish trophic biomass than fishing. *Ecological Applications*, 31(1), p.e02224.

R1.24: General Discussion Comment: Something that was not explored. Why was variation between islands less than between sites or between ecoregions? If site-scale variance is greater than island or ecoregion scale for most groups, then does this mean that conservation efforts should focus on site-level information more compared to information at higher spatial scales?

Thanks to the Reviewer for highlighting where further discussion on cross-scale spatial variation in fish biomass patterns would improve the manuscript. Coral reefs are considered to be highly hierarchical in structure, determined by processes occurring at multiple spatial and temporal scales (Hughes et al. 1999, MacNeil et al. 2009). As the Reviewer rightly highlights, within this hierarchical context, management and governance efforts are considered most effective when carried out at scales aligning with scales of ecological heterogeneity (Cumming and Dobbs 2020). We describe this in detail in the paragraph L335-345, and we link our results to scaled management and conservation efforts in L361-368 in the original manuscript. However, we now provide additional discussion exploring the relative differences in variation observed across scales, notably discussing the relatively lower variation observed at the island-scale as follows:

L389: "These findings align with previous studies that describe habitat composition at the site-level to be the likely dominant driver of reef fish metacommunity structure, including diversity and the biomass of most trophic groups, while attributing greater prevalence of planktivores to larger-scale gradients in overall ocean productivity⁴⁴. That we observed lower variation at the island-scale than site and ecoregion scales may be due to a dominant influence of habitat and background levels of productivity, over processes occurring at the island-scale. In this context, our findings suggest that management of primary consumers, piscivores, total standing biomass, and especially secondary consumers might achieve satisfactory outcomes at local within-island scales with no-take areas⁸⁰, habitat restoration, or better regulated destructive human activities⁷⁹. Local management of planktivores is no doubt also important³⁴. But, given the potential influence of regional-scale drivers on planktivore biomass production and the importance of this group as the prey base for higher trophic levels³², more nuanced, region-specific targets for recovery⁸¹ or take of planktivores may be advisable in areas of naturally lower primary production."

Follow-up comment: In the last sentence after ref. 81, do you mean "or no-take zones of..."
I would also ask the authors to consider the recent findings of Pinheiro et al. 2023.
Pinheiro, H.T., MacDonald, C., Quimbayo, J.P., Shepherd, B., Phelps, T.A., Loss, A.C., Teixeira, J.B. and Rocha, L.A., 2023. Assembly rules of coral reef fish communities along the depth gradient. *Current Biology*.

Based on this, the statement in lines 265-267 could perhaps be modified.

Methods

R1.25: Divers: Where all the surveys conducted by the same team of divers? If not, then authors must acknowledge the variability in species identifications.

Thanks for this important consideration. The National Oceanic and Atmospheric Administration (NOAA) has multiple trained fish survey divers who conduct in situ observations across sampling cruises, across distinct ecoregions and years. Divers must have a minimum of 30 underwater visual fish survey census dives conducted prior to joining a monitoring cruise. In addition, to ensure consistency in observer species identification and size estimation, NOAA employs extensive training, testing, and technical validation protocols both outside of the cruise period and during (detailed in Heenan et al. 2017). These protocols address the potential for intra- and inter-diver variability in two ways:

1. New and experienced divers receive full training in fish identification and survey protocol, in classroom and in-water sessions; prior to each survey cruise, all divers must accurately identify >90% of regional-specific fish species in a test that is specifically weighted towards rare species and those that have conspecifics with similar appearance.
2. For the timeframe of the data used here, between cruises divers conducted in-water training exercises to practice survey protocol, fish identification and fish size estimation (see response to comment R1.27 for detail on size estimation protocol and technical validation).

During the cruises, there are typically 4-10 fish survey divers who routinely discuss and compare species identification and sizes immediately after a survey, and during data entry. Diver pairs are continually rotated, and diver performance is estimated as the difference between the estimates of each diver and those of their dive partner at each site, calculated for total fish biomass, species richness, and body-size distributions of commonly observed species. As divers survey adjacent cylinders on the reef (i.e. not identical areas of the reef), real differences between diver observations are expected. But the diver performance estimate is measured to detect potential consistent bias made by each diver (i.e. if there is no consistent bias, the median difference with their buddy partner should be close to zero). Diver performance is measured every few days during survey cruises to allow for early detection of observer error (Heenan et al. 2017).

In addition to these technical validation protocols, we included a group-level random intercept for 'diver identity' in all of our statistical models of fish biomass to account for any remaining effect of

observer bias. By assuming inherent correlations among divers and their observations that affect the estimated means and associated errors, we were then able to estimate isolated population level effects (i.e. depth, human population status, bathymetric steepness) (sensu Macneil et al. 2015 Nature).

- Heenan et al. (2017). Long-term monitoring of coral reef fish assemblages in the Western central Pacific. *Sci Data* 4(1), 1-12.
- MacNeil et al. (2015). Recovery potential of the world's coral reef fishes. *Nature* 520(7547), 341-344.

We have revised the manuscript as follows to provide the required additional information:

L457: "Surveys were conducted by multiple observers across the study ecoregions and years. NOAA employs extensive training and technical validation protocols to ensure consistency and avoid bias in survey technique, fish species identification, and size estimation⁴⁹. Full details on SPC survey methods and technical validation steps are available in [49]. To further mitigate any confounding effect of observer bias among fish surveys, we included 'diver identity' as a random intercept in all statistical models (described below) (sensu⁵³)."

L514: "A random intercept for 'diver identity' was included to account for any specific effect of observer bias by assuming an inherent correlation structure among divers and their observations (sensu⁵³)."

Follow-up comment: Great, thanks for the explanation and additional information. Based on the revised text above, presumably the multiple observer effect came up as no significant? If so, then perhaps worth mentioning here, and explain that because of this it won't be discussed further in the paper.

Reviewer #2 (Remarks to the Author):

This is my second review of the manuscript submitted by Richardson and coauthors entitled: "Revisiting the paradigm of coral reef depth zonation on contemporary reefs" (NATECOLEVOL-221117855). During my initial review, I felt that the manuscript was well written and provided interesting results. I provided several recommendations for the authors to consider during the revision process. Overall, I appreciate the time and effort that the authors put in to address my recommendations. I know how much time it takes to incorporate suggestions and provide thoughtful responses. In general, I feel that authors made an effort to address my recommendations and made adequate revisions to the manuscript. The theme and structure of the manuscript remains consistent with the initial submission and continues to reinforce what has been shown in other geographies regarding depth zonation patterns in coral reef fishes. The revised version contributes to this body of work by examining these patterns across ecoregions. However, after revisiting this manuscript and evaluating the authors' responses I have a couple additional concerns and recommendations that I would like the authors to consider before the manuscript is considered for publication.

First, one of my recommendations during the initial review focused on the process by which reef fish taxa were filtered and selected in this study. The authors provided a detailed response to my initial comment citing a number of publications to justify why certain piscivores or apex predator groups were omitted from this study. I am familiar with these studies to describe movement patterns or behavioral observations of particular taxa; as well as the studies to describe patterns of reef fish assemblage structure at local, regional, or global scales. In the cited studies describing fish assemblage structure, authors chose to exclude or filter certain taxa based on a priori or a posteriori knowledge of coral reef fishes. In the case of a priori filtering or selection, fish taxa are excluded during the initial study design and in-situ surveys due to methodological limitations of observing certain species or to account for observer inexperience. For example, some monitoring efforts include a subset of large-bodied species or certain ecologically important taxa in

the survey design to reduce the number of species facilitate observers from across a range of experience levels. In studies using a posteriori filtering, species or groups of taxa are excluded during the analysis phase due to limitations in the dataset or to facilitate data comparability across multiple studies using different sampling methodologies.

However, in this study submitted by Richardson and coauthors, the authors chose to filter taxa a posteriori to remove non-reef associated taxa and species that are known to exhibit 'mobbing' behavior in certain locations. While I understand the authors motivation for filtering groups of fishes, it is unclear why entire groups of fishes were omitted when this behavior is documented for only a handful of species. For example, the authors provide a few site-specific or regional examples from the literature where certain taxa (i.e., *Caranx ignobilis* and *Carcharhinus amblyrhynchos*) are known to be abundant and exhibit 'mobbing' behavior in the absence of human fishing pressure. Further, the authors provide an example of personal observations of another common predator (i.e., *Lutjanus bohar*) exhibiting 'mobbing' behavior. However, the authors chose not to omit this species or group (snappers) because the behavior was observed in a region not included in this study. However, *L. bohar*, it is one of the most common and formattable predators on found across coral reefs of the Pacific and it is likely that the observed behavior is limited only to the Line Islands. Regardless, these examples are either species-specific or region-specific. I am therefore finding the authors filtering or selection process to be arbitrary and flawed. I am curious to know why the authors chose to exclude entire groups of ecologically important fishes rather than be more selective in their filtering especially when the groups included in the filtering include upwards of 100 species within each group and 'mobbing' behavior is generally only observed in a handful of species and only in a few locations. I don't want to be difficult but want to encourage the authors as they move forward in their research endeavors to be thoughtful of their selection process when characterizing fish assemblages and describing patterns of reef fishes across coral reef communities.

Second and related to the first, is based on the quality of the dataset included in this study. The authors point out that the reef survey data used in this study were collected by highly skilled divers from NOAA. They point out that divers responsible for collecting quantitative in situ data are trained to estimate the size and abundance of all diurnal and non-cryptic fishes observed in the survey area and make efforts to ensure data are collected with consistency and without biases. I am curious to know why the authors chose to filter or select certain taxa when NOAA invests significant resources into training divers on the survey methods to record all taxa. This includes training divers to avoid double counting individuals entering the survey area or 'mobbing' divers. Further, the standardization of survey protocols across regions and time periods represents one of the most comprehensive datasets for coral reef fishes. e classes. Did the authors attempt perform an initial analysis using all taxa before enlisting filtering? Again, simply picking and choosing to omit ecologically important species or groups likely has important implications to the results.

Third, after reading through the results and examining the figures in more detail, I am curious to know if the authors have considered the non-independence of the predictors (Steepness and Depth). Based on the figures it seems as though depth and steepness covary? Is this a result of methodological limitations where it is not possible to have a survey instance where data are collected at a site that both steep and shallow? It seems as though this could lead to a biased interpretation of the data? Please consider this potential non-independence of the predictors.

Lastly, the authors provide some interpretation of the results in the discussion (lines 360-375) where they posit that the spatial variance of the observations was greatest at the site-scale... indicating that intra-island heterogeneity in habitat availability. However, I encourage the authors to reconsider this statement. Coral reef fish data are inherently noisy and variable at the site and temporal scale. For example, if the authors were to conduct surveys at a single site multiple times the estimates would be highly variable among samples. Therefore the observed results are likely due to variability of the observed data and not directly linked to the local dynamics as they suggest.

Reviewer #3 (Remarks to the Author):

The authors make a vigorous defense of their paper, and to reiterate this is a comprehensive study

of the impacts of depth on fish assemblages – and the impacts of humans on that pattern - and I will cite it when I need to make that point. The problem of stripping away many of the complexities (how does the role of depth compare to other covariates? What is the relationship between growing population size and the depth pattern (not just unpopulated / populated?)) is that the paper lives or dies based on interest in the core question being tested – and I’m not convinced that the relationship between fish and depth is as interesting a component of ecological theory as the authors do. Everything we know about fish assemblages (including many zonation papers) suggests that depth is critical and I don’t think anyone would argue that depth isn’t important. So while this is a comprehensive treatment of the question, and does provide new insights into human impacts and the role of bathymetric steepness, I still struggle to see this as a Nature Ecology & Evolution paper. The paper still makes me wonder about the mechanism (what are fish actually responding to since depth is a proxy for a range of drivers?), what is the relative importance of human direct effects (fishing) versus indirect effects (e.g. affecting coral cover or removing nursery habitats), are there any systematic differences between populated and unpopulated islands, and what is driving the patterns seen at the different spatial scales. But I can see that this concern isn’t shared by the other reviewers and I appreciate that novelty is in the eye of the beholder. So given that there aren’t any critical flaws in the analyses I think at this point I will defer to the Editor to judge the value of the new insights provided by this paper.

*****END*****

Author Rebuttal, first revision:

Reviewer #1 (Remarks to the Author):

I would like to thank you for the thorough revision you undertook to address mine and the other reviewers' comments. I have only small follow-up questions / suggestions to make based on your revisions. To better track of those, I am copying below my original comment, your responses, followed by my new response (it will be indicated by saying "Follow-up comment" in front of it.

Thanks to Reviewer #1 again for their time and thoughtful consideration on our article. The feedback and suggestions have made valuable improvements to the manuscript.

R1.3: *Depth zonation patters: I think most people seeing this term would instinctively think of zonation because of changes in community composition. I think it is important to clarify even in the title that this concerns zonation in biomass and/or trophic groups*

Thanks for this. For now, and to favour brevity in the title for the journal, we have not incorporated this suggestion into the title, but we have revised it to make it more concise: ‘Re-visiting the paradigm of depth zonation on contemporary coral reefs’. We agree with the Reviewer that further clarification of the term zonation is needed early in the introduction. In addition to the existing definition of ecological zonation as “the distribution of organisms across space” (L58), we have added a more detailed definition of ‘depth zonation’ at first mention as follows:

L67: “Here we revisit this basic principle in the context of depth zonation of tropical coral reef communities—the distribution of reef fish biomass among distinct trophic groups.”

Follow-up comment: I still think that it would be better if the zonation refers to trophic groups, and specifically that of reef fish, so the title could be changed to: “Re-visiting the paradigm of trophic group depth zonation on coral reef fish”. However, no strong feelings there, as I know that papers in high-impact journals tend to favour generic all-encompassing titles.

Thanks for this alternative suggestion. Our preferred title is the more concise version ‘Re-visiting the paradigm of depth zonation on contemporary coral reefs’ and we think that because we clarify this point early on in the text, readers will understand early that it is trophic group biomass zonation we refer too. If preferred by the reviewer and editor(s), we will revisit the title to suggest an alternative.

Results

R1.10: *Fig. S1: I would merge it with Fig. 1 as they both show important information. However, I am not wedded to the suggestion.*

To aid clearer visualisation of the scaled effects of depth, bathymetric steepness, and human population status on the positive reef fish biomass data, the effect estimates of hurdle components (presence-absence) from the piscivore and planktivore models were not included in Fig. 1. Including them, as presented in Fig. S1, clusters the effect estimates and their percentiles such that the reader may be less able to effectively interpret relative effect size. However, we will gladly exchange Fig. 1 for Fig. S1 on request.

Follow-up comment: I think it’s fine to keep Fig. 1 and Fig. S1 separate, but you should only report the effect estimates of hurdle components in Fig. S1, since the top half is a repetition of Fig. 1. Unless it’s required for interpreting the figure.

Thanks. We have revised Fig. S1 to present only the effect estimates of the hurdle components as suggested.

Discussion

R1.18: *Lines 304-306: Can you expand a little bit more here? Do you mean that below 30m steepness has been found to have contrasting effects because the steepness in the studies sites was different?*

We see that this sentence was potentially unclear. Studies looking at depth zonation beyond the 0–30 m limit in our study report contrasting peaks in abundance of piscivores and planktivores. For example, at Linden Bank, a submerged shoal on the outer shelf of the Great Barrier Reef, reported dominance of piscivores (and mobile invertivores) between 50–70 m depth (Scott et al. 2022). In contrast, the proportion of planktivores on forereefs at Enewetak Atoll in the Marshall Islands increased from 50% at 30 m to >90% from 90–200 m, which the authors suggest may relate to upwelling processes increasing plankton and shallow reef productivity (Thresher and Colin 1986). In L304-306, we suggest that these variable peaks in trophic group biomass at mesophotic depths are potentially indicative of spatial variation in upwelling, which could be linked—among other oceanographic factors—to variable local bathymetric steepness among those study locations. To make this clearer, we have revised the manuscript as follows:

L325: “Previous studies document variable peaks in planktivorous and piscivorous fishes at mesophotic depths beyond the 30 m limit of this study^{54,67}, which may be indicative of spatial variation in upwelling, potentially linked to—among other oceanographic factors—variable local bathymetric steepness among those study locations.”

- Thresher and Colin. (1986). Trophic structure, diversity and abundance of fishes of the deep reef (30–300m) at Enewetak, Marshall Islands. *Bull Mar Sci* 38, 253–272.

- Scott et al. (2022). Variation in abundance, diversity and composition of coral reef fishes with increasing depth at a submerged shoal in the northern Great Barrier Reef. *Rev Fish Biol Fish* 32, 941–962.

Follow-up comment: I would perhaps also touch on the effect of changes in benthic community composition in reef fish trophic structure, similar to some of the comments of Reviewer 3 who asked for the role of benthos in explaining reef-fish patterns. For example, see loss of concurrent loss of herbivorous fish and hard corals in Bermuda (Stefanoudis et al. 2019 – paper you already cite); or the one from Russ et al. 2021 showing how coral cover affects fish trophic structuring in the shallows

*Russ, G.R., Rizzari, J.R., Abesamis, R.A. and Alcala, A.C., 2021. Coral cover a stronger driver of reef fish trophic biomass than fishing. *Ecological Applications*, 31(1), p.e02224.*

Thanks for this suggestion. We have revised the manuscript as follows:

L315: “Previous studies document variable peaks in planktivorous and piscivorous fishes at mesophotic depths beyond the 30 m limit of this study^{55,68}. These variable peaks may be indicative of spatial variation in upwelling, potentially linked to—among other oceanographic factors and associated changes in benthic composition³⁸—differences in local bathymetric steepness among those study locations.”

The journal limit is 100 references. As we now cite 101 references and the Stefanoudis et al. (2019) study supports the statement, we have elected not to include the additional reference of Russ et al. (2021) for now. But we are grateful for the recommendation, and will be happy to include it if required.

R1.24: *General Discussion Comment: Something that was not explored. Why was variation between islands less than between sites or between ecoregions? If site-scale variance is greater than island or ecoregion scale for most groups, then does this mean that conservation efforts should focus on site-level information more compared to information at higher spatial scales?*

Thanks to the Reviewer for highlighting where further discussion on cross-scale spatial variation in fish biomass patterns would improve the manuscript. Coral reefs are considered to be highly hierarchical in structure, determined by processes occurring at multiple spatial and temporal scales (Hughes et al. 1999, MacNeil et al. 2009). As the Reviewer rightly highlights, within this hierarchical context, management and governance efforts are considered most effective when carried out at scales aligning with scales of ecological heterogeneity (Cumming and Dobbs 2020). We describe this in detail in the paragraph L335-345, and we link our results to scaled management and conservation efforts in L361-368 in the original manuscript. However, we now provide additional discussion exploring the relative differences in variation observed across scales, notably discussing the relatively lower variation observed at the island-scale as follows:

L389: “These findings align with previous studies that describe habitat composition at the site-level to be the likely dominant driver of reef fish metacommunity structure, including diversity and the biomass of most trophic groups, while attributing greater prevalence of planktivores to larger-scale gradients in overall ocean productivity⁴⁴. That we observed lower variation at the island-scale than site and ecoregion scales may be due to a dominant influence of habitat and background levels of productivity, over processes occurring at the island-scale. In this context, our findings suggest that management of primary consumers, piscivores, total standing biomass, and especially secondary consumers might achieve satisfactory outcomes at local within-island scales with no-take areas⁸⁰, habitat restoration, or better regulated destructive human activities⁷⁹. Local management of planktivores is no doubt also important³⁴. But, given the potential influence of regional-scale drivers on planktivore biomass production and the importance of this group as the prey base for higher trophic levels³², more nuanced, region-specific targets for recovery⁸¹ or take of planktivores may be advisable in areas of naturally lower primary production.”

Follow-up comment: In the last sentence after ref. 81, do you mean “or no-take zones of...”.

Thanks for asking for clarification here. In the sentence after ref. 81, in fact we mean ‘take’ to refer to catch-limits. We have revised the manuscript accordingly:

L393: “...more nuanced, region-specific targets for recovery⁸² or catch of planktivores may be advisable in areas of naturally lower primary production.”

I would also ask the authors to consider the recent findings of Pinheiro et al. 2023..

Pinheiro, H.T., MacDonald, C., Quimbayo, J.P., Shepherd, B., Phelps, T.A., Loss, A.C., Teixeira, J.B. and Rocha, L.A., 2023. Assembly rules of coral reef fish communities along the depth gradient. Current Biology.

Based on this, the statement in lines 265-267 could perhaps be modified.

Thanks too for highlighting the recent paper by Pinheiro et al. (2023) which has come out since our previous revision and response. Their findings are relevant and complimentary to ours of course, and as such we now refer to the study in the context of our findings as follows:

L69: “Depth was recognised as a fundamental structuring force over six decades ago^{7,26–28}, and recently shown to be an important predictor of fish diversity as fewer species are found on deeper reefs²⁹.”

L252: “Recent work details declining patterns of reef fish diversity with increasing depth from the shallows to the mesophotic zone (max 150 m depth)²⁹. We build on these findings by, to our knowledge, revealing for the first time a common degree of ecological organisation in relation to both depth and bathymetric steepness across geographically distinct reefs.”

Methods

R1.25: *Divers: Where all the surveys conducted by the same team of divers? If not, then authors must acknowledge the variability in species identifications.*

Thanks for this important consideration. The National Oceanic and Atmospheric Administration (NOAA) has multiple trained fish survey divers who conduct in situ observations across sampling cruises, across distinct ecoregions and years. Divers must have a minimum of 30 underwater visual fish survey census dives conducted prior to joining a monitoring cruise. In addition, to ensure consistency in observer species identification and size estimation, NOAA employs extensive training, testing, and technical validation protocols both outside of the cruise period and during (detailed in Heenan et al. 2017). These protocols address the potential for intra- and inter-diver variability in two ways:

1. New and experienced divers receive full training in fish identification and survey protocol, in classroom and in-water sessions; prior to each survey cruise, all divers must accurately identify >90% of regional-specific fish species in a test that is specifically weighted towards rare species and those that have conspecifics with similar appearance.
2. For the timeframe of the data used here, between cruises divers conducted in-water training exercises to practice survey protocol, fish identification and fish size estimation (see response to comment **R1.27** for detail on size estimation protocol and technical validation).

During the cruises, there are typically 4-10 fish survey divers who routinely discuss and compare species identification and sizes immediately after a survey, and during data entry. Diver pairs are continually rotated, and diver performance is estimated as the difference between the estimates of each diver and those of their dive partner at each site, calculated for total fish biomass, species richness, and body-size distributions of commonly observed species. As divers survey adjacent cylinders on the reef (i.e. not identical areas of the reef), real differences between diver observations are expected. But the diver performance estimate is measured to detect potential consistent bias made by each diver (i.e. if there is no consistent bias, the median difference with their buddy partner should be close to zero). Diver performance is measured every few days during survey cruises to allow for early detection of observer error (Heenan et al. 2017).

In addition to these technical validation protocols, we included a group-level random intercept for 'diver identity' in all of our statistical models of fish biomass to account for any remaining effect of observer bias. By assuming inherent correlations among divers and their observations that affect the estimated means and associated errors, we were then able to estimate isolated population level effects (i.e. depth, human population status, bathymetric steepness) (*sensu* Macneil et al. 2015 *Nature*).

- Heenan et al. (2017). Long-term monitoring of coral reef fish assemblages in the Western central Pacific. *Sci Data* 4(1), 1-12.

- MacNeil et al. (2015). Recovery potential of the world's coral reef fishes. *Nature* 520(7547), 341-344.

We have revised the manuscript as follows to provide the required additional information:

L457: "Surveys were conducted by multiple observers across the study ecoregions and years. NOAA employs extensive training and technical validation protocols to ensure consistency and avoid bias in survey technique, fish species identification, and size estimation⁴⁹. Full details on SPC survey methods and technical validation steps are available in [⁴⁹]. To further mitigate any confounding effect

of observer bias among fish surveys, we included ‘diver identity’ as a random intercept in all statistical models (described below) (*sensu*⁵³).”

L514: “A random intercept for ‘diver identity’ was included to account for any specific effect of observer bias by assuming an inherent correlation structure among divers and their observations (*sensu*⁵³).”

Follow-up comment: Great, thanks for the explanation and additional information. Based on the revised text above, presumably the multiple observer effect came up as no significant? If so, then perhaps worth mentioning here, and explain that because of this it won't be discussed further in the paper.

Thanks to the Reviewer for highlighting where additional clarification would be useful in our description of how any potential residual observer effect was handled. The structure of these models allows us to conservatively assume (and control for) an observer effect, rather than testing for one. By assuming an inherent non-independence of observed fish counts by individual divers, we then control for (centre) any effects of observers to then isolate the estimated population level effects on fish biomass. Apologies for not including this important detail in the previous revisions. We have added it to the newly revised manuscript as follows:

L516: “A group-level random intercept for ‘diver identity’ was included to account for the potential effect of individual observer bias. By assuming an inherent non-independence within divers and their observations that might affect the estimated means and associated errors of fish biomass (*sensu*⁵⁴), we were then able to estimate isolated population level effects (i.e. depth, human population status, bathymetric steepness) (*sensu*⁵⁴). More broadly, by controlling these potential sources of variability, we can more accurately test *a-priori* hypotheses about ecological zonation occurring across spatial scales and with greater inferential strength⁴⁵.”

Reviewer #2 (Remarks to the Author):

This is my second review of the manuscript submitted by Richardson and coauthors entitled: “Revisiting the paradigm of coral reef depth zonation on contemporary reefs” (NATECOLEVOL-221117855). During my initial review, I felt that the manuscript was well written and provided interesting results. I provided several recommendations for the authors to consider during the revision process. Overall, I appreciate the time and effort that the authors put in to address my recommendations. I know how much time it takes to incorporate suggestions and provide thoughtful responses. In general, I feel that authors made an effort to address my recommendations and made adequate revisions to the manuscript. The theme and structure of the manuscript remains consistent with the initial submission and continues to reinforce what has been shown in other geographies regarding depth zonation patterns in coral reef fishes. The revised version contributes to this body of work by examining these patterns across ecoregions. However, after revisiting this manuscript and evaluating the authors’ responses I have a couple additional concerns and recommendations that I would like the authors to consider before the manuscript is considered for publication.

We thank the Reviewer again for generously providing their time and continued careful consideration of our article. We have addressed the additional concerns and recommendations as follows.

R.2ii.1: *First, one of my recommendations during the initial review focused on the process by which reef fish taxa were filtered and selected in this study. The authors provided a detailed response to my initial comment citing a number of publications to justifying why certain piscivores or apex predator groups were omitted from this study. I am familiar with these studies to describe movement patterns or behavioral observations of particular taxa; as well as the studies to describe patterns of reef fish assemblage structure at local, regional, or global scales. In the cited studies describing fish assemblage structure, authors chose to exclude or filter certain taxa based on a priori or a posteriori knowledge of coral reef fishes. In the case of a priori filtering or selection, fish taxa are excluded during the initial study design and in-situ surveys due to methodological limitations of observing certain species or to account for observer inexperience. For example, some monitoring efforts include a subset of large-bodied species or certain ecologically important taxa in the survey design to reduce the number of species facilitate observers from across a range of experience levels. In studies using a posteriori filtering, species or groups of taxa are excluded during the analysis phase due to limitations in the dataset or to facilitate data comparability across multiple studies using different sampling methodologies.*

*However, in this study submitted by Richardson and coauthors, the authors chose to filter taxa a posteriori to remove non-reef associated taxa and species that are known to exhibit ‘mobbing’ behavior in certain locations. While I understand the authors motivation for filtering groups of fishes, it is unclear why entire groups of fishes were omitted when this behavior is documented for only a handful of species. For example, the authors provide a few site-specific or regional examples from the literature where certain taxa (i.e., *Caranx ignobilis* and *Carcharhinus amblyrhynchos*) are known to be abundant and exhibit ‘mobbing’ behavior in the absence of human fishing pressure. Further, the authors provide an example of personal observations of another common predator (i.e., *Lutjanus bohar*) exhibiting ‘mobbing behavior. However, the authors chose not to omit this species or group (snappers) because the behavior was observed in a region not included in this study. However, *L. bohar*, it is one of the most common and formattable predators on found across coral reefs of the*

Pacific and it is likely that the observed behavior is limited only to the Line Islands. Regardless, these examples are either species-specific or region-specific. I am therefore finding the authors filtering or selection process to be arbitrary and flawed. I am curious to know why the authors chose to exclude entire groups of ecologically important fishes rather than be more selective in their filtering especially when the groups included in the filtering include upwards of 100 species within each group and 'mobbing' behavior is generally only observed in a handful of species and only in a few locations. I don't want to be difficult but want to encourage the authors as they move forward in their research endeavors to be thoughtful of their selection process when characterizing fish assemblages and describing patterns of reef fishes across coral reef communities.

We thank Reviewer #2 for these additional contributions. First, apologies for not clarifying sooner that the filtering approach omitted 16 species from the families Carcharhinidae, Carangidae, and Sphyrnidae, as opposed to hundreds as feared. It was an oversight to not make that explicit in the original submission. This has been corrected as detailed below, in the methods text and by including a table in the supplementary information (**Table S12**).

Nonetheless, to address the reviewer's other concerns: we agree that piscivorous fishes such as sharks and large roving jacks are ecologically important to reef trophodynamics (Boaden and Kingsford 2015; Roff et al. 2016), as are other excluded groups such as cryptobenthic reef fishes (Depczynski and Bellwood 2003). We also acknowledge that while the filtering of sharks and large-bodied semi-pelagic piscivores is commonly applied (**Table R2** in Response #1) due to well evidenced systematic bias (e.g. Parrish and Boland 2004; Williams et al. 2015), this approach is not used universally (e.g. DeMartini et al. 2008; Friedlander et al. 2010; Pinheiro et al. 2023); though we note that these cited examples are based on typically more mobile diver transect surveys rather than stationary point count surveys in our study). Nevertheless, the resulting dichotomy in how *in situ* fish assemblage survey data are handled within our field represents an interesting, relevant (and/or potentially frustrating – apologies!) area of discussion about when conservative exclusion of groups mitigates known systematic bias and/or introduces further bias as an artifact of the filtering process.

In consideration of this, we have thought through some options as follows: **Option 1) species-specific filtering** (opposed to group filtering based on shared traits such as body-size, mobility, curiosity, and trophic position; Kulbicki 1998). This approach would require *a priori* evidence of species-specific mobbing behaviour, and/or an observed statistical threshold that we (arguably arbitrarily) deem to represent a mobbing event. Unfortunately, both these data handling choices would more likely produce an artifact than does filtering by group to remove any *potential* mobbers. The most probable issue we foresee from this individual species approach is false negative error, i.e. leaving in a true mobbing species and having that inclusion artificially inflate the piscivore biomass for unpopulated islands compared to populated. In other words, individual species may display unknown and idiosyncratic patterns of spatial variability in mobbing behaviour that do not reach our radar with this approach and therefore we do not see any defensible way to filter some, but not all species within the group. **Option 2) compare candidate piscivore biomass models, both with and without the filtered group of sharks, jacks, barracuda** to assess whether and how observed patterns change as result of filtering.

Due to false negative error issues related to Option 1, we elected to pursue Option 2. We ran the analyses for the piscivore biomass group, both with and without the filtered group. The result was comparable observed zonation patterns in piscivore biomass, suggesting that the reported patterns described in our study are not an artifact of filtering (**Fig. S4**). However, the model outputs of population level effects of depth and bathymetric steepness between populated and unpopulated

locations showed much greater biomass estimates at unpopulated than at populated islands where biomass remained comparable despite differences in filtering. We derived from this that while there may indeed be simply more of these large, roving piscivores on uninhabited reefs (Asher et al. 2017), we are unable to reliably distinguish inflated observation rates due to sheer greater abundance from systematic detectability bias associated with the specific survey method across unpopulated versus populated locations in the study (Parrish and Boland 2004; Williams et al. 2015). Therefore, we determine that the broad observed zonation patterns do not substantially vary as an artifact of filtering (and would unlikely show completely new patterns by including a subset of species). As a result, we can mitigate potential issues of survey bias by conservatively excluding this group to produce more cautious estimates of piscivore biomass patterns (**Fig. S4**).

We recognise that additional explanation and justification of our data filtering to make this clearer is needed, and have added model output figures with both filtered and unfiltered piscivore biomass data to the supplementary information document (**Fig. S4**), presenting posterior estimates for population-level effects (depth, steepness) between populated and unpopulated locations:

L463: “Taxa that are not typically reef-associated were excluded from the analyses, including tuna, bonito, and milkfish (families *Chanidae*, *Myliobatidae*, *Scombridae*; **Table S12**). Sixteen species of shark, jack, and barracuda (families *Carcharhinidae*, *Carangidae*, *Sphyrnidae*) were also excluded from the analyses as these highly mobile, large-bodied, roving piscivores are known to be affected by the presence of stationary divers, typically resulting in systematic over-inflation of visual survey density estimates⁸⁶ (*sensu*^{8,54}; **Table S12**).”

L472: “Zonation patterns in piscivore biomass were comparable with and without this filtering approach. This suggests that the reported patterns in piscivore biomass were not an artifact of the data handling choice to exclude some species known to be affected and systematically overestimated by divers (**Fig. S4**). However, model outputs of population level effects of depth and bathymetric steepness between populated and unpopulated locations showed much greater biomass estimates at uninhabited than habited islands, indicating that conservative exclusion of these species mitigated potential systematic bias associated with the survey method among locations on reported patterns (**Fig. S4**).”

Table S12 Surveyed fish taxa omitted from the study analyses. Reasons for omission include known systematic detectability bias²⁻⁴ (*Carcharhinidae*, *Carangidae*, *Sphyrnidae*), species not typically reef-associated (*Scombridae*, *Chanidae*, *Myliobatidae*), and cryptobenthic eels whose body-size cannot be estimated (*Congridae*, *Muraenidae*).

Family	Species	Common name	Trophic group	Reason for omission
Carangidae	Carangoides ferdau	Blue trevally	Piscivore	Potential overinflation in estimates
	Carangoides orthogrammus	Island trevally	Piscivore	Potential overinflation in estimates
	Caranx ignobilis	Giant trevally	Piscivore	Potential overinflation in estimates
	Caranx lugubris	Black jack	Piscivore	Potential overinflation in estimates
	Caranx melampyus	Bluefin trevally	Piscivore	Potential overinflation in estimates
	Caranx papuensis	Brassy trevally	Piscivore	Potential overinflation in estimates
	Caranx sexfasciatus	Bigeye trevally	Piscivore	Potential overinflation in estimates
	Decapterus macarellus	Mackerel scad	Planktivore	Potential overinflation in estimates
	Elagatis bipinnulata	Rainbow runner	Piscivore	Potential overinflation in estimates
	Scomberoides lysan	Doublespotted queenfish	Piscivore	Potential overinflation in estimates
	Selar crumenophthalmus	Bigeye scad	Planktivore	Potential overinflation in estimates
	Seriola dumerili	Greater amberjack	Piscivore	Potential overinflation in estimates
	Trachinotus bailloni	Smallspotted dart	Piscivore	Potential overinflation in estimates
	Caranx spp.	n/a	Piscivore	Potential overinflation in estimates
Carcharhinidae	Carcharhinus amblyrhynchos	Grey reef shark	Piscivore	Potential overinflation in estimates
	Carcharhinus galapagensis	Galapagos shark	Piscivore	Potential overinflation in estimates
	Carcharhinus melanopterus	Blacktip reef shark	Piscivore	Potential overinflation in estimates
	Triaenodon obesus	Whitetip reef shark	Piscivore	Potential overinflation in estimates
Chanidae	Chanos chanos	Milkfish	Primary consumer	Non-reef-associated
Congridae	Congridae spp.	Conger eel or Garden eel	Piscivore	Cryptobenthic
Muraenidae	Echidna nebulosa	Snowflake moray eel	Secondary consumer	Cryptobenthic
	Enchelycore pardalis	Leopard moray eel	Piscivore	Cryptobenthic
	Gymnomuraena zebra	Zebra moray eel	Secondary consumer	Cryptobenthic
	Gymnothorax breedeni	Blackcheek moray eel	Piscivore	Cryptobenthic
	Gymnothorax eurostus	Abbott's moray eel	Secondary	Cryptobenthic

nature portfolio

			consumer	
	Gymnothorax flavimarginatus	Yellow-edged moray eel	Piscivore	Cryptobenthic
	Gymnothorax javanicus	Giant moray eel	Piscivore	Cryptobenthic
	Gymnothorax melatremus	Dwarf moray eel	Secondary consumer	Cryptobenthic
	Gymnothorax meleagris	Turkey moray eel	Piscivore	Cryptobenthic
	Gymnothorax steindachneri	Steindachner's moray eel	Piscivore	Cryptobenthic
	Gymnothorax undulatus	Undulated moray eel	Piscivore	Cryptobenthic
	Gymnothorax spp.	Moray eel	Piscivore	Cryptobenthic
Myliobatidae	Manta birostris	Giant manta	Planktivore	Non-reef-associated
Scombridae	Euthynnus affinis	Kawakawa	Piscivore	Non-reef-associated
	Gymnosarda unicolor	Dogtooth tuna	Piscivore	Non-reef-associated
	Thunnus albacares	Yellowfin tuna	Piscivore	Non-reef-associated
	Scombridae spp.	Striped bonito	Piscivore	Non-reef-associated
Sphyraenidae	Sphyraena qenie	Blackfin barracuda	Piscivore	Potential overinflation in estimates

Figure S4 Piscivore fish biomass, excluding and including the biomass of *Carcharhinidae*, *Carangidae*, and *Sphyrnidae* (see Table S12), across gradients of depth (a) and bathymetric steepness (b) at unpopulated (colour) and populated (grey) islands. Estimates represent conditional posterior medians (lines), 75% percentiles (shaded areas), and partial residuals (points) at the study mean values of bathymetric steepness (panel a) and depth (panel b). The y axis is limited to 1.05x the maximum value of the 75% CI so partial residuals exceeding axis limits are not displayed. $N = 5,525$ stationary point count (SPC) surveys (across 2,253 forereef sites, 35 islands, five ecoregions).

- Asher J, Williams ID, Harvey ES (2017). An assessment of mobile predator populations along shallow and mesophotic depth gradients in the Hawaiian Archipelago. *Sci Rep* 7:1-18.
- Boaden AE & Kingsford MJ (2015). Predators drive community structure in coral reef fish assemblages. *Ecosphere* 6:1-33.
- DeMartini EE, Friedlander AM, Sandin SA, Sala E (2008) Differences in fish-assemblage structure between fished and unfished atolls in the northern Line Islands, central Pacific. *Mar Ecol Prog Ser* 365:199-215.
- Depczynski M, & Bellwood DR (2003). The role of cryptobenthic reef fishes in coral reef trophodynamics. *Mar Ecol Prog Ser* 256:183-191.
- Friedlander AM, Sandin SA, DeMartini EE, Sala E (2010) Spatial patterns of the structure of reef fish assemblages at a pristine atoll in the central Pacific. *Mar Ecol Prog Ser* 410:219-231.
- Kulbicki. (1998). How the acquired behaviour of commercial reef fishes may influence the results obtained from visual censuses. *J Exp Mar Biol Ecol* 222: 11-30.

- Pinheiro HT, MacDonald C, Quimbayo JP, Shepherd B, Phelps TA, Loss AC, ... & Rocha LA (2023). Assembly rules of coral reef fish communities along the depth gradient. *Curr Biol* 33:1421–1430.
- Roff G, Doropoulos C, Rogers A, Bozec YM, Krueck NC, Aurellado E, ... & Mumby PJ (2016). The ecological role of sharks on coral reefs. *TREE*, 31:395-407.
- Williams ID, Baum JK, Heenan A, Hanson KM, Nadon MO, Brainard RE (2015). Human, oceanographic and habitat drivers of central and western Pacific coral reef fish assemblages. *PLoS ONE* 10: e0120516.

R.2ii.2: *Second and related to the first, is based on the quality of the dataset included in this study. The authors point out that the reef survey data used in this study were collected by highly skilled divers from NOAA. They point out that divers responsible for collecting quantitative in situ data are trained to estimate the size and abundance of all diurnal and non-cryptic fishes observed in the survey area and make efforts to ensure data are collected with consistency and without biases. I am curious to know why the authors chose to filter or select certain taxa when NOAA invests significant resources into training divers on the survey methods to record all taxa. This includes training divers to avoid double counting individuals entering the survey area or ‘mobbing divers. Further, the standardization of survey protocols across regions and time periods represents one of the most comprehensive datasets for coral reef fishes. e classes. Did the authors attempt perform an initial analysis using all taxa before enlisting filtering? Again, simply picking and choosing to omit ecologically important species or groups likely has important implications to the results.*

We are glad that the Reviewer recognises the extensive technical data validation protocols that NOAA employs to minimise measurement error as much as possible (e.g. doing instantaneous counts and training divers to minimise double counting in a single survey). While these protocols ensure that NOAA divers are highly skilled at avoiding double counting, they do not eliminate the problem of spatially variable mobbing behaviour of mobile species because the protocol cannot control for it. For example, these divers could achieve a perfect count of mobbing jacks within an SPC survey cylinder on one island, but still have a systematic bias if the same species avoids those divers at other islands. The NOAA National Coral Reef Monitoring Programme does indeed invest significant resources in collecting this high-quality dataset and this is because the fish data are used for a wide variety of purposes, many of which would not involve or require filtering of species in the way we have for the current paper. For example, managers that have specific questions about *Lutjanus bohar*, sharks, or reef fish assemblage diversity or biomass at their specific islands or regions. We did not take the decision to filter these species out lightly and we agree they are ecologically important. However, because our priority focus was to make Pacific-wide assessment, we exclude known biases among islands driven by strong spatial variance in behaviour among these highly mobile roving piscivore species (per **R.2ii.1**).

R.2ii.3: *Third, after reading through the results and examining the figures in more detail, I am curious to know if the authors have considered the non-independence of the predictors (Steepness and Depth). Based on the figures it seems as though depth and steepness covary? Is this a result of methodological limitations where it is not possible to have a survey instance where data are collected at a site that*

both steep and shallow? It seems as though this could lead to a biased interpretation of the data? Please consider this potential non-independence of the predictors.

Thanks for this question. We checked for non-independence of our predictors (including depth and steepness) for each fish biomass model by plotting bivariate correlations between the posterior samples (MCMC draws) of predictor coefficients and quantifying the Pearson correlation coefficients between paired samples, as follows:

(“b_predictor” relates to the model slope coefficient for that population-level effect; “_c” indicates that the predictor is centred and scaled; SITE_SLOPE_400m_c is the data name for the bathymetric steepness effect)

Planktivore

Correlation coefficients were all <5%, bar one: a single pairwise correlation coefficient for hurdle components depth and steepness in the planktivore model which was still relatively low at 28%. Therefore, we derived that the predictors were sufficiently independent so as to not bias the posterior estimates. All study data and code, including code for these and all other model validation plots are available in this open repository: <https://github.com/LauraERichardson/Depth-Fish>.

We have added these plots to the supplementary information (**Fig. S6**) and revised the methods to make this clearer as follows:

L579: “Non-independence of population-level predictors was assessed by plotting bivariate correlations between the posterior samples (MCMC draws) of predictor coefficients and quantifying Pearson correlation coefficients between paired samples (**Fig. S6**)⁹⁴. Correlation coefficients were all <5%, bar one: a single pairwise correlation coefficient for hurdle components depth and steepness in the planktivore model which was still relatively low at 28%.”

L588: “...independence of model predictors assessed with ggpairs in *GGally* 2.1.2¹⁰¹.”

R.2ii.4: Lastly, the authors provide some interpretation of the results in the discussion (lines 360-375) where they posit that the spatial variance of the observations was greatest at the site-scale... indicating

that intra-island heterogeneity in habitat availability. However, I encourage the authors to reconsider this statement. Coral reef fish data are inherently noisy and variable at the site and temporal scale. For example, if the authors were to conduct surveys at a single site multiple times the estimates would be highly variable among samples. Therefore the observed results are likely due to variability of the observed data and not directly linked to the local dynamics as they suggest.

Thanks for this point. We have revised the manuscript to position our results in the discussion more cautiously as advised:

L368: “We found that the greatest spatial variance was at the site-scale for primary and secondary consumers, piscivores, and total biomass. We note that unmeasured temporal stochasticity at the site-level due to factors like fish recruitment, mobility, or behaviour can influence small-scale single time point observations and their associated variability at that scale⁴⁵. Nonetheless, the importance of site-scale characteristics, indicated by this intra-island heterogeneity, supports numerous studies that identify habitat availability⁷⁹, local hydrodynamics¹⁵, and local disturbances^{23,69,80} as predominant mediators of the biomass of those groups⁴⁵.”

L384: “That we observed lower variation at the island-scale than site and ecoregion scales may be due, in part, to a dominant influence of local variation in habitat, hydrodynamics, or disturbances and variable background levels of productivity across ecoregions, over processes occurring at the island-scale.”

Reviewer #3 (Remarks to the Author):

The authors make a vigorous defense of their paper, and to reiterate this is a comprehensive study of the impacts of depth on fish assemblages – and the impacts of humans on that pattern - and I will cite it when I need to make that point. The problem of stripping away many of the complexities (how does the role of depth compare to other covariates? What is the relationship between growing population size and the depth pattern (not just unpopulated / populated?)) is that the paper lives or dies based on interest in the core question being tested – and I'm not convinced that the relationship between fish and depth is as interesting a component of ecological theory as the authors do. Everything we know about fish assemblages (including many zonation papers) suggests that depth is critical and I don't think anyone would argue that depth isn't important. So while this is a comprehensive treatment of the question, and does provide new insights into human impacts and the role of bathymetric steepness, I still struggle to see this as a Nature Ecology & Evolution paper. The paper still makes me wonder about the mechanism (what are fish actually responding to since depth is a proxy for a range of drivers?), what is the relative importance of human direct effects (fishing) versus indirect effects (e.g. affecting coral cover or removing nursery habitats), are there any systematic differences between populated and unpopulated islands, and what is driving the patterns seen at the different spatial scales. But I can see that this concern isn't shared by the other reviewers and I appreciate that novelty is in the eye of the beholder. So given that there aren't any critical flaws in the analyses I think at this point I will defer to the Editor to judge the value of the new insights provided by this paper.

We thank the reviewer again for thoughtfully engaging with the fit of our overarching study aim and context. We are very pleased that they recognise the study to be comprehensive and robust with no critical flaws, offering novel insight into human impacts on classic depth zonation and the role of bathymetric steepness in mediating reef fish biomass baselines. We are glad that the reviewer highlights that no-one would argue that depth isn't important to the zonation of coral reef fish as this widespread assumption is at the heart our study. For the first time, we provide unequivocal empirical grounding for, until now, the broadly assumed but untested (at sufficient scale) paradigm of depth zonation. With this now well documented through a rigorous test at scale, we are in a better position to embark on tackling the next phase of more complex questions. We fully agree that research into the mechanisms of zonation and how their effects are damped or disrupted by human impacts would be interesting and is indeed the next logical direction for future studies.

Decision Letter, second revision:

3rd July 2023

Dear Laura,

Thank you for submitting your revised manuscript "Re-visiting the paradigm of depth zonation on contemporary coral reefs" (NATECOLEVOL-221117855B). I'm sorry it has taken us longer than desirable to get this decision to you. We had been hoping to get comments from Reviewer 2 on this revision, but we have not heard back from that reviewer. We have discussed your responses to all reviewers' previous comments and feel confident that we can now offer, in principle, to publish it in Nature Ecology & Evolution, pending minor revisions to comply with our editorial and formatting guidelines.

If the current version of your manuscript is in a PDF format, please email us a copy of the file in an editable format (Microsoft Word or LaTeX) -- we can not proceed with PDFs at this stage.

[REDACTED]

Our ref: NATECOLEVOL-221117855B

17th July 2023

Dear Dr. Richardson,

Thank you for your patience as we've prepared the guidelines for final submission of your Nature Ecology & Evolution manuscript, "Re-visiting the paradigm of depth zonation on contemporary coral reefs" (NATECOLEVOL-221117855B). Please carefully follow the step-by-step instructions provided in the attached file, and add a response in each row of the table to indicate the changes that you have made. Please also check and comment on any additional marked-up edits we have proposed within the text. Ensuring that each point is addressed will help to ensure that your revised manuscript can be swiftly handed over to our production team.

****We would like to start working on your revised paper, with all of the requested files and forms, as soon as possible (preferably within two weeks). Please get in contact with us immediately if you anticipate it taking more than two weeks to submit these revised files.****

When you upload your final materials, please include a point-by-point response to any remaining

reviewer comments.

In recognition of the time and expertise our reviewers provide to Nature Ecology & Evolution's editorial process, we would like to formally acknowledge their contribution to the external peer review of your manuscript entitled "Re-visiting the paradigm of depth zonation on contemporary coral reefs". For those reviewers who give their assent, we will be publishing their names alongside the published article.

Nature Ecology & Evolution offers a Transparent Peer Review option for new original research manuscripts submitted after December 1st, 2019. As part of this initiative, we encourage our authors to support increased transparency into the peer review process by agreeing to have the reviewer comments, author rebuttal letters, and editorial decision letters published as a Supplementary item. When you submit your final files please clearly state in your cover letter whether or not you would like to participate in this initiative. Please note that failure to state your preference will result in delays in accepting your manuscript for publication.

Cover suggestions

As you prepare your final files we encourage you to consider whether you have any images or illustrations that may be appropriate for use on the cover of Nature Ecology & Evolution.

Nature Ecology & Evolution has now transitioned to a unified Rights Collection system which will allow our Author Services team to quickly and easily collect the rights and permissions required to publish your work. Approximately 10 days after your paper is formally accepted, you will receive an email in providing you with a link to complete the grant of rights. If your paper is eligible for Open Access, our Author Services team will also be in touch regarding any additional information that may be required to arrange payment for your article.

Please note that *Nature Ecology & Evolution* is a Transformative Journal (TJ). Authors may publish their research with us through the traditional subscription access route or make their paper immediately open access through payment of an article-processing charge (APC). Authors will not be required to make a final decision about access to their article until it has been accepted. [Find out more about Transformative Journals](https://www.springernature.com/gp/open-research/transformative-journals)

Authors may need to take specific actions to achieve [compliance](https://www.springernature.com/gp/open-research/funding/policy-compliance-faqs) with funder and institutional open access mandates. If your research is supported by a funder that requires immediate open access (e.g. according to [Plan S principles](https://www.springernature.com/gp/open-research/plan-s-compliance)) then you should select the gold OA route, and we will direct you to the compliant route where possible. For authors selecting the subscription publication route, the journal's standard licensing terms will need to be accepted, including [self-archiving and license to publish](https://www.nature.com/nature-portfolio/editorial-policies/self-archiving-and-license-to-publish). Those licensing terms will supersede any other terms that the author or any third party may assert apply to any version of the manuscript.

[REDACTED]

[REDACTED]

Reviewer #2:
None

Final Decision Letter:

22nd August 2023

Dear Laura,

Thanks for your patience with the final stages of paperwork for your Article "Local human impacts disrupt depth-dependent zonation of tropical reef fish communities". This email is just to let you know that it has now been accepted for publication in Nature Ecology & Evolution. Thanks also for your patience in the earlier stages - I'm sorry some were slower than we'd usually aim for.

Over the next few weeks, your paper will be copyedited to ensure that it conforms to Nature Ecology and Evolution style. Once your paper is typeset, you will receive an email with a link to choose the appropriate publishing options for your paper and our Author Services team will be in touch regarding any additional information that may be required

Due to the importance of these deadlines, we ask you please us know now whether you will be difficult

to contact over the next month. If this is the case, we ask you provide us with the contact information (email, phone and fax) of someone who will be able to check the proofs on your behalf, and who will be available to address any last-minute problems. Once your paper has been scheduled for online publication, the Nature press office will be in touch to confirm the details.

Acceptance of your manuscript is conditional on all authors' agreement with our publication policies (see www.nature.com/authors/policies/index.html). In particular your manuscript must not be published elsewhere and there must be no announcement of the work to any media outlet until the publication date (the day on which it is uploaded onto our web site).

Please note that *Nature Ecology & Evolution* is a Transformative Journal (TJ). Authors may publish their research with us through the traditional subscription access route or make their paper immediately open access through payment of an article-processing charge (APC). Authors will not be required to make a final decision about access to their article until it has been accepted. [Find out more about Transformative Journals](https://www.springernature.com/gp/open-research/transformative-journals)

Authors may need to take specific actions to achieve [compliance with funder and institutional open access mandates](https://www.springernature.com/gp/open-research/funding/policy-compliance-faqs). If your research is supported by a funder that requires immediate open access (e.g. according to [Plan S principles](https://www.springernature.com/gp/open-research/plan-s-compliance)) then you should select the gold OA route, and we will direct you to the compliant route where possible. For authors selecting the subscription publication route, the journal's standard licensing terms will need to be accepted, including [those licensing terms](https://www.nature.com/nature-portfolio/editorial-policies/self-archiving-and-license-to-publish) will supersede any other terms that the author or any third party may assert apply to any version of the manuscript.

We welcome the submission of potential cover material (including a short caption of around 40 words) related to your manuscript; suggestions should be sent to Nature Ecology & Evolution as electronic files (the image should be 300 dpi at 210 x 297 mm in either TIFF or JPEG format). Please note that such pictures should be selected more for their aesthetic appeal than for their scientific content, and that colour images work better than black and white or grayscale images. Please do not try to design a cover with the Nature Ecology & Evolution logo etc., and please do not submit composites of images related to your work. I am sure you will understand that we cannot make any promise as to whether any of your suggestions might be selected for the cover of the journal.

nature portfolio

You can generate the link yourself when you receive your article DOI by entering it here: <http://authors.springernature.com/share>.

Thanks again for choosing NEE to publish your work. I look forward to seeing it published soon!

[REDACTED]

P.S. Click on the following link if you would like to recommend Nature Ecology & Evolution to your librarian <http://www.nature.com/subscriptions/recommend.html#forms>

** Visit the Springer Nature Editorial and Publishing website at http://editorial-jobs.springernature.com?utm_source=ejp_NEcoE_email&utm_medium=ejp_NEcoE_email&utm_campaign=ejp_NEcoE for more information about our career opportunities. If you have any questions please click [here](mailto:editorial.publishing.jobs@springernature.com).**